# Learning nonparametric latent causal graphs with unknown interventions

**Yibo Jiang**
University of Chicago
yiboj@uchicago.edu

**Bryon Aragam**
University of Chicago
bryon@chicagobooth.edu

## Abstract

We establish conditions under which latent causal graphs are nonparametrically identifiable and can be reconstructed from unknown interventions in the latent space. Our primary focus is the identification of the latent structure in measurement models without parametric assumptions such as linearity or Gaussianity. Moreover, we do not assume the number of hidden variables is known, and we show that at most one unknown intervention per hidden variable is needed. This extends a recent line of work on learning causal representations from observations and interventions. The proofs are constructive and introduce two new graphical concepts—*imaginary subsets* and *isolated edges*—that may be useful in their own right. As a matter of independent interest, the proofs also involve a novel characterization of the limits of edge orientations within the equivalence class of DAGs induced by *unknown* interventions. These are the first results to characterize the conditions under which causal representations are identifiable without making any parametric assumptions in a general setting with unknown interventions and without faithfulness.

## 1  Introduction

Among the many challenges in modern machine learning and artificial intelligence, learning and reasoning about causes and effects from data remains a key challenge. In practice, one of the hurdles that must be overcome is that we often do not have direct access to measurements on causally meaningful variables, and instead can only measure primitive, indirect measurements such as pixel intensities in vision, gene expression values in biology, letters and words in language, or frequency signals in audio. In these applications, it is necessary to first learn representations with meaningful causal signals, a problem known as causal representation learning [52]. Besides learning representations or features of data, we are also interested in understanding what happens when we intervene on these learned features, which is essential for causal reasoning. As such, broadly speaking, causal representation learning can be broken down into two steps: (1) learning high-level features from raw data and (2) learning causal relations between these features.

Given the proliferation of recent work of identifying latent representations in the observational setting [2, 13, 23, 26, 29, 30, 32, 38, 40, 43–45, 51, 55, 63, 66, 70, 71, 73], in this paper we consider the case of interventions. Data arising from interventions opens the door for a causal interpretation of the learned representations, which is often described by a directed acyclic graph (DAG). This setting raises new challenges, namely that the interventions are both *latent* and *unknown*. Moreover, in practical applications, flexible deep neural networks are used to learn nonlinear representations of the data, which necessitates the consideration of nonparametric assumptions. Motivated by these challenges, we seek to answer the following important question:

> *Given a list of latent, unknown interventions, when is it possible to identify the underlying (latent) causal relationships without making parametric assumptions?*

37th Conference on Neural Information Processing Systems (NeurIPS 2023).

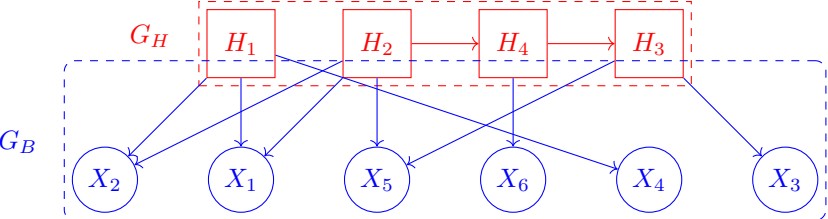

Figure 1: Illustration of the main concepts used in this paper. $G = G_B \cup G_H$ is a measurement model with bipartite DAG $G_B$ (blue edges) and latent DAG $G_H$ (red edges), $\{X_5, X_6\}$ is an imaginary subset, $\{X_1, X_2\}$ is a replaceable subset while $\{X_1, X_2, X_5\}$ is a non-replaceable subset, and $H_2 \to H_4$ is an isolated edge. In fact, $\{X_5, X_6\}$ is a non-replaceable imaginary subset. This is also not a maximal measurement model although it still illustrates the main concepts. For completeness, the undirected dependency graphs under different interventional targets are provided in Figure 2.

In particular, we focus on the problem of learning structure [47, 56], and leave the problem of learning the representations themselves to existing work, since one can then use deep latent variable models to infer the latent distributions from the latent structure, which is well-studied [e.g. 20, 28, 35, 46, 64, 65]. We adopt the measurement model [36, 43, 54] where there are no direct causal edges between observed covariates. This model has been used extensively to model causal representations [e.g. 1, 25, 32, 57, 61, 66–68]. Surprisingly, we show that it is possible to learn latent causal structure without learning the distributions of latent representations, unlike [1, 32, 57, 61]. Furthermore, unlike [1, 57, 61], we do not impose any parametric assumptions: We allow for general noisy, nonlinear transformations between latents and observed as well as arbitrary nonlinear relationships between the latents, which makes the problem significantly harder. Part of our motivation is to better understand the *minimal* assumptions for learning causal structure from unknown interventions.

## 1.1 Contributions

Our main contribution is a set of nonparametric assumptions under which the entire latent causal graph is identifiable given unknown interventions in the latent space, up to edges that we show cannot be oriented without additional assumptions (in a similar sense to reversible edges in a Markov equivalence class). To the best of our knowledge, structure identification in a nonparametric setting given *unknown* and *latent* interventions has not been considered in-depth in the literature.

More specifically, we make the following contributions:

1. We introduce two new graphical concepts—*imaginary subsets* (Definition 3.2) and *isolated edges* (Definition 3.3)—that are key to identifiability and orientability of edges in the true causal graph. These are illustrated in Figure 1 and discussed in detail in Sections 4-5.

2. We combine these concepts with nonparametric, graphical assumptions to show that the causal graph is identifiable up to isolated edges (Theorem 3.4).

3. We show the limitation of edge orientations using CI relations alone under unknown interventions even when there are no latent variables (Theorem 5.3).

The implications of these results are twofold: 1) It *is* possible to learn the entire DAG without making parametric assumptions, albeit at the cost of nontrivial graphical assumptions, and 2) If we wish to relax these graphical conditions, alternative assumptions are needed. That is, in Appendix C (Examples 3-6), we prove that our assumptions are nearly necessary in the sense that if any individual assumption is relaxed, then identifiability fails. Figure 1 illustrates our graphical conditions in a simple example that will be referred back to throughout the paper. Finally, we verify our theoretical results in a simulation study.

## 1.2 Related work

Learning the Markov equivalence class of causal graphs from observational distributions is a well-researched area [56]. Although our focus is on learning measurement models with interventions, we note that the observational case has been extensively studied [25, 32, 36, 43, 54, 66–68].

**Intervention design.** In the classical setting of known interventions on observables, Eberhardt et al. [16] show that for causal graphs with more than $n > 2$ variables, $n - 1$ single node interventions are sufficient, and in the worst case, they are also necessary. Kocaoglu et al. [34] consider intervention design in the presence of latents, and propose efficient algorithms. Hauser and Bühlmann [24] derive a notion of interventional equivalence for hard/structural interventions, while Tian and Pearl [60] and Yang et al. [69] do the same for soft/parametric interventions.

**Unknown interventions.** There has been growing literature on learning under unknown interventions recently as well. A recent line of work studies soft interventions where the unknown interventional targets are observed [7, 15, 19, 21, 27, 33, 49, 58]. In particular, Jaber et al. [27] consider the case where the causal graph consists of measured and unmeasured latent variables, but the intervention targets, though unknown, are from measured variables. Squires et al. [58] assume no hidden variables and use direct $\mathcal{I}$-faithfulness to identify the unknown intervention targets. Perry et al. [49] utilize independent causal mechanisms as a key assumption and measure the number of mechanism changes to identify the true DAG. Castelletti and Peluso [7], Eaton and Murphy [15], and Faria et al. [19] propose Bayesian methods to learn DAGs under unknown interventions.

**Latent interventions.** A very different setting arises when the interventions are both unknown *and* latent. Perhaps the earliest approach to this general setting is *causal feature learning*, introduced in [8, 9]. More closely related to our paper are Liu et al. [42], Squires et al. [57], and Varici et al. [61], which consider unknown interventions on latent variables under parametric assumptions. Squires et al. [57] assume hard interventions on latent variables under a linear model. Varici et al. [61] allow nonlinearities in the latent space with a linear map between hidden and observed, using the score function for identification. Liu et al. [42] assume the existence of an auxiliary observed variable $u$ that modulates the variant weights among latent causal variables where the setup is similar to that of iVAE [31]. Ahuja et al. [1] assume a polynomial decoder and the intervention target on the latent is known. Brehmer et al. [4] consider the weakly supervised setting with paired pre- and post-intervention samples, where the interventions are random and unknown. Lippe et al. [41], on the other hand, work with latent interventions on temporal sequences.

While our work was under review, we were made aware of several concurrent works that study the same problem under different assumptions [5, 37, 72].

## 2 Preliminaries

Our basic setup is a standard graphical model with both observed and latent variables. Appendix A contains a comprehensive overview of all the necessary formal graphical concepts; we briefly outline the basics here. Let $G = (V, E)$ be a directed acyclic graph (DAG) with $V = (X, H)$ where $X$ denotes the observed part and $H$ denotes the hidden or latent part. Define $n := |X|$ and $m := |H|$. For a given node $v$, we use standard notation such as $\mathrm{pa}(v)$, $\mathrm{ch}(v)$, $\mathrm{an}(v)$ and $\mathrm{de}(v)$ for parents, children, ancestors, and descendants respectively. Given a subset $V' \subseteq V$, $\mathrm{pa}(V') := \cup_{v \in V'} \mathrm{pa}(v)$, given two subsets $A, B \subseteq V$, $\mathrm{pa}_B(A) = \mathrm{pa}(A) \cap B$ and given a subgraph $G' \subseteq G$, $\mathrm{pa}_{G'}(V') := \mathrm{pa}(V') \cap G'$. Similar notation can be defined for children, ancestors, and descendants. For disjoint subsets $A, B, C \subseteq V$, define d-$\mathrm{sep}_G(AB \mid C)$ to mean that $A, B$ are $d$-separated by $C$ in the graph $G$. A DAG encodes a set of $d$-separation relations, defined by

$$\mathcal{T}_{V'}(G) = \{\langle A, B, C \rangle : \text{d-sep}_G(AB \mid C) \text{ for disjoint subsets } A, B, C \subseteq V'\}$$

where $V' \subseteq V$. For simplicity, we write $\mathcal{T}(G) = \mathcal{T}_V(G)$. Recall that two DAGs $G_1$ and $G_2$ are Markov equivalent if $\mathcal{T}(G_1) = \mathcal{T}(G_2)$. The Markov equivalence relations define a set of equivalence classes of DAGs called the Markov equivalence class (MEC).

Every distribution $P$ on $V$ defines a collection of conditional independence (CI) relations:

$$\mathcal{T}_{V'}(P) = \{\langle A, B, C \rangle : A \perp\!\!\!\perp B \mid C \text{ in } P \text{ for disjoint subsets } A, B, C \subseteq V'\}$$

where $V' \subseteq V$, and as before $\mathcal{T}(P) = \mathcal{T}_V(P)$. We denote the marginal of $V'$ by $P_{V'}$.

*Remark* 2.1. Throughout the paper, *unconditional* (i.e. marginal) independence and $d$-separation will play a prominent role. Thus, unless otherwise stated, when we say $A$ and $B$ are dependent or $d$-connected, or that there is an active path between $A$ and $B$, without explicit mention of the separating set $C$, we consider it implies that $C = \emptyset$.

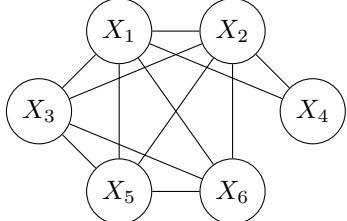

(a) When the intervention target is $\emptyset, \{H_1\}$ or $\{H_2\}$, the UDG contains maximal cliques: $\{1, 2, 3, 5, 6\}, \{1, 2, 4\}$.

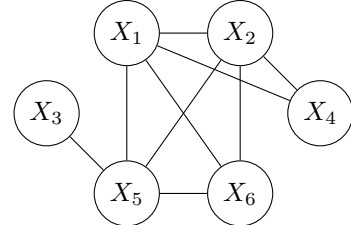

(b) When the intervention target is $\{H_3\}$, the UDG contains maximal cliques: $\{1, 2, 5, 6\}, \{1, 2, 4\}, \{3, 5\}$.

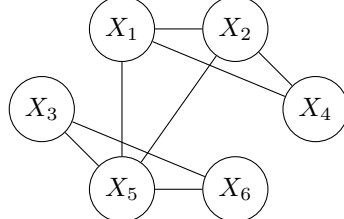

(c) When the intervention target is $\{H_4\}$, the UDG contains maximal cliques: $\{1, 2, 5\}, \{1, 2, 4\}, \{3, 5, 6\}$.

Figure 2: UDG under different intervention targets for Figure 1.

**Interventions.** For background, Eberhardt and Scheines [17] provides a detailed account of the different types of interventions in causal systems. In this paper, we consider *hard* (or *structural*) interventions on a *single node*. Let $I \subseteq V$ be a set of intervention targets. The **intervention graph** of $G$ is the DAG $G^{(I)} = (V, E^{(I)})$, where $E^I := \{(a, b)|(a, b) \in E, b \notin I\}$. Similarly, let $P^{(I)}$ be the interventional distribution. We further restrict intervention targets to be latent variables in this paper ($I \subseteq H$), which necessitates considering *unknown* intervention targets. Thus, we have access to a family of intervention targets $\mathcal{I} = \{I_0, ..., I_k\}$ with $|I_j| \leq 1$. By convention, we let $I_0 = \emptyset$ so that $G^{(I_0)} = G$ and $P^{(I_0)} = P$ is the observational distribution.

A subtle but important point is that different interventional distributions may be the same, i.e. $P_X^{(I)} = P_X^{(I')}$ but $I \neq I'$. This implies, in particular, that the number of latent variables is unknown (see Remark C.1).

**Example 1.** Known and unknown interventions have different implications in terms of edge orientations. To see this, consider the unoriented edge between two variables $X_1$ and $X_2$. If we know the intervention target is $\{X_1\}$ and we observe that $X_1$ and $X_2$ become independent under this intervention, then we know $X_1 \leftarrow X_2$. However, if we do not know the intervention target is $X_1$ and we observe the same independence between $X_1$ and $X_2$, then it is possible the true DAG is $X_1 \leftarrow X_2$ with intervention target $X_1$ or the true DAG is $X_1 \rightarrow X_2$ and the intervention target $X_2$.

**Measurement models.** Our main results apply to so-called *measurement models* [36, 54] in which every observed variable only has incoming edges and no outgoing edges (i.e. $\text{ch}(X) = \emptyset$). This assumption cleanly encapsulates the problem of reconstructing latent causal structure and captures relevant applications where the relationships between raw observations are less relevant than causal features and is a standard model adopted in prior work [e.g. 25, 43, 66–68].

For any measurement model, $G$ decomposes as the union of two subgraphs $G = G_B \cup G_H$ where $G_B$ is a directed, bipartite graph pointing from $H$ to $X$, and $G_H$ is a DAG over the latent variables $H$. See Figure 1.

Following Markham and Grosse-Wentrup [43], for any distribution $P$ over $V$, define the undirected dependency graph (UDG), denoted $D(P)$, to be the undirected graph over $X$ in which there is an edge between $X_i$ and $X_j$ if and only if they are marginally dependent (i.e. given the empty set, cf. Remark 2.1). Clearly, $D(P)$ is easily constructed from $\mathcal{T}_X(P)$ by checking if each pair of observed variables is marginally independent or not.

# 3 Main result

We can now state our goal formally as follows:

*Given a set of interventional distributions $\{P_X^{(I)}\}_{I \in \mathcal{I}}$, can we recover $G_B$ and $G_H$?*

Here, $\{P_X^{(I)}\}_{I \in \mathcal{I}}$ is a *set* and not a tuple: If two different interventions lead to the same interventional distributions, then we only observe one copy of $P_X^{(I)}$ in this set. As a result, we do not know the number of latent variables (see Remark C.1); we show how this is learned in the proof. Of course, in practice, we only have sample access to $P_X^{(I)}$. In this paper, we focus on identifiability and leave estimation from finite samples to future work.

*Remark* 3.1. Learning from the tuple $(P_X^{(I)})_{I \in \mathcal{I}}$ is easier than learning from the set $\{P_X^{(I)}\}_{I \in \mathcal{I}}$, since one can trivially reduce the tuple problem to a set problem by removing duplicates. Thus, there is no loss of generality in our setting.

## 3.1 Assumptions

To solve this problem, we make the following assumptions:

**Assumption 1** (Graphical conditions). The DAG $G$ satisfies the following conditions for every $I \in \mathcal{I}$ and pair of hidden variables $H_i \neq H_j$:

(a) $P^{(I)}$ is Markov with respect to $G^{(I)}$, i.e. $\mathcal{T}(G^{(I)}) \subseteq \mathcal{T}(P^{(I)})$.

(b) $X_i \perp\!\!\!\perp_{P^{(I)}} X_j \implies \text{d-sep}_{G^{(I)}}(\{X_i\}\{X_j\} \mid \emptyset)$, i.e. marginal independence in $X$ implies $d$-separation.

(c) $\text{d-sep}_{G^{(I)}}(\{H_i\}\{H_j\} \mid \emptyset) \implies$ there exists $X_i \in \text{ch}_X(H_i)$ and $X_j \in \text{ch}_X(H_j)$ such that $\text{d-sep}_{G^{(I)}}(\{X_i\}\{X_j\} \mid \emptyset)$.

**Assumption 2** (Complete family of targets). $\mathcal{I} = \{\emptyset, \{H_1\}, \dots, \{H_m\}\}$.

Intuitively, the graphical conditions in Assumption 1 ensure that hidden variables and their dependencies have detectable signatures in observed distributions. Of course, Assumption 1(a) is just the usual Markov assumption that relates the graph $G$ to the distribution $P$. Assumption 1(b) requires that marginal dependencies of observed variables are reflected in the underlying graph, and is much weaker than similar conditions in the graphical modeling literature. Assumption 2 ensures that the effect of each hidden variable is measured. Furthermore, with the exception of Assumption 1(c), which arises from our fully nonparametric setup, each of these assumptions has appeared previously in the literature [1, 14, 18, 32, 43, 48, 53, 59, 61]. See also Remark 4.3. A detailed discussion of these assumptions is deferred to Appendix C. In particular, with the exception of the Markov property Assumption 1(a), we give counterexamples to show that when any *one* of these assumptions is violated (but the rest continue to hold), there are two graphs that have the same set of observed distributions under different interventions.

*Remark* 3.2. It is worth noting that the well-known subset condition [18, 32, 48] is implied by Assumption 1 and Assumption 2 (Lemma C.1 in Appendix C.4). However, this arises from the fact that Assumption 2 is needed for *exact* recovery. If the main objective is instead partial recovery and Assumption 2 is relaxed, then one needs to additionally assume the subset condition. See Appendix C.4 for details.

*Remark* 3.3. Assumption 1 and Assumption 2 also generalize the well-known *pure child* condition, which is widely used and has applications in NLP and topic modeling [e.g. 3, 6, 45, 66]; see also Section 4.3. It is easy to see that the existence of pure children for each latent implies Assumption 1(c).

## 3.2 Maximal measurement model

Under Assumptions 1-2, two different measurement models can still induce exactly the same interventional distributions of observed variables. Even without interventions, there may be ambiguities, an observation that dates back to [50] in the definition of a maximal ancestral graph and was more recently used in [32]. Example 6 in Appendix C.5 gives a concrete example where two measurement

models share the same set of observed distributions under interventions. Fortunately, the ambiguity is limited: There is always a *maximal* measurement model that encodes as many non-redundant dependencies as possible, as defined below:[1]

**Definition 3.1** (Maximal measurement model). A measurement model $G$ is called **maximal** if it satisfies Assumption 1 and the following two conditions:

(a) $\mathrm{pa}(X_i) \neq \emptyset$ for all $i \in [n]$,

(b) There is no DAG $G' = (V, E')$ also satisfying Assumption 1 such that, $\{\mathcal{T}_X(G^{(I)})\}_{I \in \mathcal{I}} = \{\mathcal{T}_X(G'^{(I)})\}_{I \in \mathcal{I}}$, and $E \subsetneq E'$.

Our definition of maximality here mirrors the concept of maximality introduced in [50], extended to include interventions. Throughout the paper, if not mentioned explicitly, the measurement model is considered to be maximal. More discussion on maximality is provided in Appendix C.5.

### 3.3 Imaginary subsets and isolated edges

To identify $G$, we will break the problem into two phases: First, we learn the bipartite graph $G_B$, and then we use this to learn the latent DAG $G_H$. Each of these phases introduces a new graphical concept, which are defined here.

#### 3.3.1 Imaginary subsets

Latent variables induce cliques in the UDG $D(P_X^{(I)})$, which provide a way to identify the existence of a latent [43]. Unfortunately, the identification of $G_B$ is complicated by *imaginary subsets*: Intuitively, imaginary subsets are ambiguous subsets of observed variables that may not be the children of a single latent. Given $D(P_X^{(I)})$, let $\Omega_P^{(I)}$ be the set of maximal cliques in $D(P_X^{(I)})$. We also let $\Omega = \cup_{I \in \mathcal{I}} \Omega_P^{(I)}$.

**Definition 3.2** (Imaginary subsets). A subset $X' \subseteq X$ is a **valid subset** if for all $I \in \mathcal{I}$, there exists a maximal clique $C \in \Omega_P^{(I)}$ such that $X' \subseteq C$. A valid subset is **maximal** if there is no other valid subset $X''$ such that $X' \subsetneq X'' \subseteq C$ for every maximal clique $C \in \Omega$ containing $X'$. A maximal valid subset $X' \subseteq X$ is **imaginary** if there is no hidden variable $H_i$ such that $X' \subseteq \mathrm{ch}_X(H_i)$.

Imaginary subsets (more precisely, the lack thereof) are crucial to the identification of $G_B$, although at first glance they may seem a bit abstract. Indeed, handling imaginary subsets turns out to be a nontrivial issue, so we defer further discussion of this concept to Section 4, where several examples and intuitions are given.

#### 3.3.2 Isolated edges

Once we identify $G_B$, we'd like to identify $G_H$. Unfortunately, Example 1 gave an example where unknown interventions are incapable of orienting an edge using CI information only. This type of edge can be identified more broadly as follows:

**Definition 3.3** (Isolated edge). We say an edge $x \to y$ is **isolated** if $x$ does not have any parent ($\mathrm{pa}(x) = \emptyset$) and $y$ only has $x$ as its parent ($\mathrm{pa}(y) = \{x\}$).

The importance of isolated edges is that these are precisely the edges that cannot be oriented using CI information only (see Section 5.2).

### 3.4 Identifiability of causal graph

We can now state the main result of this paper:

**Theorem 3.4.** *Let $G$ be a maximal measurement model satisfying Assumption 1. Then, given a complete family of interventions (Assumption 2), the following statements are true:*

(a) *If there are no imaginary subsets (Definition 3.2), then $G$ is identifiable from $\{P^{(I)}\}_{I \in \mathcal{I}}$ up to isolated edges (Definition 3.3) in $G_H$;*

---

[1]Here, non-redundant means that the edge encodes a genuine dependence. In other words, adding any more edges would change the underlying model.

*(b) Isolated edges in $G_H$ cannot be oriented using CI information only.*

*In particular, the unknown number of latents $m$ is also identified.*

In other words, $G$ can be maximally identified in the sense that any edge in the latent space that isn't oriented cannot be oriented from the given list of interventions using CI information only: Additional assumptions are needed (e.g. conditional invariances and direct $\mathcal{I}$-faithfulness as in [58]).

We devote a significant effort in the sequel to interpreting and understanding the assumption of no imaginary subsets, which turns out to be subtle and complicated. Thus, in Section 4, we provide several additional sufficient conditions as well as examples of imaginary subsets to help build intuition for this condition.

The proof of this result is broken down into two main steps:

1. Identifying the bipartite graph $G_B$ (See Section 4);
2. Identifying the skeleton of the latent DAG $G_H$ and orienting the edges in $G_H$ (See Section 5);

Sections 4-5 outline the basic ideas behind these constructions. As the ideas are independently interesting and may be more broadly useful beyond just proving Theorem 3.4, we treat these independently.

## 4    Imaginary subsets and the bipartite graph

Theorem 3.4 indicates that as long as there are no imaginary subsets, $G_B$ can be identified. In this section, we show how identifiability of $G_B$ is related to the absence of imaginary subsets (Definition 3.2), and provide several different conditions for this to hold. Throughout this section, we assume as in Theorem 3.4 that $G = G_B \cup G_H$ is a maximal measurement model satisfying Assumption 1, and that Assumption 2 also holds.

### 4.1    Identifying $G_B$ with no imaginary subsets

The following proposition explains why maximal valid subsets (Definition 3.2) are useful:

**Proposition 4.1.** *For any hidden variable $H_i$, $\mathrm{ch}_X(H_i)$ is a maximal valid subset.*

Proposition 4.1 suggests that we assign a latent variable to each maximal valid subset. However, not every maximal valid subset corresponds to a single latent variable: For example, in Figure 1, $\{X_5, X_6\}$ is a maximal valid subset that does not correspond to any hidden variable, and hence is an imaginary subset. Unfortunately, these examples are not pathological; this turns out to be endemic and must be resolved carefully.

The first issue is that a maximal valid subset can be contained in another maximal valid subset. An example is $\{X_1, X_2\} \subset \{X_1, X_2, X_5\}$ in Figure 1 (see also Example 7 in Appendix D). This typically happens when two such subsets appear in different cliques, which does not violate our definition of maximal valid subsets:

**Definition 4.2** (Replaceable subset). A maximal valid subset $X' \subseteq X$ is **replaceable** if there exists another maximal valid subset $X''$ such that $X' \subsetneq X''$.

An example of a replaceable subset is in Figure 1. The advantage of replaceable subsets is that they can be identified and ignored. In particular, any replaceable imaginary subset is a non-issue. Thus, we have the following important result, which is proved in Appendix D.1:

**Theorem 4.3.** *If there are no non-replaceable imaginary subsets, then a subset $X'$ is a non-replaceable maximal valid subset if and only if there exists a hidden variable $H_i$ such that $\mathrm{ch}_X(H_i) = X'$. In particular, it follows that $G_B$ is identifiable.*

Since one can check if $X'$ is replaceable, one might hope that simply eliminating replaceable subsets fixes the problem. Unfortunately, this is not enough: The devil is *non-replaceable imaginary subsets*; i.e. imaginary subsets that cannot be identified from the data. Moreover, non-replaceable imaginary subsets are a genuine phenomenon ($\{X_5, X_6\}$ in Figure 1). Thus, as stated, Theorem 4.3 has two drawbacks:

1. Checking if non-replaceable imaginary subsets exist and getting rid of them is not easy; and
2. Even when there are imaginary subsets, $G_B$ may still be identifiable.

Given the difficulties with non-replaceable imaginary subsets, we will provide two sufficient conditions to guarantee there are no imaginary subsets (Section 4.2) and also show how one can identify $G_B$ even when there are imaginary subsets (Section 4.3).

## 4.2 Sufficient conditions for no imaginary subsets

In this section, we provide two additional sufficient conditions to guarantee there are no imaginary subsets: The first—single source node—is intuitive and interpretable, but cannot always be checked. The second—no fractured subsets—is less intuitive but can be explicitly checked.

**Single latent source.** A latent source is any latent variable such that $\mathrm{pa}(H_i) = \emptyset$. We can show that imaginary subsets do not exist if there is only one latent source. This still allows for arbitrarily many hidden variables $m = |H| > 1$ (i.e. the descendants of the latent source in $G_H$).

**Theorem 4.4.** *Under Assumptions 1-2, if $G_H$ has one latent source, there are no imaginary subsets.*

Therefore, with only one latent source node, we can recover the bipartite graph by Theorem 4.3.

**Fractured subset condition.** A necessary condition for an imaginary subset is that the subset is *fractured*. The definition is somewhat complicated, but worth it since fractured subsets can be identified and checked in practice:

**Definition 4.5** (Fractured subset). Given a collection of maximal valid subsets $\mathcal{S}$, a clique $C$ is called **shattered** by $\mathcal{S}$ if there exists a subset $\mathcal{S}' \subseteq \mathcal{S}$ such that $\cup \mathcal{S}' = C$. A collection of maximal valid subsets $\{S_i\}_{i=1}^{k}$ is **complete** if for every intervention target $I \in \mathcal{I}$, all shattered cliques in $D(P^{(I)})$ form an edge cover of $D(P^I)$. A maximal valid subset $X' \subseteq X$ is **fractured** if there exists a complete collection $\{S_i\}_{i=1}^{k}$ such that $S_i \not\subseteq X'$ for all $S_i$.

Intuitively, a fractured subset provides redundant information as every connection between nodes in the fractured subset can be explained by other maximal valid subsets. The aforementioned necessity is shown by the following lemma:

**Lemma 4.6.** *If a subset $X' \subseteq X$ is imaginary, then $X'$ is also fractured.*

Therefore, we have the following identifiability corollary:

**Corollary 4.7.** *Under Assumptions 1-2 and if there are no fractured subsets, then $G_B$ is identifiable. Furthermore, the absence of fractured subsets can be checked and verified, and if it fails, a certificate is provided.*

*Remark* 4.1. Technically, we only need the condition that there are no *non-replaceable* fractured subsets.

*Remark* 4.2. One might suggest getting rid of all fractured subsets, however, even a non-imaginary subset can be a fractured subset (Example 9 in Appendix D). The no fractured subset condition navïely captures such ambiguities of imaginary subsets, but is overkill. In fact, it is still possible to have identifiability with fractured subsets under different assumptions (Theorem 4.4, Theorem 4.8).

## 4.3 Identifying $G_B$ when there are imaginary subsets

Finally, we show that under the well-known pure child assumption, $G_B$ can be identified even when there are imaginary subsets. The pure child assumption has been made in many existing works [e.g. 3, 6, 45, 66], typically along with additional parametric assumptions.

**Assumption 3** (Pure child). For every $H_i \in H$, there exists at least one $X_i \in X$ with $\mathrm{pa}(X_i) = \{H_i\}$, i.e. $X_i$ only has one parent and that parent is $H_i$.

*Remark* 4.3. Assumption 1(c) is a much weaker assumption than Assumption 3. In particular, if a measurement model satisfies Assumption 3, it also satisfies Assumption 1(c).

**Theorem 4.8.** *Under Assumptions 1-3, the complete collection (cf. Definition 4.5) with the smallest cardinality is exactly $\{\mathrm{ch}_X(H_i)\}_{i=1}^{m}$ and thus $G_B$ can be identified.*

*Remark* 4.4. Under Assumption 3, there still could be imaginary subsets (Example 11 in Appendix D).

# 5 Identifying the latent DAG

Once we have learned the bipartite graph $G_B$, the next step is to learn the DAG $G_H$ over the latent variables $H$. Learning the skeleton of $G_H$ turns out to be straightforward: Assumption 1(b-c) suggest that two hidden variables $H_i$ and $H_j$ are $d$-separated if and only if $\text{ch}_X(H_i)$ and $\text{ch}_X(H_j)$ are in different cliques (Lemma E.2). Therefore, the idea is to use unconditional $d$-separations of latent variables under interventions for identification which is harder than having access to all conditional $d$-separations in the fully observational case (see Appendix E for details).

*Remark* 5.1. In fact, we do not have access to full conditional $d$-separation statements of latents because observed variables are descendants of latent nodes.

The more interesting question is how to orient the edges with *unknown* interventions. Unlike known interventions, edge orientation might not always be possible even when there are no latent variables as shown in Example 1. At the same time, it is sometimes possible as demonstrated by Example 13 in Appendix F. This raises the question of which edges provably *cannot* be oriented under our assumptions: It turns out these are precisely the *isolated* edges (Definition 3.3).

**Theorem 5.1.** *Let $G$ be a maximal measurement model satisfying Assumption 1 and assume we are given a complete family of interventions (Assumption 2) as well as the bipartite DAG $G_B$. Then the true latent DAG $G_H$ is identifiable up to isolated edges. Moreover, isolated edges cannot be oriented without making additional assumptions.*

Thus, as long as we identify $G_B$ (cf. Section 4), we can identify $G_H$ up to isolated edges.

*Remark* 5.2. The proof of Theorem 5.1 in Appendix G provides a constructive algorithm. Pseudocode for the overall approach can be found in Algorithm 2 in Appendix G.

In fact, the non-orientability of isolated edges is not restricted to latent edges or even the measurement model; this fact applies to general, fully (or partially) observed DAGs.

## 5.1 Isolated equivalence

We now introduce an equivalence relation on DAGs that refines the notion of Markov equivalence to account for the extra information conveyed by unknown interventions. Unlike known interventions, which suffice to identify the entire DAG, Example 1 shows that unknown interventions carry strictly less information vs. known interventions. *These definitions are purely graphical* and can be studied in their own right.

For this result, we do not need the measurement model assumptions nor Assumption 1-2. So, for now, consider the case of an arbitrary, fully observed DAG (i.e. $H = \emptyset$). By the transformational properties of MEC [12], we know that two Markov equivalent DAGs can be transformed into one another by a sequence of covered edge reversals (Definition B.3). Similarly, let's define the following:

**Definition 5.2.** (Isolated equivalence class) Two DAGs $G_1$ and $G_2$ are isolated equivalent, denoted $G_1 \sim_E G_2$, if there exists a sequence of isolated edge reversals to transform one into another.

An example of two isolated-equivalent DAGs can be obtained from Figure 1: Since the latent edge $H_2 \to H_4$ is isolated, reversing it yields a DAG that is isolated equivalent to $G$.

*Remark* 5.3. Despite what the name might suggest, an isolated edge $X \to Y$ does not mean that $X$ and $Y$ are disconnected from all other nodes. In fact, $X$ and $Y$ can still have outgoing edges (Definition 3.3) and $X \to Y$ is not just an isolated connected component. Therefore, the IEC is not just a union of disjoint edges.

*Remark* 5.4. Since an isolated edge is covered by definition, $G_1$ and $G_2$ are Markov equivalent if they are isolated equivalent. It is easy to check that isolated equivalence is an equivalence relation. *Therefore, the isolated equivalence class (IEC) is a finer partition of the Markov equivalence class (MEC).* An IEC can and often will be a singleton (i.e. a DAG that is not isolated equivalent to any other DAG in the MEC).

*Remark* 5.5. This is also different from the interventional Markov equivalence class where the intervention target is known [69].

## 5.2 (Non-)Orientability of isolated edges

The value of isolated equivalence is that it identifies which DAGs cannot be distinguished (from CI information alone) using unknown interventions. This is formalized in the following theorem:

**Theorem 5.3.** *Suppose $G_1$ and $G_2$ are in the same IEC. If the tuple $(G_1, \mathcal{I}_1)$ induces $\{\mathcal{T}(G_1^{(I)})\}_{I \in \mathcal{I}_1}$, then there exists a family of interventional targets $\mathcal{I}_2$ (possibly the same as $\mathcal{I}_1$) such that the tuple $(G_2, \mathcal{I}_2)$ also induces $\{\mathcal{T}(G_1^{(I)})\}_{I \in \mathcal{I}_1}$.*

Theorem 5.3 shows that it is impossible to distinguish DAGs in the same IEC by looking at $d$-separations only. While it is impossible to distinguish within an IEC, it is possible to do so between different IECs (see Theorem F.7 in the appendix). Together, Theorems 5.3 and F.7 establish that isolated edges are precisely those edges that cannot be oriented by unknown interventions under our assumptions. This does not imply, of course, that this is impossible in practice: We simply need to impose additional assumptions. For example, one could use conditional invariances to improve the identifiability of edge orientations under additional assumptions like direct $\mathcal{I}$-faithfulness [58], which we have not assumed in this paper.

## 6 Experiments

We test the theoretical results on simulated datasets under two settings: pure child and single latent source. Because this paper is primarily theoretical, the purpose of experiments is simply to verify the theory. In particular, for single latent source experiments, we still adopt the pure child structure but we explicitly test our identification strategy - no imaginary subset (Theorem 4.4). We generate random causal graphs under different settings of $m$, $n$. *We do not enforce Assumption 1 (b) nor maximality (Definition 3.1).* For each variable $V_i$ in the causal graph, the structural equation is simply $V_i \leftarrow \sum_{V_j \in \mathrm{pa}(V_i)} f(V_j) + \epsilon$, where $\epsilon$ is Gaussian noise, and $f$ is a nonlinear function. We set $f$ to be a quadratic function. To test independence, we adopt Chatterjee's coefficient [10]. The metric we use is the Structural Hamming Distance (SHD) between the estimated DAG and the true DAG. The results show that even without the graphical assumptions, our method can be effective in recovering the DAG for nonlinear models.

Table 1: Experiments on simulated data show the effectiveness of our identification theory. The table shows SHD and standard errors are over 100 runs.

| (M,N) | (2, 5) | (3, 8) | (4, 7) | (4, 8) |
|---|---|---|---|---|
| PURE CHILD | $0.01 \pm 0.01$ | $0.54 \pm 0.13$ | $1.35 \pm 0.17$ | $2.35 \pm 0.30$ |
| SINGLE SOURCE | $0.02 \pm 0.02$ | $0.92 \pm 0.18$ | $1.52 \pm 0.21$ | $2.81 \pm 0.30$ |

## 7 Conclusion

Using unknown, latent interventions, we have provided nonparametric identifiability results for a class of graphical measurement models that are commonly used to learn causal representations. For this, we introduced two important graphical concepts: *Imaginary subsets* and *isolated edges*. Our proofs are constructive and can be implemented on finite samples. Obvious relaxations of interest include finding better sufficient conditions for identifying bipartite graphs and extensions to multi-node and/or soft interventions.

## 8 Acknowledgments

The authors would like to acknowledge the support of the NSF via IIS-1956330, the NIH via R01GM140467, and the Robert H. Topel Faculty Research Fund at the University of Chicago Booth School of Business. This work was done in part while B.A. was visiting the Simons Institute for the Theory of Computing.

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

# A Definitions

Let $G = (V, E)$ be a DAG with $V = (X, H)$ where $X$ denotes the observed part and $H$ denotes the hidden or latent part. In the measurement model, $G$ decomposes as the union of two subgraphs $G = G_B \cup G_H$ where $G_B$ is a directed, bipartite graph of edges pointing from $H$ to $X$, and $G_H$ is a DAG over the latent variables $H$. We let $E^B$ be the set of edges in $G_B$ and let $E^H$ be the set of edges in $G_H$. See Figure 1.

For a given node $v$, we use standard notation such as $\mathrm{pa}(v), \mathrm{ch}(v), \mathrm{an}(v), \mathrm{de}(v)$ for parents, children, ancestors, descendants respectively. A node $w$ is a parent of $v$ if there exists an edge $w \to v$. A node $w$ is a child of $v$ if there exists an edge $v \to w$. A node $w$ is an ancestor of $v$ if there exists a directed path from $w$ to $v$. A node $w$ is a descendent of $v$ if there exists a directed path from $v$ to $w$. We also define $\overline{\mathrm{pa}}(v) = \mathrm{pa}(v) \cup \{v\}$, $\overline{\mathrm{ch}}(v) = \mathrm{ch}(v) \cup \{v\}$, $\overline{\mathrm{an}}(v) = \mathrm{an}(v) \cup \{v\}$ and $\overline{\mathrm{de}}(v) = \mathrm{de}(v) \cup \{v\}$. Given a subset $V' \subseteq V$, $\mathrm{pa}(V') := \cup_{v \in V'} \mathrm{pa}(v)$, given two subsets $A, B \subseteq V$, $\mathrm{pa}_B(A) = \mathrm{pa}(A) \cap B$ and given a subgraph $G' \subseteq G$, $\mathrm{pa}_{G'}(V') := \mathrm{pa}(V') \cap G'$. Similar notation can be defined for children, ancestors, and descendants. We use $\mathrm{so}(G')$ to denote the source nodes of $G'$ which are the nodes in $G'$ that do not have incoming edges. We also adopt the convention that $H$ is identified with the indices $[m] = \{1, ..., m\}$, and similarly $X$ is identified with $[n] = \{1, ..., n\}$. In particular, we use $\mathrm{pa}(i)$ and $\mathrm{pa}(H_i)$ interchangeably when the context is clear.

Recall that a node $V_1$ is called a collider if it is in the form

$$V_2 \to V_1 \leftarrow V_3.$$

For disjoint subsets, $A, B, C \subseteq V$, define $\mathrm{d\text{-}sep}_G(AB \,|\, C)$ to mean that $A, B$ are $d$-separated by $C$ in the graph $G$ which means that are no active path between $A$ and $B$ given by $C$.

A path between $A$ and $B$ is active given $C$ if the following hold:

(a) For every collider in the path, either the collider or one of its descendants is in $C$.

(b) $C$ does not include any non-colliders in the path

Recall the definitions of $\mathcal{T}_{V'}(G)$ and $\mathcal{T}_{V'}(P)$ as follows:

$$\mathcal{T}_{V'}(G) = \{\langle A, B, C \rangle : \mathrm{d\text{-}sep}_G(AB \,|\, C) \text{ for disjoint subsets } A, B, C \subseteq V'\}$$

$$\mathcal{T}_{V'}(P) = \{\langle A, B, C \rangle : A \perp\!\!\!\perp B \,|\, C \text{ in } P \text{ for disjoint subsets } A, B, C \subseteq V'\}$$

where $V' \subseteq V$.

# B Preliminaries

**Markov equivalence class.** Usually, there is more than one DAG that encodes the same set of $d$-separations. To formalize this, Markov equivalence can be defined as follows [39]:

**Definition B.1.** Two DAGs $G_1$ and $G_2$ are called Markov equivalent if $\mathcal{T}(G_1) = \mathcal{T}(G_2)$.

In particular, we use $\mathcal{E}(G)$ to denote the Markov equivalence class (MEC) that $G$ belongs to. Fortunately, MEC has a convenient graphical characterization. Recall that the skeleton of a DAG is the undirected graph of the DAG by ignoring its edge orientations. And a $v$-structure in a DAG $G$ is ordered triple of variables $(V_1, V_2, V_3)$ such that (1) $G$ contains edges $V_1 \to V_2$ and $V_3 \to V_2$ and (2) $V_1$ and $V_3$ are not adjacent in $G$.

**Theorem B.2** (Verma and Pearl [62])**.** *Two DAGs are Markov equivalent if and only if they have the same skeletons and the same $v$ structures.*

There is also another transformational characterization of Markov equivalent DAGs [12].

**Definition B.3** (Covered Edge)**.** We say an edge $x \to y$ is covered if $x$ and $y$ share the same parent excluding $x$ (i.e. $\mathrm{pa}(x) \cup \{x\} = \mathrm{pa}(y)$).

**Theorem B.4** (Chickering [12])**.** *If $G_1$ and $G_2$ are Markov equivalent, then $G_1$ and $G_2$ can be transformed into one another by a sequence of covered edge reversals.*

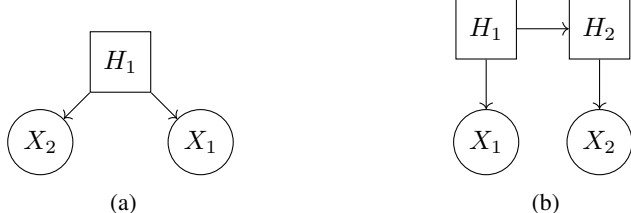

Figure 3: Distinguishable measurement models under latent interventions.

**PDAGs and CPDAGs.** A PDAG is a partially directed acyclic graph that contains both directed and undirected edges [11]. If a DAG $G$ and a PDAG $P$ have the same skeleton and $v$-structures and if every directed edge in $P$ has the same orientations in $G$, then $G$ is called a consistent extension of $P$.

We use completed PDAGs (CPDAG) to represent Markov equivalence classes of DAGs [11]. We define an edge to be compelled if, for every DAG of the Markov equivalence class, this edge has the same orientation. A reversible edge is an edge that is not compelled. Therefore, the completed PDAG of a MEC is a PDAG that has a directed edge for every compelled edge and an undirected edge for every reversible edge. A CPDAG uniquely represents an equivalence class of DAGs and every DAG in the equivalence class is a consistent extension of this CPDAG.

**Measurement models and UDG.** Recall that throughout the paper, we consider a specific type of latent causal graph called the *measurement model* [36, 54] in which there are no direct edges connecting the observed variables.

For any distribution $P$ over $V$, recall the definition of the undirected dependency graph (UDG), denoted $D(P)$, to be the undirected graph over $X$ in which there is an edge between $X_i$ and $X_j$ if and only if they are marginally dependent (i.e. given the empty set, cf. Remark 2.1). Similarly, we can also define UDG for a measurement model $G$, denoted $D(G)$, to be the undirected graph over $X$ in which there is an edge between two $X_i$ and $X_j$ if and only if they are $d$-connected (i.e. by the empty set, cf. Remark 2.1) in $G$. Parallel to the definiton of $\Omega_P^{(I)}$ which is the set of maximal cliques in $D(P_X^{(I)})$, we can also define $\Omega_G^{(I)}$ to the set of maximal cliques in $D(G^{(I)})$. We also let $\Omega = \cup_{I \in \mathcal{I}} \Omega_P^{(I)}$.

Although it is always possible (i.e. by ignoring independencies and allowing for degenerate edges) to represent any latent variable model with densely connected, independent latents, this ignores precisely the latent (causal) structure that we seek to capture. We aim to uncover latent structures that can best capture the observed (in)dependencies. Markham and Grosse-Wentrup [43] show that, with only access to the observational distribution, we only need a minimal measurement model, where every latent variable corresponds to a clique of a minimum clique cover set of $D(G)$ and there is no edge between latent variables, to represent all the dependencies of the observed variables $X$. With interventions, however, such a minimal measurement model may not capture all of the observable dependencies, as illustrated by Example 2.

**Example 2.** Without interventions, the graphs in Figure 3(a) and Figure 3(b) share the same CI statements over $X$, and the first graph would be the one returned as the minimal measurement model. But they do not share the same CI statements under interventions. Specifically, we can intervene on $H_2$ in Figure 3(b) which makes $X_1$ and $X_2$ independent, and this is not possible to achieve in Figure 3(a).

## C  Discussion of assumptions

In this section, we discuss the main assumptions made in this paper (Assumption 1 and Assumption 2). With the exception of the Markov property (Assumption 1(a)), which is standard, we will also show why each assumption is needed by providing a counterexample showing that if each of the other assumptions hold but the assumption of interest fails, there exist two sets of measurement models and interventional targets $(G_1, \mathcal{I}_1)$ and $(G_2, \mathcal{I}_2)$, which can generate identical sets of observational distributions under unknown interventions. This implies, in particular, that none of our assumptions

can be removed without imposing additional assumptions. It is an interesting direction for future work to explore possible alternative assumptions more carefully.

We use the following notation in the examples:

- $\sigma(p)$ is a Bernoulli random variable such that

$$\sigma(p) = \begin{cases} 1 & \text{with probability } p \\ 0 & \text{otherwise} \end{cases}$$

- $\tau(p)$ is a random variable such that

$$\tau(p) = \begin{cases} 1 & \text{with probability } p \\ -1 & \text{otherwise} \end{cases}$$

- $\oplus$ means XOR;
- $\wedge$ means AND.

For ease of reference, we recall our two main assumptions here:

**Assumption 1** (Graphical conditions). The DAG $G$ satisfies the following conditions for every $I \in \mathcal{I}$ and pair of hidden variables $H_i \neq H_j$:

(a) $P^{(I)}$ is Markov with respect to $G^{(I)}$, i.e. $\mathcal{T}(G^{(I)}) \subseteq \mathcal{T}(P^{(I)})$.

(b) $X_i \perp\!\!\!\perp_{P^{(I)}} X_j \implies \text{d-sep}_{G^{(I)}}(\{X_i\}\{X_j\} \mid \emptyset)$, i.e. marginal independence in $X$ implies $d$-separation.

(c) $\text{d-sep}_{G^{(I)}}(\{H_i\}\{H_j\} \mid \emptyset) \implies$ there exists $X_i \in \text{ch}_X(H_i)$ and $X_j \in \text{ch}_X(H_j)$ such that $\text{d-sep}_{G^{(I)}}(\{X_i\}\{X_j\} \mid \emptyset)$.

**Assumption 2** (Complete family of targets). $\mathcal{I} = \{\emptyset, \{H_1\}, \ldots, \{H_m\}\}$.

*Remark* C.1. Although we know the set of intervention targets, we do not know the mapping from targets to distribution. In fact, it is possible that for two intervention targets $I_1$ and $I_2$, we have $P_X^{(I_1)} = P_X^{(I_2)}$. For instance, if the measurement model has multiple source nodes, then intervening on these source nodes would not change the structures of the DAG. And it is not hard to have parametric examples where the distributions are also kept the same (for instance, see Example 4). So the number of elements in the set $\{P_X^{(I)}\}_{I \in \mathcal{I}}$ could be less than $m + 1$. Therefore, we do not even know the number of hidden variables.

In particular, by Assumption 1(a) and (b), $D(P^{(I)})$ is the same as $D(G^{(I)})$ and thus $\Omega_P^{(I)}$ equals $\Omega_G^{(I)}$. We use these concepts interchangeably in this paper.

It is important to highlight that two distributions $P_1$ and $P_2$ over two measurement models $G_1$ and $G_2$ share the same UDG (i.e. $D(P_1) = D(P_2)$) if and only if $\mathcal{T}_X(G_1) = \mathcal{T}_X(G_2)$ (Lemma D.12). Therefore, for Assumption 1(b), (c), and Assumption 2, we in addition show that the two measurement models in counterexamples are also *structurally equivalent* in terms of observed variables, which means that $\{\mathcal{T}_X(G_1^{(I)})\}_{I \in \mathcal{I}_1}$ is the same as $\{\mathcal{T}_X(G_2^{(I)})\}_{I \in \mathcal{I}_2}$.

## C.1   Assumption 1(b)

Assumption 1(b) requires that the UDG $D(P^{(I)})$ correctly encodes the marginal independence structure of $P^{(I)}$. It is a type of "marginal faithfulness" assumption, however, this is a significantly weaker assumption than faithfulness and should not be confused with ordinary faithfulness, which requires that *all* conditional independence statements in $P^{(I)}$ are encoded in the DAG $G^{(I)}$. To appreciate the sizable, difference, note that Assumption 1(b) imposes $O(n^2)$ constraints whereas faithfulness imposes $O(4^n)$ constraints. Similar "marginal"-type assumptions have been invoked previously; see [43, 59] and the references therein. Example 3 shows why Assumption 1(b) is needed. Specifically, because we allow nonlinear functions between observed and latent, without any restrictions, the observed distributions (i.e. over $X$, both observational and interventional) can be arbitrary.

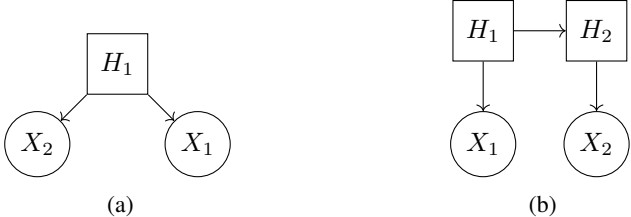

Figure 4: Counterexample for Assumption 1(b)

**Example 3** (Counterexamples for Assumption 1(b)). Consider the two measurement models $G_{(a)}$ and $G_{(b)}$ in Figure 4. We will construct structural causal models for $G_{(a)}$ and $G_{(b)}$ such that they both satisfy Assumption 1(a), and (c). With the complete family of interventional targets (Assumption 2), the interventional distributions of observed variables for both DAGs are the same. Thus, without Assumption 1(b), there are ambiguities.

Specifically, for $G_{(a)}$, we have the following structural equations:

$$
\begin{aligned}
H_1 &\leftarrow \sigma(0.5) \\
X_1 &\leftarrow H_1 \oplus \epsilon_1^a \quad \epsilon_1^a \leftarrow \sigma(0.5) \\
X_2 &\leftarrow H_1 \oplus \epsilon_2^a \quad \epsilon_2^a \leftarrow \sigma(0.5)
\end{aligned}
$$

For $G_{(b)}$, we have the following structural equations:

$$
\begin{aligned}
H_1 &\leftarrow \sigma(0.5) \\
H_2 &\leftarrow H_1 \oplus \epsilon_1^b \quad \epsilon_1^b \leftarrow \sigma(0.5) \\
X_1 &\leftarrow H_1 \oplus \epsilon_2^b \quad \epsilon_2^b \leftarrow \sigma(0.5) \\
X_2 &\leftarrow H_2 \oplus \epsilon_3^b \quad \epsilon_3^b \leftarrow \sigma(0.5)
\end{aligned}
$$

Assumption 1(b) is violated because $X_1$ and $X_2$ are independent in both cases.

Let $P_X^*(X_1, X_2)$ be the joint distribution over $X_1, X_2$ where $X_1$ and $X_2$ are independent bernoulli variables with parameter 0.5.

For $G_{(a)}$, if the intervention target $I$ is $\emptyset$, then $P_X^{(I)} = P_X^*$. If the intervention target $I$ is $\{H_1\}$ but the interventional marginal distribution of $H_1$ is still a Bernoulli distribution, then $P_X^{(I)} = P_X^*$.

For $G_{(b)}$, if the intervention target $I$ is $\emptyset$, then $P_X^{(I)} = P_X^*$. If the intervention target $I$ is $\{H_1\}$ but the interventional marginal distribution of $H_1$ is still a Bernoulli distribution, then $P_X^{(I)} = P_X^*$. If the intervention target $I$ is $\{H_2\}$ and we have $H_2 \leftarrow \sigma(0.5)$, then $P_X^{(I)} = P_X^*$.

Therefore, both $G_{(a)}$ and $G_{(b)}$ can induce $\{P_X^{(I)}\}_{I \in \mathcal{I}} = \{P_X^*\}$.

### C.2 Assumption 1(c)

Assumption 1(c) ensures the relationships between the hidden variables leave observable signatures in the observed data. Of all the assumptions, this one is new to the best of our knowledge and arises due to the fact that (a) We allow for dependencies between the latents, and (b) We make no parametric assumptions. The latter is what crucially distinguishes our approach from previous work that inevitably leverages parametric assumptions to either implicitly guarantee Assumption 1(c) or sidestep it altogether. In the fully nonparametric setting, this cannot be avoided, as shown by Example 4.

**Example 4** (Counterexample for Assumption 1(c)). Consider the two measurement models $G_{(a)}$ and $G_{(b)}$ in Figure 5. We will construct structural causal models for both $G_{(a)}$ and $G_{(b)}$ such that they satisfy Assumption 1(a), (b), and (c). With the complete family of interventional targets (Assumption 2), the interventional distributions of observed variables for both DAGs are the same. In

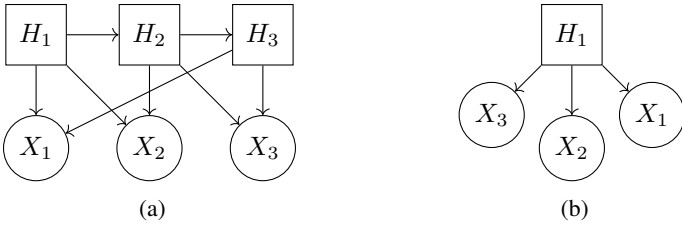

Figure 5: Counterexample for Assumption 1(c)

particular, these two DAGs have the same set of $d$-separation statements of observed variables under interventions. Therefore, Assumption 1(c) is needed to avoid ambiguities.

First of all, let us examine the structure of these two measurement models. For $G_{(a)}$, if the intervention target $I$ is either $\emptyset$, $\{H_1\}$, $\{H_2\}$ or $\{H_3\}$, $X_1$, $X_2$ and $X_3$ always stays $d$-connected. For $G_{(b)}$, if the intervention target $I$ is $\emptyset$ or $\{H_1\}$, $X_1$, $X_2$ and $X_3$ also stays $d$-connected. Therefore, by Lemma D.10, for the complete family of interventional targets $\mathcal{I}_{(a)}$ of $G_{(a)}$ and $\mathcal{I}_{(b)}$ of $G_{(b)}$, $\{\mathcal{T}_X(G_{(a)}^{(I)})\}_{I \in \mathcal{I}_{(a)}}$ is the same as $\{\mathcal{T}_X(G_{(b)}^{(I)})\}_{I \in \mathcal{I}_{(b)}}$.

Now let's consider this parametric example. Specifically, for $G_{(a)}$, we have the following structural equations:

$$
\begin{aligned}
H_1 &\leftarrow \sigma(0.5) \\
H_2 &\leftarrow H_1 \oplus \epsilon_1^a & \epsilon_1^a &\leftarrow \sigma(0.5) \\
H_3 &\leftarrow H_2 \oplus \epsilon_2^a & \epsilon_1^a &\leftarrow \sigma(0.5) \\
X_1 &\leftarrow H_1 + H_2 \\
X_2 &\leftarrow H_2 + H_3 \\
X_3 &\leftarrow H_1 + H_3
\end{aligned}
$$

And for $G_{(b)}$, we have the following structural equations:

$$
\begin{aligned}
\epsilon_1^b &\leftarrow \sigma(0.5) & \epsilon_2^b &\leftarrow \sigma(0.5) & \epsilon_3^b &\leftarrow \sigma(0.5) \\
H_1 &\leftarrow (\epsilon_1^b, \epsilon_2^b, \epsilon_3^b) \\
X_1 &\leftarrow \langle (1,1,0), H_1 \rangle \\
X_2 &\leftarrow \langle (0,1,1), H_1 \rangle \\
X_3 &\leftarrow \langle (1,0,1), H_1 \rangle
\end{aligned}
$$

For $G_{(a)}$, if the intervention target $I$ is $\emptyset$, denote $P_X^{(I)} = P_X^1$. If the intervention target $I$ is $\{H_1\}$ and $H_1 \leftarrow \sigma(0.6)$, denote $P_X^{(I)} = P_X^2$. If the intervention target $I$ is $\{H_2\}$ and $H_2 \leftarrow \sigma(0.5)$ or the intervention target $I$ is $\{H_3\}$ and $H_3 \leftarrow \sigma(0.5)$, we have that $P_X^{(I)} = P_X^1$.

For $G_{(a)}$, if the intervention target $I$ is $\emptyset$, let $P_X^{(I)} = P_X^1$. And if if the intervention target $I$ is $\{H_1\}$ and we set the distribution of $\epsilon_1^b$ to be $\sigma(0.6)$, we also have that $P_X^{(I)} = P_X^2$.

Therefore, both $G_{(a)}$ and $G_{(b)}$ can induce $\{P_X^{(I)}\}_{I \in \mathcal{I}} = \{P_X^1, P_X^2\}$. And one can check that all the other assumptions still hold.

### C.3 Assumption 2

Assumption 2 ensures that the effect of each hidden variable is measured. Example 5 shows why Assumption 2 is needed. This assumption is also made in recent work on causal representation learning [1, 53].

**Example 5** (Counterexample for Assumption 2). Consider the two measurement models $G_{(a)}$ and $G_{(b)}$ in Figure 6. Consider the family of *incomplete* interventional targets $\{\emptyset, \{H_2\}\}$, and assume

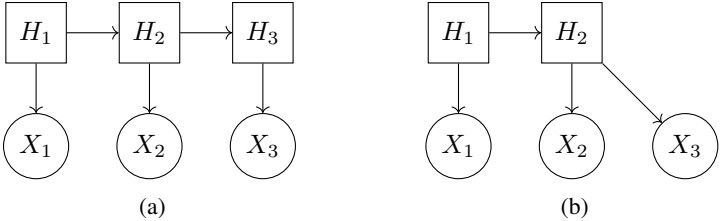

Figure 6: Counterexample for Assumption 2

that $\mathcal{I}_{(a)} = \mathcal{I}_{(b)} = \{\emptyset, \{H_2\}\}$. We will construct structural causal models for both $G_{(a)}$ and $G_{(b)}$ such that they satisfy Assumption 1 and the interventional distributions of observed variables for both DAGs are the same. In particular, these two DAGs have the same set of $d$-separation statements of observed variables under interventions. Therefore, Assumption 2 is needed to avoid ambiguities.

Let us examine the structure of these two measurement models. For $G_{(a)}$, if the intervention target $I$ is $\emptyset$, $X_1$, $X_2$ and $X_3$ are $d$-connected. If the intervention target $I$ is $\{H_2\}$, then only $X_2$ and $X_3$ are $d$-connected. Similarly, for $G_{(b)}$, if the intervention target $I$ is $\emptyset$, $X_1$, $X_2$ and $X_3$ also stays $d$-connected. And if the intervention target $I$ is $\{H_2\}$, then only $X_2$ and $X_3$ are $d$-connected. Therefore, by Lemma D.10, $\{\mathcal{T}_X(G_{(a)}^{(I)})\}_{I\in\mathcal{I}_{(a)}}$ is the same as $\{\mathcal{T}_X(G_{(b)}^{(I)})\}_{I\in\mathcal{I}_{(b)}}$.

Now let's consider this parametric example. Specifically, for $G_{(a)}$, we have the following structural equations:

$$
\begin{aligned}
H_1 &\leftarrow \mathcal{N}(0,1) \\
H_2 &\leftarrow H_1 + \epsilon_1^a \quad \epsilon_1^a \leftarrow \mathcal{N}(0,1) \\
H_3 &\leftarrow H_2 + \epsilon_2^a \quad \epsilon_2^a \leftarrow \mathcal{N}(0,1) \\
X_1 &\leftarrow H_1 \\
X_2 &\leftarrow H_2 \\
X_3 &\leftarrow H_3
\end{aligned}
$$

And for $G_{(b)}$, we have the following structural equations:

$$
\begin{aligned}
H_1 &\leftarrow \mathcal{N}(0,1) \\
H_2 &\leftarrow H_1 + \epsilon_1^b \quad \epsilon_1^b \leftarrow \mathcal{N}(0,1) \\
X_1 &\leftarrow H_1 \\
X_2 &\leftarrow H_2 \\
X_3 &\leftarrow H_2 + \epsilon_2^b \quad \epsilon_2^b \leftarrow \mathcal{N}(0,1)
\end{aligned}
$$

For $G_{(a)}$, if the intervention target $I$ is $\emptyset$, $P_X^{(I)} = \mathcal{N}(0, \Sigma_1)$ and $\Sigma_1 = \begin{pmatrix} 1 & 1 & 1 \\ 1 & 2 & 2 \\ 1 & 2 & 3 \end{pmatrix}$. If the intervention target $I$ is $\{H_2\}$, and $H_2 \leftarrow \mathcal{N}(0, \alpha)$, then $P_X^{(I)} = \mathcal{N}(0, \Sigma_2)$ where $\Sigma_2 = \begin{pmatrix} 1 & 0 & 0 \\ 0 & \alpha & \alpha \\ 0 & \alpha & \alpha+1 \end{pmatrix}$.

On the other hand, for $G_{(b)}$, if the intervention target $I$ is $\emptyset$, $P_X^{(I)} = \mathcal{N}(0, \Sigma_1)$. And If the intervention target $I$ is $\{H_2\}$, and $H_2 \leftarrow \mathcal{N}(0, \alpha)$, then $P_X^{(I)} = \mathcal{N}(0, \Sigma_2)$.

Therefore, both $G_{(a)}$ and $G_{(b)}$ can induce $\{P_X^{(I)}\}_{I\in\mathcal{I}} = \{\mathcal{N}(0, \Sigma_1), \mathcal{N}(0, \Sigma_2)\}$. And one can check that all the other assumptions still hold.

### C.4 Discussion of subset condition

Subset condition is a common assumption used for latent graph identification that dates back to Pearl and Verma [48] (see also [18, 32]. It is defined as follows:

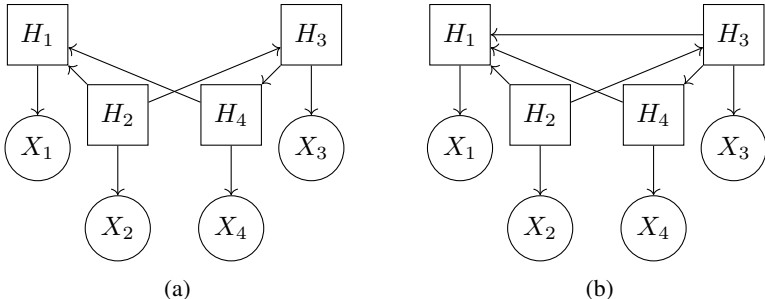

Figure 7: An Example of maximal measurement model

**Assumption** (Subset condition). For any $H_i \neq H_j$, $\mathrm{ch}_X(H_i)$ is not a subset of $\mathrm{ch}_X(H_j)$ and vice versa.

The subset condition ensures that latent variables have observable signatures (i.e. in the observed marginal $P(X)$) and dates back to the 1990s [48], where it was used to study graphical latent variable models, and has more recently been applied on similar problems [18, 32].

It turns out that the subset condition is implied by Assumption 1 and Assumption 2.

**Lemma C.1.** *Under Assumption 1 and Assumption 2, the subset condition always holds.*

*Proof.* Suppose the subset condition is violated, then without loss of generality, there exist two hidden variables $H_i$ and $H_j$ such that $\mathrm{ch}_X(H_i) \subseteq \mathrm{ch}_X(H_j)$. Then by Lemma E.4, there exists an interventional target such that $H_i$ and $H_j$ are $d$-seperated. But this would violate Assumption 1(c). □

It is, however, worth noting that the subset section is implied because of Assumption 2. If the complete family of intervention targets is not assumed to be given, one might need to assume the subset condition. This is because Assumption 1(c) relies on the international targets provided. For example, if the latent graph is complete and the one and only interventional target is the empty set, then Assumption 1(c) is not violated for any graph.

### C.5 Maximal measurement models

Recall the definition of the maximal measurement model:

**Definition 3.1** (Maximal measurement model). A measurement model $G$ is called **maximal** if it satisfies Assumption 1 and the following two conditions:

(a) $\mathrm{pa}(X_i) \neq \emptyset$ for all $i \in [n]$,

(b) There is no DAG $G' = (V, E')$ also satisfying Assumption 1 such that, $\{\mathcal{T}_X(G^{(I)})\}_{I \in \mathcal{I}} = \{\mathcal{T}_X(G'^{(I)})\}_{I \in \mathcal{I}}$, and $E \subsetneq E'$.

Example 6 shows why the maximality condition is needed.

**Example 6** (Maximal measurement model). Consider the two measurement models $G_{(a)}$ and $G_{(b)}$ in Figure 7. We will construct structural causal models for both $G_{(a)}$ and $G_{(b)}$ such that they satisfy Assumption 1 and with a complete family of interventional targets (Assumption 2), the interventional distributions of observed variables for both DAGs are the same. In particular, the CI relations of observed variables are the same under different interventions. Furthermore, $G_{(b)}$ encodes strictly more dependencies than $G_{(a)}$ and is a maximal measurement model. Since these two graphs cannot be distinguished from the set of interventional distributions $\{P_X^{(I)}\}_{I \in \mathcal{I}}$ alone, and one encodes more information than the other, this motivates why we consider maximal measurement models.

Let's first examine the structure of these two measurement models and show that $G_{(b)}$ is maximal.

For $G_{(a)}$, if the intervention target $I$ is $\emptyset$, $X_1$, $X_2$, $X_3$ and $X_4$ are $d$-connected. If the intervention target $I$ is $\{H_1\}$, then only $X_2, X_3, X_4$ are $d$-connected. If the intervention target $I$ is $\{H_2\}$, then

$X_1$, $X_2$, $X_3$ and $X_4$ are $d$-connected. If the intervention target $I$ is $\{H_3\}$, then $X_1$, $X_3$, $X_4$ are $d$-connected and $X_2$ is only $d$-connceted to $X_1$. If the intervention target $I$ is $\{H_4\}$, then $X_1$, $X_2$, $X_3$ are $d$-connected and $X_4$ is only $d$-connceted to $X_1$.

Similarly, for $G_{(b)}$, if the intervention target $I$ is $\emptyset$, if the intervention target $I$ is $\emptyset$, $X_1$, $X_2$, $X_3$ and $X_4$ are $d$-connected. If the intervention target $I$ is $\{H_1\}$, then only $X_2$, $X_3$, $X_4$ are $d$-connected. If the intervention target $I$ is $\{H_2\}$, then $X_1$, $X_2$, $X_3$ and $X_4$ are $d$-connected. If the intervention target $I$ is $\{H_3\}$, then $X_1$, $X_3$, $X_4$ are $d$-connected and $X_2$ is only connceted to $X_1$. If the intervention target $I$ is $\{H_4\}$, then $X_1$, $X_2$, $X_3$ are $d$-connected and $X_4$ is only connceted to $X_1$.

Therefore, by Lemma D.10, for the complete family of interventional targets $\mathcal{I}_{(a)}$ of $G_{(a)}$ and $\mathcal{I}_{(b)}$ of $G_{(b)}$, $\{\mathcal{T}_X(G_{(a)}^{(I)})\}_{I \in \mathcal{I}_{(a)}}$ is the same as $\{\mathcal{T}_X(G_{(b)}^{(I)})\}_{I \in \mathcal{I}_{(b)}}$.

One can also check $G_{(b)}$ is maximal because adding additional edges to $G_{(b)}$ would lead to different $d$-separation statements of observed variables under interventions.

Now we construct structural causal models for each DAG. Specifically, for $G_{(a)}$, we have the following structural equations:

$$
\begin{aligned}
H_2 &\leftarrow \epsilon_1^a & \epsilon_1^a &\leftarrow \sigma(0.5) \\
H_3 &\leftarrow H_2 \epsilon_2^a & \epsilon_2^a &\leftarrow \tau(0.5) \\
H_4 &\leftarrow H_3 \epsilon_3^a & \epsilon_3^a &\leftarrow \sigma(0.5) \\
H_1 &\leftarrow H_2 H_4 \epsilon_4^a & \epsilon_4^a &\leftarrow \sigma(0.5) \\
X_1 &\leftarrow H_1 \\
X_2 &\leftarrow H_2 \\
X_3 &\leftarrow H_3 \\
X_4 &\leftarrow H_4
\end{aligned}
$$

And for $G_{(b)}$, we have the following structural equations:

$$
\begin{aligned}
H_2 &\leftarrow \epsilon_1^b & \epsilon_1^b &\leftarrow \sigma(0.5) \\
H_3 &\leftarrow H_2 \epsilon_2^b & \epsilon_2^b &\leftarrow \tau(0.5) \\
H_4 &\leftarrow H_3 \epsilon_3^b & \epsilon_3^b &\leftarrow \sigma(0.5) \\
H_1 &\leftarrow H_2 H_3^2 H_4 \epsilon_4^b & \epsilon_4^b &\leftarrow \sigma(0.5) \\
X_1 &\leftarrow H_1 \\
X_2 &\leftarrow H_2 \\
X_3 &\leftarrow H_3 \\
X_4 &\leftarrow H_4
\end{aligned}
$$

For $G_{(a)}$, if the intervention target $I$ is $\emptyset$, then $X_2 = \epsilon_1^a$, $X_3 = \epsilon_1^a \epsilon_2^a$, $X_4 = \epsilon_1^a \epsilon_2^a \epsilon_3^a$ and $X_1 = \epsilon_1^a \epsilon_2^a \epsilon_3^a \epsilon_4^a$.

For $G_{(b)}$, if the intervention target $I$ is $\emptyset$, then $X_2 = \epsilon_1^b$, $X_3 = \epsilon_1^b \epsilon_2^b$, $X_4 = \epsilon_1^b \epsilon_2^b \epsilon_3^b$ and $X_1 = \epsilon_1^b \epsilon_2^b \epsilon_3^b \epsilon_4^b$.

For $G_{(a)}$, if the intervention target is $\{H_2\}$ and $H_2 \leftarrow \epsilon_1'$ where $\epsilon_1'$ is another Bernoulli random variable, then $X_2 = \epsilon_1'$, $X_3 = \epsilon_1' \epsilon_2^a$, $X_4 = \epsilon_1' \epsilon_2^a \epsilon_3^a$ and $X_1 = \epsilon_1' \epsilon_2^a \epsilon_3^a \epsilon_4^a$.

For $G_{(b)}$, if the intervention target is $\{H_2\}$ and $H_2 \leftarrow \epsilon_1'$ where $\epsilon_1'$ is another Bernoulli random variable, then $X_2 = \epsilon_1'$, $X_3 = \epsilon_1' \epsilon_2^b$, $X_4 = \epsilon_1' \epsilon_2^b \epsilon_3^b$ and $X_1 = \epsilon_1' \epsilon_2^b \epsilon_3^b \epsilon_4^b$.

For $G_{(a)}$, if the intervention target is $\{H_3\}$ and $H_3 \leftarrow \epsilon_2'$ where $\epsilon_2'$ is another Bernoulli random variable, then $X_2 = \epsilon_1^a$, $X_3 = \epsilon_2'$, $X_4 = \epsilon_2' \epsilon_3^a$ and $X_1 = \epsilon_1^a \epsilon_2' \epsilon_3^a \epsilon_4^a$.

For $G_{(b)}$, if the intervention target is $\{H_3\}$ and $H_3 \leftarrow \epsilon_2'$ where $\epsilon_2'$ is another Bernoulli random variable, then $X_2 = \epsilon_1^b$, $X_3 = \epsilon_2'$, $X_4 = \epsilon_2' \epsilon_3^b$ and $X_1 = \epsilon_1^b \epsilon_2' \epsilon_3^b \epsilon_4^b$.

For $G_{(a)}$, if the intervention target is $\{H_4\}$ and $H_4 \leftarrow \epsilon_3'$ where $\epsilon_3'$ is another Bernoulli random variable, then $X_2 = \epsilon_1^a$, $X_3 = \epsilon_1^a \epsilon_2^a$, $X_4 = \epsilon_3'$ and $X_1 = \epsilon_1^a \epsilon_3' \epsilon_4^a$.

For $G_{(b)}$, if the intervention target is $\{H_4\}$ and $H_4 \leftarrow \epsilon_3'$ where $\epsilon_3'$ is another Bernoulli random variable, then $X_2 = \epsilon_1^b$, $X_3 = \epsilon_1^b \epsilon_2^b$, $X_4 = \epsilon_3'$ and $X_1 = \epsilon_1^b \epsilon_3' \epsilon_4^b$.

For $G_{(a)}$, if the intervention target $I$ is $\{H_1\}$ and $H_1 \leftarrow \epsilon_4'$, then $X_2 = \epsilon_1^a$, $X_3 = \epsilon_1^a \epsilon_2^a$, $X_4 = \epsilon_1^a \epsilon_2^a \epsilon_3^a$ and $X_1 = \epsilon_4'$.

For $G_{(b)}$, if the intervention target $I$ is $\{H_1\}$ and $H_1 \leftarrow \epsilon_4'$, then $X_2 = \epsilon_1^b$, $X_3 = \epsilon_1^b \epsilon_2^b$, $X_4 = \epsilon_1^b \epsilon_2^b \epsilon_3^b$ and $X_1 = \epsilon_4'$.

Note that the four pairs of $(\epsilon_1^a, \epsilon_1^b)$, $(\epsilon_2^a, \epsilon_2^b)$, $(\epsilon_3^a, \epsilon_3^b)$ and $(\epsilon_4^a, \epsilon_4^b)$ have the same distributions.

Therefore, both $G_{(a)}$ and $G_{(b)}$ can induce the same $\{P_X^{(I)}\}_{I \in \mathcal{I}}$. And one can check that Assumption 1 still hold.

In general, there are multiple DAGs that are Markov to a given distribution, and the question is how do we decide on the correct "minimal" representation. Without latents, there is no ambiguity: We can always test all possible CI relations and obtain a complete picture to obtain a minimal I-map. With latents, we must be careful:

- If we can check CI relations over all of $(X, H)$, then the usual notion of a minimal I-map prevails. But in practice, we cannot access $P(X, H)$ since H is unobserved.

- Thus, in practice, we should restrict our attention to information about $P(X)$ only. In this case, we argue that we should only remove an edge if its removal can be justified on the basis of information about $P(X)$ only. This is the essence of maximality: We only remove an edge if it follows from the observed data $X$. Otherwise, we remain agnostic: We do not want to remove an edge that may in fact reflect a "real" dependence over H.

This encapsulates the concept of maximality, and the underlying intuition is akin to that of maximal ancestral graphs. As a result, our characterization of the maximal measurement model aligns with the core principles of the maximal ancestral graph. Measurement models have latent variables and we only have access to partial information (ie., observed variables). Since two measurement models can encode the same set of conditional independencies over $X$ and the absence of edges encodes nontrivial information, the removal of an edge should be justified carefully on the data we have available.

# D  Proofs for Section 4

In this section, we provide all proofs that were left out in Section 4. Recall that we assume that $G$ is a maximal measurement model (Definition 3.1) satisfying Assumption 1, and that Assumption 2 also holds.

This first proposition shows that maximal valid subsets can be used to identify bipartite graphs.

**Proposition 4.1.** *For any hidden variable $H_i$, $\mathrm{ch}_X(H_i)$ is a maximal valid subset.*

*Proof.* By Assumption 1(a) and Assumption 1(b), two observed variables $X_i$ and $X_j$ are dependent if and only if $X_i$ and $X_j$ are $d$-connected. Therefore, $D(G^{(I)}) = D(P^{(I)})$ under any intervention target $I$.

Any two variables in $\mathrm{ch}_X(H_i)$ would stay $d$-connected under any unknown intervention on latent variables via the common parent $H_i$. So $\mathrm{ch}_X(H_i)$ are always in the same clique for any $D(G^{(I)})$ and thus $\mathrm{ch}_X(H_i)$ must be a valid subset.

Suppose $\mathrm{ch}_X(H_i)$ is not maximal. Then there exists a valid subset $X' \subseteq X$ such that $\mathrm{ch}_X(H_i) \subsetneq X'$, and by definition, for any intervention target $I$ and any maximal clique $C \in \Omega_G^{(I)}$ such that $\mathrm{ch}_X(H_i) \subseteq C$, we have $\mathrm{ch}_X(H_i) \subsetneq X' \subseteq C$. Let $e$ be an element in $X' \setminus \mathrm{ch}_X(H_i)$. Consider the following new graph $G'$ with added edge $H_i \to e$. We know that for any maximal clique $C$ such that $\mathrm{ch}_X(H_i) \subseteq C$, we have $\mathrm{ch}_X(H_i) \subseteq \mathrm{ch}_X(H_i) \cup \{e\} \subseteq X' \subseteq C$.

To get a contradiction, we need to show that $D(G^{(I)}) = D(G'^{(I)})$, for any intervention target $I$. Because we are adding an additional edge to $G'$, existing active paths between any observed variables

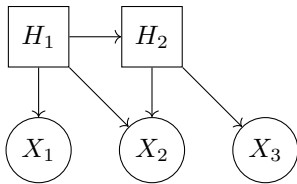

Figure 8: In this example, $\{X_2\}$ is a maximal valid subset and is contained in another maximal valid subset $\{X_1, X_2\}$

in $G$ still exist in $G'$. Therefore, $D(G'^{(I)})$ can only have more edges, and any potential additional edges in $D(G'^{(I)})$ must be between $e$ and some observed node $X_i$. There are two cases. (a) Suppose $X_i \in \mathrm{ch}_X(H_i)$, then because $e$ is always in the same clique as $\mathrm{ch}_X(H_i)$, the edge $X_i - e$ must already exist in $D(G^{(I)})$. (b) Suppose $X_i \in \mathrm{ch}_X(H_j)$ where $H_i \neq H_j$. Then $X_i$ and $e$ are $d$-connected because there is an active path between $H_i$ and $H_j$ under intervention target $I$. By Lemma D.13, there exists a maximal clique $C \in \Omega_G^{(I)}$ where $\mathrm{ch}_X(H_i) \cup \mathrm{ch}_X(H_j) \subseteq C$. However, we also know that $\mathrm{ch}_X(H_i) \subseteq \mathrm{ch}_X(H_i) \cup \{e\} \subseteq X' \subseteq C$. Therefore, the edge $X_i - e$ must already exist in $D(G^{(I)})$. Then, $\{\Omega_G^{(I)}\}_{I \in \mathcal{I}}$ is the same as $\{\Omega_{G'}^{(I)}\}_{I \in \mathcal{I}}$. And by Lemma D.10, $\{\mathcal{T}_X(G^{(I)})\}_{I \in \mathcal{I}}$ is the same as $\{\mathcal{T}_X(G'^{(I)})\}_{I \in \mathcal{I}}$. By Lemma D.17, this is not possible because $G$ is maximal. $\qquad \square$

### D.1 Replaceable and imaginary subsets

Somewhat unintuitively, it is possible that a maximal valid subset can be contained in another maximal valid subset. The reason is that $X''$ in Definition 3.2 is not allowed to depend on the clique $C$.

**Example 7.** Consider $G$ defined in Figure 8. When $I = \{H_1\}$ or $I = \emptyset$, there is one maximal clique $C = \{X_1, X_2, X_3\} \in \Omega_G^{(I)}$. When $I = \{H_2\}$, then there are two maximal cliques in $\Omega_G^{(I)}$: $\{X_1, X_2\}$ and $\{X_2, X_3\}$. Thus,

$$\Omega = \{\{X_1, X_2\}, \{X_2, X_3\}, \{X_1, X_2, X_3\}\},$$

and the valid subsets are $\{X_1\}$, $\{X_2\}$, $\{X_3\}$, $\{X_1, X_2\}$, and $\{X_2, X_3\}$. It is clear that $\{X_1, X_2\}$ is a maximal valid subset by the definition. But it turns out that $\{X_2\}$ is also maximal: Even though there are valid sets between $\{X_2\}$ and each clique in $\Omega$, there is not a *single* valid subset with this property.

**Theorem 4.3.** *If there are no non-replaceable imaginary subsets, then a subset $X'$ is a non-replaceable maximal valid subset if and only if there exists a hidden variable $H_i$ such that $\mathrm{ch}_X(H_i) = X'$. In particular, it follows that $G_B$ is identifiable.*

*Proof.* ($\Rightarrow$) Suppose there exists a non-replaceable maximal valid subset $X'$ such that there is no hidden variable $H_i$ with $\mathrm{ch}_X(H_i) = X'$. Then $X'$ is imaginary, which is a contradiction. Note that $X'$ cannot be a proper set of any $\mathrm{ch}_X(H_i)$ either, because it is non-replaceable and all $\mathrm{ch}_X(H_i)$ are maximal valid subsets by Proposition 4.1.

($\Leftarrow$) On the other hand, there exists a hidden variable $H_i$ such that $\mathrm{ch}_X(H_i) = X'$. Then it must be non-replaceable. Suppose it is replaceable, then, by definition, there exists another maximal valid subset $X''$ such that $X' \subsetneq X''$. Then by the subset condition (Lemma C.1), $X''$ must be imaginary. Suppose $X''$ is replaceable, then there must exist another non-replaceable imaginary subset that contains both $X'$ and $X''$, which is also a contradiction. $\qquad \square$

There might exist a hidden variable $H_i$ such that $\mathrm{ch}_X(H_i)$ is replaceable (Example 8).

**Example 8** (Non-imaginary set can be replaceable). Consider Figure 9 below. When the intervention target $I$ is $\emptyset$, $\{H_4\}$ or $\{H_1\}$, then there are two maximal cliques: $\{X_1, X_2, X_3, X_5\}$ and $\{X_2, X_3, X_4, X_5\}$. When the intervention target $I$ is $\{H_2\}$, then there are three maximal cliques: $\{X_1, X_5\}$, $\{X_4\}$ and $\{X_2, X_3, X_5\}$. When the intervention target is $\{H_3\}$, then there are three maximal cliques: $\{X_2, X_4\}$, $\{X_3, X_5\}$ and $\{X_1, X_2, X_5\}$. Both $\{X_2\}$ and $\{X_2, X_5\}$ are maximal valid subsets. But $\{X_2, X_5\}$ is imaginary. One might be tempted to add another edge $H_2 \rightarrow X_5$ which would create a clique of $\{X_2, X_4, X_5\}$ when intervening on $H_3$.

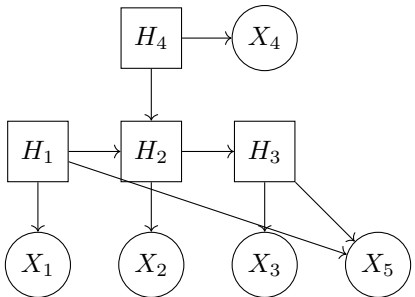

Figure 9: In this example, $\{X_2\}$ is a maximal valid subset and is contained in another maximal valid subset $\{X_2, X_5\}$ even though $\{X_2\}$ is the child set of $H_2$.

## D.2 Single source node

It can be shown that imaginary subsets do not exist if there is only one single hidden source node, i.e. a hidden variable such that $\mathrm{pa}(H_i) = \emptyset$. Note that this still allows for arbitrarily many hidden variables $|H| > 1$.

For any two observed variables $X_a$ and $X_b$. Let $P^{(a,b)}$ be the set of all active paths between $X_a$ and $X_b$ in $G^{(\emptyset)}$.

**Definition D.1** (Dominated path)**.** An active path $P$ between $X_a$ and $X_b$ is called a $(X_a, X_b)$-dominated path if either of the followings is true:

(1) $P$ is a common parent path (i.e., $P$ is $X_a \leftarrow H_k \rightarrow X_b$ for some hidden variable $H_k$) and every directed path to $\mathrm{so}(P)$ has at least one node $H_i \neq H_k$ such that such that $\{X_a, X_b\} \subseteq \mathrm{ch}_X(H_i)$.

(2) $P$ is not a common parent path and (a) every directed path to $\mathrm{so}(P)$ has at least one node $H_i$ such that $\{X_a, X_b\} \subseteq \mathrm{ch}_X(H_i)$ or (b) there exists a hidden node $H_j$ in $P$ such that $X_a \leftarrow H_j \rightarrow X_b$ is a non-$(X_a, X_b)$-dominated path (see (1)).

*Remark* D.1. We consider the empty path from $H_i$ to $H_i$ as a directed path.

*Remark* D.2. To slightly abuse notation, when $\mathrm{so}(P)$ only has one element, we also use $\mathrm{so}(P)$ to mean that element.

Appendix D.2.1 gives some lemmas to characterize dominated paths.

**Theorem 4.4.** *Under Assumptions 1-2, if $G_H$ has one latent source, there are no imaginary subsets.*

*Proof.* Suppose there is an imaginary subset $X'$. Then it must have at least three elements in it. If $|X'| = 1$, then it must be a subset of $\mathrm{ch}_X(H_i)$ for some $H_i$. If $|X'| = 2$, then these two nodes do not share the same parent. Thus, there must exist at least two active paths between them in $G^{(\emptyset)}$ because of Lemma E.4. By definition, these two active paths are non-dominated. But because there is only one single source node in the latent DAG, by Proposition D.5, this is not possible.

For every pair of nodes in $X'$, there could only be one non-dominated path between them because the necessary conditions in Proposition D.5 cannot be satisfied when there is only one single source node and Lemma D.1 shows that there is at least one non-dominated path.

By Proposition D.6, $X'$ cannot be imaginary. □

### D.2.1 Key lemmas and propositions for proving Theorem 4.4

The following two lemmas characterize dominated paths.

**Lemma D.1.** *If $X_a$ and $X_b$ are $d$-connected in $G^{(\emptyset)}$, then $P^{(a,b)}$ has at least one non-$(X_a, X_b)$-dominated path.*

*Proof.* Because $X_a$ and $X_b$ are $d$-connected in $G^{(\emptyset)}$, by definition, $P^{(a,b)}$ is not empty.

Now let's consider two cases:

(a) There exists at least one hidden node that is a common parent of $X_a$ and $X_b$. Without loss of generality, let $H_1$ be a common parent of $X_a$ and $X_b$. Because there is no circle and there is only a finite number of hidden variables, there exists a hidden variable $H_2$ that could be $H_1$ or an ancestor of $H_1$ such that $\{X_a, X_b\} \subseteq \text{ch}_X(H_2)$ and $H_2$ does not have an ancestor such that both $X_a$ and $X_b$ are its children. By definition, $X_a \leftarrow H_2 \rightarrow X_b$ is non-$(X_a, X_b)$-dominated.

(b) There is no hidden node that is a common parent of $X_a$ and $X_b$. Then every path in $P^{(a,b)}$ is non-$(X_a, X_b)$-dominated. Because $P^{(a,b)}$ is not empty, $P^{(a,b)}$ has at least one non-$(X_a, X_b)$-dominated path.

$\square$

**Lemma D.2.** *If $\{X_a, X_b\}$ is a valid subset and $P^{a,b}$ has only one non-$(X_a, X_b)$-dominated path, then the non-$(X_a, X_b)$-dominated path must be a common parent path.*

*Proof.* Let's consider two cases:

(a) There is no hidden node that is a common parent of $X_a$ and $X_b$. Because $\{X_a, X_b\}$ is a valid subset, there must exist at least two active paths between $X_a$ and $X_b$ in $G^\emptyset$. Otherwise, by Lemma E.4, $X_a$ and $X_b$ must be $d$-separated for an intervention target. By definition, any active paths between $X_a$ and $X_b$ are non-$(X_a, X_b)$-dominated paths. So, this case is not possible.

(b) There exists at least one hidden node that is a common parent of $X_a$ and $X_b$. Without loss of generality, let $H_1$ be a common parent of $X_a$ and $X_b$. If the active path $X_a \leftarrow H_1 \rightarrow X_b$ is $(X_a, X_b)$-dominated, we can find an ancestor of $H_1$ called $H_2$ such that $\{X_a, X_b\} \subseteq \text{ch}_X(H_2)$. Because there is only a finite number of hidden variables and there is no circle, we can eventually find a common parent $H'$ of $X_a$ and $X_b$ such that $X_a \leftarrow H' \rightarrow X_b$ is a non-$(X_a, X_b)$-dominated path. Therefore at least one of the non-$(X_a, X_b)$-dominated paths is a common parent path. And if there is only one non-$(X_a, X_b)$-dominated path, it must be a common parent path.

$\square$

**Lemma D.3.** *For any two nodes $X_a$ and $X_b$ where $P^{a,b}$ has only one non-$(X_a, X_b)$-dominated path $P_*$, then there does not exist a directed path from $H_1$ to $H_2$ with either $H_1 \rightarrow X_a, H_2 \rightarrow X_b$ or $H_2 \rightarrow X_a, H_1 \rightarrow X_b$ such that $\text{so}(P) \in \text{de}(H_2)$.*

*Proof.* By Lemma D.2, $P_*$ is a common parent path. Let $H_*$ be the common parent. Now here are two cases:

A. $H_1 = H_2$. In this case, $P_*$ would not be a non-dominated path by definition. So this is not possible.

B. $H_1 \neq H_2$. Because this path is dominated, then either (a) $H_*$ must be on this path or (b) there exists a common parent of $X_a, X_b$: $H'$ such that $H' \in \overline{\text{an}}(H_1)$. Case (a) is not possible because $H_* \in \text{de}(H_2)$. Case (b) is also not possible by Lemma D.4.

$\square$

**Lemma D.4.** *For any two nodes $X_a$ and $X_b$ where $P^{a,b}$ has only one non-$(X_a, X_b)$-dominated path $P_*$, for any hidden node $H'$ that is a common parent of $X_a, X_b$ and not the same as $\text{so}(P_*)$, $\text{so}(P_*)$ is an ancestor of $H'$.*

*Proof.* Suppose $\text{so}(P_*)$ is not an ancestor of $H'$. Then because $X_a \leftarrow H' \leftarrow X_b$ is a $(X_a, X_b)$-dominated path, and there is only a finite number of hidden variables, we can eventually find a common parent $H''$ of $X_a$ and $X_b$ such that $X_a \leftarrow H'' \rightarrow X_b$ is a non-$(X_a, X_b)$-dominated path. But $\text{so}(P_*)$ is not an ancestor of $H'$. So it cannot be $H''$. This is a contradiction to the fact that $P^{a,b}$ has only one non-$(X_a, X_b)$-dominated path. $\square$

**Proposition D.5.** *If $\{X_a, X_b\} \subseteq X'$ where $X'$ is an imaginary subset and $P^{(a,b)}$ has at least two non-$(X_a, X_b)$-dominated paths, then the following two conditions hold:*

(a) *There exists a pair of non-$(X_a, X_b)$-dominated paths $(P_1, P_2)$ such that given any intervention target $I \in \mathcal{I}$, at least one of $P_1$ and $P_2$ is active in $G^{(I)}$*

(b) *For any pair of non-$(X_a, X_b)$-dominated paths $(P_1, P_2)$ that satisfies (a), there does not exist a single hidden node $H_\#$ that is shared by all directed path to $\mathrm{so}(P_1)$ and $\mathrm{so}(P_2)$ in $G^{(\emptyset)}$.*

*Proof.* First, let's prove the first condition. We consider two conditions:

(a) There is no hidden node that is a common parent of $X_a$ and $X_b$. Because $\{X_a, X_b\}$ is a valid subset, there must exist at least two active paths between $X_a$ and $X_b$ in $G^{(\emptyset)}$. Otherwise, by Lemma E.4, $X_a$ and $X_b$ must be $d$-separated for an intervention target. By definition, any active paths between $X_a$ and $X_b$ are non-$(X_a, X_b)$-dominated paths.

(b) There exists at least one hidden node that is a common parent of $X_a$ and $X_b$. Without loss of generality, let $H_1$ be a common parent of $X_a$ and $X_b$. If the active path $X_a \leftarrow H_1 \rightarrow X_b$ is $(X_a, X_b)$-dominated, we can find an ancestor of $H_1$ called $H_2$ such that $\{X_a, X_b\} \subseteq \mathrm{ch}_X(H_2)$. Because there is only a finite number of hidden variables and there is no circle, we can eventually find a common parent $H'$ of $X_a$ and $X_b$ such that $X_a \leftarrow H' \rightarrow X_b$ is a non-$(X_a, X_b)$-dominated path. Therefore at least one of the non-$(X_a, X_b)$-dominated paths is a common parent path. Because $P^{(a,b)}$ has at least two non-$(X_a, X_b)$-dominated paths, and given any intervention target, the common parent path will not be destroyed, the first condition is true.

Let's denote $P_1$ to be a non-$(X_a, X_b)$-dominated active path with hidden variables $H_{a,1}$ and $H_{b,1}$ such that $X_a \leftarrow H_{a,1}$ and $X_b \leftarrow H_{b,1}$ are in $P_1$. Let $P_2$ to be an a non-$(X_a, X_b)$-dominated active path with hidden variables $H_{a,2}$ and $H_{b,2}$ such that $X_a \leftarrow H_{a,2}$ and $X_b \leftarrow H_{b,2}$ are in $P_2$.

Now let's prove the second condition. Suppose, on the contrary, there exists such a hidden node $H_\#$. Because $P_1$ and $P_2$ are non-$(X_a, X_b)$-dominated, there must exist one directed path to $P_1$ and one directed path to $P_2$ such that these two directed paths do not have any hidden node $H_i$ with $\{X_a, X_b\} \subseteq \mathrm{ch}_X(H_i)$, except for maybe $\mathrm{so}(P_1)$ and $\mathrm{so}(P_2)$. On the other hand, these two directed paths share at least one common node $H_\#$. Let $H_*$ be the common nodes with the lowest ranking in topological ordering. Then $H_*$ has the property that, under any intervention target $I$, one of the two directed paths $P_{1,*}$ and $P_{2,*}$, where $P_{1,*}$ is the directed path from $H_*$ to $\mathrm{so}(P_1)$ and $P_{2,*}$ is the directed path from $H_*$ to $\mathrm{so}(P_2)$, still exists. If this is not true, then there is one intervention target $I_*$ such that intervening on this node would break $P_{1,*}$ and $P_{2,*}$. Then $I_*$ would be a common node on $P_{1,*}$ and $P_{2,*}$ other than $H_*$. This violates the assumption $H_*$ has the lowest ranking in the topological ordering. Note that $\mathrm{so}(P_1)$ or $\mathrm{so}(P_2)$ only has one element in it by Lemma E.3.

We now have four paths $P_1$, $P_2$ $P_{1,*}$ and $P_{2,*}$. Under any intervention target $I$, one of $P_1$, $P_2$ is active and one of $P_{1,*}$ and $P_{2,*}$ is still active. Let's consider the following three cases:

A. $P_{1,*}$ has no shared hidden node with $P_2$ and $P_{2,*}$ has no shared hidden node with $P_1$.

First of all, $H_*$ cannot be a common parent of $X_a$ and $X_b$. Suppose $H_*$ is a common parent of $X_a$ and $X_b$. Then by our construction, it is only possible if $H_* = \mathrm{so}(P_1)$ and $H_* = \mathrm{so}(P_2)$. Because $P_1$ and $P_2$ are both non-dominated paths, they must both be common parent paths in this case. This would imply they are the same path, which is a contradiction.

Note that by Lemma E.3, an active path between hidden nodes $H_1$ and $H_2$ can only be a common ancestor path or a directed path. Either way, there is only one source node of the path and every other node in the path is a descendant of that source node. Therefore, $P_{1,*}$ cannot have a shared hidden node with $P_1$ except for $\mathrm{so}(P_1)$, otherwise there will be a directed circle. Similarly, $P_{2,*}$ cannot have a shared hidden node with $P_2$ except for $\mathrm{so}(P_2)$. For any intervention target $I$, there are three cases:

(a) None of $P_1$, $P_2$ $P_{1,*}$ and $P_{2,*}$ are affected by $I$. Then all of $H_{a,1}, H_{a,2}, H_{b,1}, H_{b,2}$ are descendents of $H_*$.

(b) One of $P_{1,*}$ and $P_{2,*}$ is destroyed. Without loss of generality, suppose $P_{2,*}$ no longer exists. The intervention target $I$ must be a part of the path $P_{2,*}$ and it cannot be $H_*$. Because $P_{2,*}$ has no shared hidden node with $P_1$, intervening on $I$ means that $P_1$ is still intact. Therefore, $H_{a,1}, H_{b,1}$ are descendents of $H_*$.

(c) One of $P_1$ and $P_2$ is destroyed. Without loss of generality, suppose $P_2$ no longer exists. The intervention target $I$ must be a part of the path $P_2$ and it cannot be $\mathrm{so}(P_2)$. Therefore, $P_{1,*}$ is still intact. $H_{a,1}, H_{b,1}$ are descendents of $H_*$.

Finally, in all three cases, at least one pair of $(H_{a,1}, H_{b,1})$ and $(H_{a,2}, H_{b,2})$ is a set of descendents of $H_*$. Now consider graph $G' = (V, E')$ with $E' = E \cup \{H_* \to X_a\} \cup \{H_* \to X_b\}$. Note that $E'$ is not the same as $E$ because at least one of $\{H_* \to X_a\}$ and $\{H_* \to X_b\}$ is missing. Otherwise, $H_*$ would be a common parent of $X_a$ and $X_b$. By Lemma D.15, $G'$ violates the maximality.

B. Without loss of generality, $P_{1,*}$ has shared hidden nodes with $P_2$ and $P_1$ is not a common parent path.

Note that in this case, $P_{2,*}$ cannot have a shared hidden node with $P_1$ except for one special case. Suppose the opposite is true. Let the shared node between $P_{1,*}$ and $P_2$ be $H_1$. Let the shared node between $P_{2,*}$ and $P_1$ be $H_2$. There must be a circle between $H_1$ and $H_2$ if $H_1 \neq H_2$ which is a contradiction. If $H_1 = H_2$, then $H_1 = H_2 = \mathrm{so}(P_1) = \mathrm{so}(P_2)$. In this case, there is always at least one pair of $(H_{a,1}, H_{b,1})$ and $(H_{a,2}, H_{b,2})$ that is a set of descendents of $\mathrm{so}(P_1)$ and $\mathrm{so}(P_2)$ under any intervention target. In this case, adding $\{X_a, X_b\}$ to $\mathrm{ch}_X(\mathrm{so}(P_1))$ would not change $\Omega_G^{(I)}$ by the proof of Lemma D.15. By Lemma D.17, this is not possible. So we will ignore this special case.

Now let's consider the following cases under any intervention target $I$ for when $P_{2,*}$ does not have a shared hidden node with $P_1$:

(a) None of $P_1$, $P_2$ $P_{1,*}$ and $P_{2,*}$ are affected by $I$. Then $H_{a,1}, H_{b,1}$ are descendents of $\mathrm{so}(P_1)$.

(b) $P_{1,*}$ is destroyed. Then $P_1$ must be intact because the intervention target is in $P_{1,*}$. $H_{a,1}, H_{b,1}$ are descendents of $\mathrm{so}(P_1)$.

(c) $P_{2,*}$ is destroyed. Because $P_{2,*}$ does not have a shared hidden node with $P_1$, $H_{a,1}, H_{b,1}$ are descendents of $\mathrm{so}(P_1)$.

(d) $P_2$ is destroyed. Then $P_1$ must be intact. Then $H_{a,1}, H_{b,1}$ are descendents of $\mathrm{so}(P_1)$.

(e) $P_1$ is destroyed. Then $P_2$ must be intact. $P_{1,*}$ must also be intact because the intervention target is in $P_1$ and it is not $\mathrm{so}(P_1)$. Because $P_{2,*}$ cannot have a shared hidden node with $P_1$, $P_{2,*}$ is also active. Therefore, $H_{a,2}, H_{b,2}$ are connected to $\mathrm{so}(P_1)$ via the common ancestor $H_*$.

Let's consider graph $G' = (V, E')$ with $E' = E \cup \{\mathrm{so}(P_1) \to X_a\} \cup \{\mathrm{so}(P_1) \to X_b\}$. Note that $E'$ is not the same as $E$ because at least one of $\{\mathrm{so}(P_1) \to X_a\}$ and $\{\mathrm{so}(P_1) \to X_b\}$ is missing because $P_1$ is not a common parent path and $P_1$ is also non-$(X_a, X_b)$-dominated.

For any intervention target $I$ that induces $(a), (b), (c), (d)$, adding $\{X_a, X_b\}$ to $\mathrm{ch}_X(\mathrm{so}(P_1))$ would not change $\Omega_G^{(I)}$ by the proof of Lemma D.15.

For any intervention target $I$ that induces $(e)$, $D(G'^I)$ does not have more edges than $D(G^I)$. If the opposite is true, then the new edge in $D(G'^I)$ must involve the added node $X_a$ or $X_b$. Without loss of generality, suppose the new edge is between $X_a$ and $X_k$, it must be that a parent of $X_k$ is $d$-connected to $\mathrm{so}(P_1)$. Suppose that the parent of $X_k$ is $H_k$. By Lemma E.3, there are 2 cases. If $\mathrm{so}(P_1)$ has a directed path to $H_k$, then $H_k$ is connected to $X_a$ and $X_b$ via the common ancestor $H_*$. If $H_k$ has a directed path to $\mathrm{so}(P_1)$ or $H_k$ and $\mathrm{so}(P_1)$ share a common ancestor, by our assumption, $H_k$ and $\{X_a, X_b\}$ are connected by common ancestor $H_\#$. Therefore, $X_k$ and $X_a$ are $d$-connected in $D(G^I)$.

So adding $\{X_a, X_b\}$ to $\mathrm{ch}_X(\mathrm{so}(P_1))$ would not change $\Omega_G^{(I)}$.

Finally, by Lemma D.17, this is not possible.

C. Without loss of generality, $P_{1,*}$ has shared hidden nodes with $P_2$ and $P_1$ is a common parent path.

Let $H_!$ be the shared node between $P_{1,*}$ and $P_2$ that has the lowest topological ranking. Therefore, for any intervention target $I$, at least one of $P_2$ and the directed path from $H_!$ to $\mathrm{so}(P_1)$ is intact. Otherwise, $H_!$ is not the shared node between $P_{1,*}$ and $P_2$ that has the lowest topological ranking.

First, let's consider two scenarios.

(1) Both $P_1$ and $P_2$ are common parent paths. First of all, $\mathrm{so}(P_1)$ is not the same as $\mathrm{so}(P_2)$. Otherwise, $P_1$ and $P_2$ would be the same path. On the other hand, if $\mathrm{so}(P_2)$ appears in $P_{1,*}$ because there is the only possible node that can be shared and is not $\mathrm{so}(P_1)$, it would violate how $P_{1,*}$ is chosen. So this scenario is not possible.

(2) $P_1$ is a common parent path and $P_2$ is not. First, $H_!$ is not the same as $\mathrm{so}(P_1)$ because $H_!$ is also on $P_2$ which is a non-$(X_a, X_b)$-dominated path. By definition, this is not possible. On the other hand, $H_!$ is not a common parent of $X_a$ and $X_b$ because it would violate how $P_{1,*}$ is chosen. We'll study this scenario next.

Similar to the previous case, $P_{2,*}$ cannot have a shared hidden node with $P_1$. Now let's consider the following cases under any intervention target $I$:

(a) None of $P_1$, $P_2$ $P_{1,*}$ and $P_{2,*}$ are affected by $I$. Then $H_{a,1}, H_{b,1}$ are descendents of $H_!$.

(b) $P_{2,*}$ is destroyed. Because $P_{2,*}$ does not have a shared hidden node with $P_1$, $H_{a,1}, H_{b,1}$ are descendents of $H_!$.

(c) $P_2$ is destroyed. Then the directed path from $H_!$ to $\mathrm{so}(P_1)$ is still intact. Then $H_{a,1}, H_{b,1}$ are descendents of $H_!$.

(d) $P_{1,*}$ is destroyed but the directed path from $H_!$ to $\mathrm{so}(P_1)$ is still intact. Then $P_1$ must be intact because the intervention target is in $P_{1,*}$. $H_{a,1}, H_{b,1}$ are descendents of $H_!$.

(e) $P_1$ is destroyed or the the directed path from $H_!$ to $\mathrm{so}(P_1)$ is destroyed. In either case, $P_2$ must be intact. The directed path from $H_*$ to $H_!$ must also be intact. Because $P_{2,*}$ cannot have a shared hidden node with $P_1$ or $P_{1,*}$, $P_{2,*}$ is also active. Therefore, $H_{a,2}, H_{b,2}$ are connected to $H_!$ via the common ancestor $H_*$.

Now consider graph $G' = (V, E')$ with $E' = E \cup \{H_! \to X_a\} \cup \{H_! \to X_b\}$. Note that $E'$ is not the same as $E$ because at least one of $\{H_! \to X_a\}$ and $\{H_! \to X_b\}$ is missing. By the same argument as fore, $G'$ also violates the maximality assumption by Lemma D.17.

Finally, we have a contradiction. So the second condition is also necessary. □

**Proposition D.6.** *Suppose $X' \subseteq X$ is a valid subset with $|X'| \geq 2$ and every pair of nodes in $X'$ has only one non-dominated path. And the latent DAG only has one source node. Then the followings are true:*

*(1) there exists a hidden node $H'$ with $X' \subseteq \mathrm{ch}_X(H')$*

*(2) for any node $X_i \in X'$, one can find at least another node $X_j \in X'$ such that $X_i \leftarrow H' \to X_j$ is a non-$(X_i, X_j)$-dominated path.*

*Proof.* We'll show this by induction.

**Base step:** This is true because of Lemma D.2.

**Inductive Hypothesis:** Suppose the statement is true for all $X'$ with $|X'| = k$.

**Inductive Step:** We will show the statement is true for $|X'| = k + 1$. Without loss of generality, let's suppose $X' = \{X_1, ..., X_{k+1}\}$. Consider the subset $X'' = \{X_1, ..., X_k\}$. By the inductive hypothesis, there exists a hidden node $H'$ with $X' \subseteq \mathrm{ch}_X(H')$ and again without loss of generality, two nodes $X_1$ and $X_2$, such that $X_1 \leftarrow H' \to X_2$ is a non-$(X_1, X_2)$-dominated path.

Now consider another subset $X_{1,k+1} = \{X_1, X_{k+1}\}$. By our assumption and Lemma D.2, there exists a hidden node $H_{1,k+1}$ such that $X_1 \leftarrow H_{1,k+1} \rightarrow X_{k+1}$ is a non-$(X_1, X_{k+1})$-dominated path. We can define $H_{2,k+1}$ in a similar way.

If $H_{1,k+1} = H'$, then we are done.

Let's now consider the cases when $H_{1,k+1} \neq H'$. Because there is only one single source node, there must exist at least one common ancestor of $H_{1,k+1}$ and $H'$: $H_*$. Now let's consider three cases:

(1) $H_{1,k+1}$ is an ancestor of $H'$. Then there is a directed path $P$ from $H_{1,k+1}$ to $H'$. Now we have an active path between $X_2$ and $X_{k+1}$ via $P$. Because this path is $(X_2, X_{k+1})$-dominated, we have a few cases.

    (a) There exists a common parent of $X_2, X_{k+1}$ that is in $\overline{\mathrm{an}}(H_{1,k+1})$, which means that $H_{2,k+1}$ is an ancestor of $H_{1,k+1}$ by Lemma D.4 (case (c) covers when $H_{2,k+1} = H_{1,k+1}$). By Lemma D.3, this is not possible because we have a path between $X_1$ and $X_2$ via the directed path from $H_{2,k+1}$ to $H_{1,k+1}$ but $H'$ is a descendent of $H_{1,k+1}$.

    (b) $H_{2,k+1}$ is on $P$ but it is not the same as $H_{1,k+1}$ nor $H'$. By the same argument as the case (a), this is also not possible because of Lemma D.3.

    (c) $H_{2,k+1} = H_{1,k+1}$. This means that $H_{1,k+1}$ is a common parent of $X_1, X_2$. So it cannot be an ancestor of $H'$. This is a contradiction.

    (d) When $H_{2,k+1} = H'$, then $H'$ is the hidden node we are looking for because $X' \subseteq \mathrm{ch}_X(H')$ and for any node $X_i \in X'$, there exists another node $X_j \in X'$ such that $X_i \leftarrow H' \rightarrow X_j$ is a non-$(X_i, X_j)$-dominated path (for $X_{k+1}$, one can choose $X_2$ such that $X_2 \leftarrow H' \rightarrow X_{k+1}$ is a non-dominated path).

(2) $H'$ is an ancestor of $H_{1,k+1}$. Then there is a directed path $P$ from $H'$ to $H_{1,k+1}$. Let $X_q$ be an arbitrary node in $X''$ that is not $X_1$. Then we can define hidden node $H_{q,k+1}$ such that $X_q \leftarrow H_{q,k+1} \rightarrow X_{k+1}$ is a non-$(X_q, X_{k+1})$-dominated path. Now we have an active path between $X_q$ and $X_{k+1}$ via $P$. Because this path is $(X_q, X_{k+1})$-dominated, we have a few cases.

    (a) There exists a common parent of $X_q, X_{k+1}$ that is in $\overline{\mathrm{an}}(H')$, which means that $H_{q,k+1}$ is an ancestor of $H'$ by Lemma D.4 (case (c) covers when $H_{q,k+1} = H'$). By Lemma D.3. This is not possible because we have a path between $X_1$ and $X_{k+1}$ via the directed path from $H_{q,k+1}$ to $H'$ but $H_{1,k+1}$ is a descendent of $H'$.

    (b) $H_{q,k+1}$ is on $P$ but it is not the same as $H_{1,k+1}$ nor $H'$. By the same argument as the case (a), this is also not possible because of Lemma D.3.

    (c) $H_{q,k+1} = H'$. This means $H'$ is a common parent of $X_1, X_{k+1}$. So it cannot be an ancestor of $H_{1,k+1}$. This is a contradiction.

    (d) $H_{q,k+1} = H_{1,k+1}$. This is possible. But because $X_q$ is chosen arbitrarily. We have $H_{q,k+1} = H_{1,k+1}$ for $q = 2, ..., k$. Therefore $X' \subseteq \mathrm{ch}_X(H_{1,k+1})$. And for any node $X_i \neq X_{k+1}$, we have that $X_i \leftarrow H_{1,k+1} \rightarrow X_{k+1}$ is a non-dominated path.

(3) There exist a common ancestor $H_*$ that is neither $H_{1,k+1}$ nor $H'$, $H'$ is not an ancestor of $H_{1,k+1}$ and $H_{1,k+1}$ is not an ancestor of $H'$. $H_*$ is a common ancester of $X_2$ and $X_{k+1}$. Therefore, there is a common ancestor path between $X_2$ and $X_{k+1}$. But this path is a $(X_2, X_{k+1})$-dominated path.

There are two cases: (a) $H_{2,k+1} \in \overline{\mathrm{an}}(H_{1,k+1})$ when $H_{2,k+1}$ is on the directed path from $H_*$ to $H_{1,k+1}$ or there is another common parent of $X_2, X_{k+1}$ that is in $\overline{\mathrm{an}}(H_*)$ (Lemma D.4), (b) $H_{2,k+1} \in \overline{\mathrm{an}}(H')$ when $H_{2,k+1}$ is on the directed path from $H_*$ to $H'$ or there is another common parent of $X_2, X_{k+1}$ that is in $\overline{\mathrm{an}}(H_*)$ (Lemma D.4). For case (a), we have a $(X_1, X_2)$-dominated path cased by the path from $H_{2,k+1}$ to $H_{1,k+1}$. In this case, because $X_1 \leftarrow H' \rightarrow X_2$ is a non-$(X_1, X_2)$-dominated path. $H'$ must be an ancestor for $H_{1,k+1}$, which is a contradiction. By a similar argument, case (b) is also not possible. $\square$

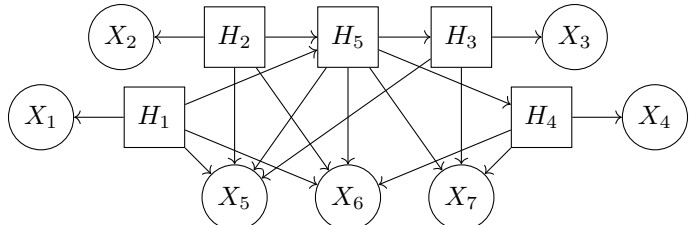

Figure 10: In this example, the non-imaginary subset $\{X_5, X_6, X_7\}$ is fractured

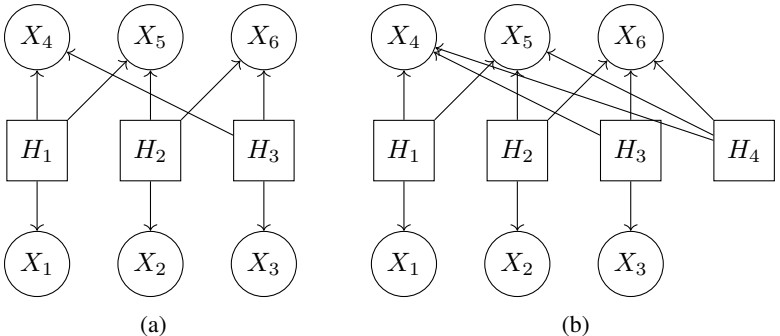

Figure 11: An example where two maximal measurement models share the same UDG

### D.3 Fractured subset condition

A necessary condition for a non-replaceable imaginary subset is that the subset is fractured. The definition is restated below. Intuitively, a fractured subset provides redundant information as every connection between nodes in the fractured subset can be explained by other maximal valid subsets.

**Definition 4.5** (Fractured subset). Given a collection of maximal valid subsets $\mathcal{S}$, a clique $C$ is called **shattered** by $\mathcal{S}$ if there exists a subset $\mathcal{S}' \subseteq \mathcal{S}$ such that $\cup \mathcal{S}' = C$. A collection of maximal valid subsets $\{S_i\}_{i=1}^k$ is **complete** if for every intervention target $I \in \mathcal{I}$, all shattered cliques in $D(P^{(I)})$ form an edge cover of $D(P^I)$. A maximal valid subset $X' \subseteq X$ is **fractured** if there exists a complete collection $\{S_i\}_{i=1}^k$ such that $S_i \nsubseteq X'$ for all $S_i$.

The aforementioned necessity is shown by the following:

**Lemma 4.6.** *If a subset $X' \subseteq X$ is imaginary, then $X'$ is also fractured.*

*Proof.* By Lemma D.16, we know that $\{\mathrm{ch}_X(H_i)\}_{i=1}^m$ is a complete cover. And because $X'$ is an imaginary subset, $X' \nsubseteq \mathrm{ch}_X(H_i)$ for any $H_i$. $\qquad\square$

One might suggest getting rid of all fractured subsets. However, even a non-imaginary subset can be a fractured subset (Example 9).

**Example 9** (Non-imaginary set can be fractured). Consider Figure 10. If the intervention target $I$ is $\{H_1\}$, $\{H_2\}$, $\emptyset$, then we have two maximal cliques: $\{X_1, X_3, X_4, X_5, X_6, X_7\}$, $\{X_2, X_3, X_4, X_5, X_6, X_7\}$. If the intervention target $I$ is $\{H_3\}$, then there are three maximal cliques: $\{X_3, X_5, X_7\}$, $\{X_2, X_4, X_5, X_6, X_7\}$, $\{X_1, X_4, X_5, X_6, X_7\}$. If the intervention target $I$ is $\{H_4\}$, then there are three maximal cliques: $\{X_4, X_6, X_7\}$, $\{X_2, X_3, X_5, X_6, X_7\}$, $\{X_1, X_3, X_5, X_6, X_7\}$. When the intervention target is $\{H_5\}$, there are three maximal cliques: $\{X_1, X_5, X_6\}$, $\{X_2, X_5, X_6\}$, $\{X_3, X_4, X_5, X_6, X_7\}$. By Proposition 4.1, all these five valid subsets are maximal: $\{X_1, X_5, X_6\}$, $\{X_2, X_5, X_6\}$, $\{X_3, X_5, X_7\}$, $\{X_4, X_6, X_7\}$ and $\{X_5, X_6, X_7\}$. But $\{X_5, X_6, X_7\}$ is fractured. In fact, $\{X_1, X_5, X_6\}$, $\{X_2, X_5, X_6\}$, $\{X_3, X_5, X_7\}$, $\{X_4, X_6, X_7\}$ constitutes a complete collection of maximal valid subsets.

Even under the maximality condition, it is still possible that two DAGs can share the same UDG (Example 10). The no fractured subset condition captures such ambiguities but is overkill.

**Example 10.** Consider Figure 11. For both $G_{(a)}$ and $G_{(b)}$, under any intervention target, there are four maximal cliques: $\{X_1, X_4, X_5\}$, $\{X_2, X_5, X_6\}$, $\{X_3, X_4, X_6\}$, $\{X_4, X_5, X_6\}$. And one can check that both $G_{(a)}$ and $G_{(b)}$ satisfy the maximality condition.

### D.4 Pure child assumption

We can also use the pure child assumption (Assumption 3) to get identifiability. The pure child assumption has been made in many existing works [6, 22, 66] with additional parametric assumptions to get identifiability.

**Assumption 3** (Pure child). For every $H_i \in H$, there exists at least one $X_i \in X$ with $\mathrm{pa}(X_i) = \{H_i\}$, i.e. $X_i$ only has one parent and that parent is $H_i$.

Note that under Assumption 3, there still could be imaginary subsets.

**Example 11.** Figure 12 satisfies Assumption 3 but there is an imaginary subset $\{X_5, X_6\}$.

**Theorem 4.8.** *Under Assumptions 1-3, the complete collection (cf. Definition 4.5) with the smallest cardinality is exactly $\{\mathrm{ch}_X(H_i)\}_{i=1}^m$ and thus $G_B$ can be identified.*

*Proof.* By Assumption 1(a) and Assumption 1(b), two observed variables $X_i$ and $X_j$ are dependent if and only if $X_i$ and $X_j$ are $d$-connected. Therefore, $D(G^{(I)})$ is the same as $D(P^{(I)})$ under any intervention target $I$.

By Assumption 3, let's associate each hidden node $H_i$ with one pure child $X_i$.

First of all, there does not exist an imaginary subset with two pure children $X_i$ and $X_j$ of different hidden parents. If the opposite is true, then $\{X_i, X_j\}$ must be valid which is impossible by Lemma D.14. Therefore, every complete collection must have at least $m$ maximal subsets.

By Lemma D.16, $\{\mathrm{ch}_X(H_i)\}_{i=1}^m$ is a complete collection with $m$ maximal subsets.

In fact, $\{\mathrm{ch}_X(H_i)\}_{i=1}^m$ is the only complete collection with $m$ maximal subsets. Suppose there exists a complete collection with imaginary subsets. There are two cases:

A. None of the imaginary subsets contains a pure child. Then we still need at least $m$ non-imaginary subsets to cover pure children.

B. At least one of the imaginary subsets contains a pure child. Suppose one such imaginary subset is $S_i$ and it contains $X_i$. Then there must exist another maximal subset from the collection that contains $X_i$.

Suppose the opposite is true. Because $S_i$ is an imaginary subset, there exists $X_k \in S_i$ such that $X_k$ and $X_i$ do not share the same parent. And for any intervention target $I$, if there is any edge between $X_i$ and $X_j$ in $D(G^{(I)})$, by definition, there exists a shattered clique $C \in D(G^{(I)})$ such that $\{X_i, X_j\} \subseteq C$ and $S_i \subseteq C$. Therefore, there is an edge between $X_j$ and $X_k$ as well. Now consider graph $G' = (V, E')$ where $E' = E \cup \{H_i \to X_k\}$. Under any intervention target $I$, $D(G'^{(I)})$ can only have more edges than $D(G^{(I)})$ and if there exists a new edge, these edges must involve $X_k$. Suppose one of these newly added edges is between $X_k$ and $X_j$. Then it must be that a parent of $X_j$ is $d$-connected to $H_i$. This would mean that $X_i$ is $d$-connected to $X_j$ in $D(G^{(I)})$. By the previous argument, $X_k$ is also $d$-connected to $X_j$ in $D(G^{(I)})$. Thus, $G'$ would violate the maximality condition by Lemma D.10 and Lemma D.17 which is a contradiction. Therefore, if there is one

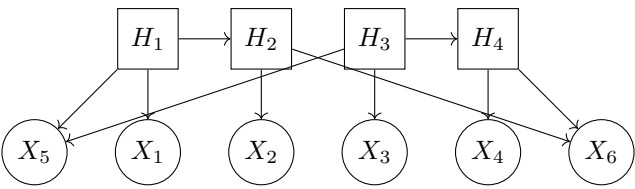

Figure 12: In this example, the graph satisfies pure child but $\{X_5, X_6\}$ is an imaginary subset.

imaginary subset in the complete collection that contains a pure child, there must be at least two imaginary subsets that contain such a pure child. This means that to cover all $m$ pure children, we need more than $m$ maximal subsets.

On the other hand, suppose there exists a complete collection with at least one maximal subset $X'$ such that $X' \subseteq \mathrm{ch}_X(H_i)$. Then $X'$ must not include the pure child of $H_i$: $X_i$. This is because, if $X_i \in X'$, then for any maximal clique $C \in \Omega$ such that $X' \subseteq C$, we have that $\{X_i\} \subseteq X' \subseteq \mathrm{ch}_X(H_i) \subseteq C$ because any node in $C \setminus \{X_i\}$ is $d$-connected by $X_i$ via $H_i$ since $X_i$ is a pure child. Therefore, $X'$ is not maximal which is a contradiction. Because $X'$ does not contain a pure child of $H_i$, we still need to cover all $m$ pure children. We need more than $m$ maximal subsets. $\qquad\square$

For algorithmic purposes, under the pure child assumption, one only needs to check if shattered maximal cliques form an edge cover.

**Lemma D.7.** *Under Assumptions 1-3, for any intervention target I, all maximal cliques $C \in \Omega_P^{(I)}$ that is shattered by $\{\mathrm{ch}_X(H_i)\}_{i=1}^m$ is an edge cover of $D(P^{(I)})$.*

*Proof.* By Assumption 1(a) and Assumption 1(b), two observed variables $X_i$ and $X_j$ are dependent if and only if $X_i$ and $X_j$ are $d$-connected. Therefore, $D(G^{(I)})$ is the same as $D(P^{(I)})$ under any intervention target $I$. By Assumption 3, let's associate each hidden node $H_i$ with one pure child $X_i$.

Suppose the statement is not true. Then for some intervention target $I \in \mathcal{I}$, there exists one clique $C$ in $\Omega_G^{(I)}$ that is not shattered and that clique contains one edge that is not covered by other shattered cliques in $\Omega_G^{(I)}$. Suppose that edge is between $X_1$ and $X_2$, then there exist $H_1$ and $H_2$ with $X_1 \in \mathrm{ch}_X(H_1)$ and $X_2 \in \mathrm{ch}_X(H_2)$ and $H_1$ is $d$-connected to $H_2$. Let one such active path between $H_1$ and $H_2$ be $P_{1,2}$. By Lemma E.3, there exists one source node $\mathrm{so}(P_{1,2})$ (If $H_1 = H_2$, then $\mathrm{so}(P_{1,2})$ is just $H_1$). Note that we can always find one node $H_*$ in $\overline{\mathrm{an}}(\mathrm{so}(P_{1,2})$ that has no incoming edge. Let's denote that the pure child of $H_*$ is $X_*$.

Then $C_* = \cup_{H_i \in \overline{\mathrm{de}}(H_*)} \mathrm{ch}_X(H_i)$ must be a shattered maximal clique in $\Omega_G^{(I)}$. Suppose $C_*$ is not a maximal clique. Then the maximal clique that contains $C_*$ must contain another element $X_3$ not in $C_*$. There must exist another hidden variable $H_3$ with $X_3 \in \mathrm{ch}_X(H_3)$ that $H_*$ is $d$-connected to because $X_3$ is connected to the pure child of $H_*$. But this is not possible by Lemma E.3 and the fact that $H_*$ is a source node while $X_3$ is not a descendant of $H_*$.

$\qquad\square$

### D.5 Examples of measurement model with no non-replaceable fractured subsets

Lemma 4.6 shows that an imaginary subset is a fractured subset. Because a replaceable imaginary subset can be found, our concern should focus solely on the non-replaceable ones (Theorem 4.3). In fact, Corollary 4.7 can be strengthed to say that as long as there are no non-replaceable fractured subsets, then the bipartite graph can be identified. In this section, we will show a class of measurement models with no non-replaceable fractured subsets (see Remark 4.1).

Let's first present a simple example where there are multiple source nodes and the pure child assumption is violated, and there is still no non-replaceable fractured subset.

**Example 12.** Consider Figure 13. If the intervention target $I$ is $\emptyset$, $H_1$ or $H_3$, then two maximal cliques are $\{X_1, X_2, X_3\}$ and $\{X_2, X_3, X_4\}$. If the intervention target is $H_2$, then there are three maximal clique: $\{X_1, X_2\}$, $\{X_2, X_3\}$ and $\{X_3, X_4\}$. One can easily check that there is no non-replaceable fractured subset.

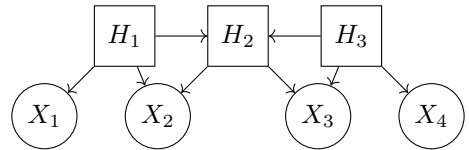

Figure 13: In this example, there is no fractured set.

In fact, we can extend Figure 13 to a family of measurement models as follows:

**Definition D.2** (Sparse Measurement Model)**.** For a given measurement model $G = (V, E)$ and $V = X \cup H$, if $X_1, X_2 \in X$ is $d$-connected, then it must be that

1. There exists at most one hidden node $H_i$ such that $\{X_1, X_2\} \subseteq \mathrm{ch}_X(H_i)$ (Type-1 path).

2. There exists at most one active path $P$ between $X_1$ and $X_2$ such that $\mathrm{pa}_P(X_1) \neq \mathrm{pa}_P(X_2)$ (Type-2 path).

And for any tuple $(X_1, X_2, X_3) \subseteq X$ such that $\{X_1, X_2\} \subseteq \mathrm{ch}_X(H_2)$ and $\{X_1, X_3\} \subseteq \mathrm{ch}_X(H_3)$ with $H_2 \neq H_3$, $X_2$ and $X_3$ can only be $d$-connected via type-2 path $P$ with the property that $\mathrm{pa}_P(X_2) = H_2$ and $\mathrm{pa}_P(X_3) = H_3$.

**Theorem D.8.** *Under Assumptions 1-2 and with maximal sparse measurement model, there is no imaginary subset.*

*Proof.* Suppose there is an imaginary subset $X'$. Then $|X'| \geq 3$. If $X'$ has only one element, then it must be a subset of $\mathrm{ch}_X(H')$ for some hidden node $H'$. If $X'$ has two nodes, by the definition of sparse measurement, there could only be one active path between them and there exists an intervention target that can break this path. So $X'$ would not be valid.

Next, we'll use induction to show that for any valid subset $X'$ with $|X'| \geq 3$, there exists a hidden node $H'$ such that $X' \subseteq \mathrm{ch}_X(H')$. In other words, $X'$ cannot be imaginary.

**Base Step:** If $|X'| = 3$, then let $X' = \{X_1, X_2, X_3\}$. Suppose $X'$ is imaginary. Then every pair of nodes in $X'$ must have a type-1 path and a type-2 path because there exists an intervention target that can destroy the type-2 path but every pair is valid. Therefore, we have $H_{1,2}$ as the common parent of $X_1, X_2$, $H_{2,3}$ as the common parent of $X_2, X_3$ and $H_{1,3}$ as the common parent of $X_1, X_3$. Note that these three hidden nodes are distinct, otherwise, $X'$ would not be imaginary. This is also not possible by the definition of the sparse measurement model.

**Inductive Hypothesis:** For any valid subset $X'$ with $|X'| = k \geq 3$, there exists a hidden node $H'$ such that $X' \subseteq \mathrm{ch}_X(H')$.

**Inductive Step:** We'll show this is true for $|X'| = k + 1$. Without loss of generality, suppose $X' = \{X_1, ..., X_{k+1}\}$. By the inductive hypothesis, $X'' = \{X_1, ..., X_k\}$ and $X''' = \{X_2, ..., X_{k+1}\}$ are both non-imaginary. In other words, there exist $H''$ and $H'''$ such that $X'' \subseteq \mathrm{ch}_X(H'')$ and $X''' \subseteq \mathrm{ch}_X(H''')$. Because $|X'| \geq 4$, both $H''$ and $H'''$ are common parent of $X_2$ and $X_3$. By the definition of the sparse measurement model, $H'' = H'''$. $\qquad\square$

**Theorem D.9.** *Under Assumptions 1-2 and with maximal sparse measurement model, there is no non-replaceable fractured subset.*

*Proof.* By Theorem D.8, there is no imaginary subset. Therefore, if $X'$ is a maximal valid subset, then $X' \subseteq \mathrm{ch}_X(H_i)$ for some hidden variable $H_i$. By definition, if $X'$ is non-replaceable, there must exist a hidden variable $H_i$ such that $X' = \mathrm{ch}_X(H_i)$.

In addition, if $X'$ is fractured, there exists a complete collection $\mathcal{S} = \{S_j\}_{j=1}^k$ such that $S_j \not\subseteq X'$ for all $S_j$. Note for any $S_j$, there exists a hidden node $H_j$ such that $S_j \subseteq \mathrm{ch}_X(H_j)$ because there is no imaginary subset.

First of all, $X'$ must have at least two elements in it. If $|X'| = 1$, then by the subset condition (Lemma C.1), no other hidden variable can be a parent of the single node in $X'$ and the aforementioned complete collection does not exist.

Let $X_1, X_2$ be two elements in $X'$. Because the measurement model is sparse, by Lemma E.4, there exists an intervention target $I$ (could be the empty set) such that, the type-2 path between $X_1, X_2$ is destroyed. Under this intervention target, there must exist a $C \in D(G^{(I)})$ that is shattered by $\mathcal{S}$ and contains $\{X_1, X_2\}$. Therefore, there exists a subset of $\mathcal{S}$: $\{S_j'\}_{j=1}^{k'}$ with $\{X_1, X_2\} \subseteq C = \cup S_j'$.

Because $S_j' \not\subseteq X'$, there exist two maximal subsets $S_1'$ and $S_2'$ such that $X_1 \in S_1'$ and $X_2 \in S_2'$. This also means that there exist $H_1$ and $H_2$ with $X_1 \in S_1' \subseteq \mathrm{ch}_X(H_1)$ and $X_2 \in S_1' \subseteq \mathrm{ch}_X(H_2)$. $H_1$

and $H_2$ must be two different nodes other than $H_i$ by the defition of the sparse measurement model. In addition, because $S_1' \not\subseteq X'$, there exists $X_3 \in S_1'$ where $X_3 \notin X'$.

Because $\{X_1, X_2\} \subseteq \mathrm{ch}_X(H_i)$ and $\{X_1, X_3\} \subseteq \mathrm{ch}_X(H_1)$ and $X_2, X_3$ are $d$-connected under the intervention target $I$, by the definition of the sparse measurement model, $X_2$ and $X_3$ are $d$-connected via the path between $H_i$ and $H_1$. This is not possible because this path creates another type-2 path between $X_1$ and $X_2$.

$\square$

## D.6 Useful lemmas

The first two lemmas show that we only need to care about maximal cliques of undirected dependency graphs.

**Lemma D.10.** *For two measurement model $G_1$ and $G_2$ and two intervention targets $I_1$ and $I_2$, we have that $\Omega_{G_1}^{(I_1)} = \Omega_{G_2}^{(I_2)}$ if and only if $\mathcal{T}_X(G_1^{(I_1)}) = \mathcal{T}_X(G_2^{(I_2)})$.*

*Proof.* If $\mathcal{T}_X(G_1^{(I_1)}) = \mathcal{T}_X(G_2^{(I_2)})$, then $D(G_1^{(I_1)})$ is the same as $D(G_2^{(I_2)})$ by our construction of undirected dependency graphs. The obviously $\Omega_{G_1}^{(I_1)} = \Omega_{G_2}^{(I_2)}$. On the other hand, if $\Omega_{G_1}^{(I_1)} = \Omega_{G_2}^{(I_2)}$, then $D(G_1^{(I_1)})$ is the same as $D(G_2^{(I_2)})$ by Lemma D.11. By Proposition 6 of [43], $\mathcal{T}_X(G_1^{(I_1)}) = \mathcal{T}_X(G_2^{(I_2)})$.

$\square$

**Lemma D.11.** *If $G_1 = (V, E_1)$ and $G_2 = (V, E_2)$ share the same maximal cliques, then $E_1 = E_2$.*

*Proof.* Let's first show $E_1 \subseteq E_2$. Suppose $(A \to B) \in E_2$ and $(A \to B) \notin E_1$. Then there exists a maximal clique that contains $A, B$ for $G_2$. But there is no clique that contains $A, B$ at the same time for $G_1$. Therefore $E_1 \subseteq E_2$. Similarly, $E_2 \subseteq E_1$.

$\square$

**Lemma D.12.** *Under Assumption 1(a) and (b), two distributions $P_1$ and $P_2$ over two measurement models $G_1$ and $G_2$ share the same UDG if and only if $\mathcal{T}_X(G_1) = \mathcal{T}_X(G_2)$.*

*Proof.* Because of Assumption 1(a) and (b), $D(P_1) = D(G_1)$ and $D(P_2) = D(G_2)$. By Lemma D.10 and Lemma D.11, the statement is true.

$\square$

**Lemma D.13.** *If two hidden nodes $H_i$ and $H_j$ are $d$-connected in $G^{(I)}$ under intervention target $I$, then there exists a maximal clique $C \in \Omega_G^{(I)}$ such that $\mathrm{ch}_X(H_i) \cup \mathrm{ch}_X(H_j) \subseteq C$*

*Proof.* Because $H_i$ and $H_j$ are $d$-connected, $\mathrm{ch}_X(H_i)$ and $\mathrm{ch}_X(H_j)$ form a clique. Therefore, there exists a maximal clique that is a superset of $\mathrm{ch}_X(H_i) \cup \mathrm{ch}_X(H_j)$.

$\square$

**Lemma D.14.** *If a set of two observed elements $\{X_a, X_b\}$ is valid and $X_a, X_b$ do not share the same parent, then one of $X_a$ and $X_b$ must have at least two parents.*

*Proof.* $X_a$ and $X_b$ do not share the same parent. Therefore, there must exist at least two hidden nodes $H_{a,1}$ and $H_{b,1}$ such that $X_a \in H_{a,1}$ and $X_b \in H_{b,1}$. However, if there are only two such hidden nodes, Lemma E.4 suggests there exists an intervention target $I$ that makes $X_a$ and $X_b$ $d$-separated and thus $X_a$ and $X_b$ $d$-separated. This is a violation of the fact that $\{X_a, X_b\}$ is a valid subset.

$\square$

**Lemma D.15.** *For a hidden variable $H_i$, there does not exist an observed node $X_i$ such that $X_i \notin \mathrm{ch}_X(H_i)$ but under any intervention target $I \in \mathcal{I}$, $X_i$ is a descendant of $H_i$.*

*Proof.* This violates maximality.

Consider $G' = (V, E')$ with $E' = E \cup \{H_i \to X_i\}$. For any intervention target $I$, $D(G'^{(I)})$ does not have more edges than $D(G^I)$. If the opposite is true, then the new edge in $D(G'^{(I)})$ must involve $X_i$. Suppose the new edge is between $X_i$ and $X_j$, it must be that a parent of $X_j$ is $d$-connected to $H_i$. However, because $X_i$ is a descendant of $H_i$, by Lemma E.3, $X_i$ must be $d$-connected to $X_j$ in $D(G^I)$. Therefore, by Lemma D.10, $\{\mathcal{T}_X(G^{(I)})\}_{I \in \mathcal{I}}$ is the same as $\{\mathcal{T}_X(G^{(I)})\}_{I \in \mathcal{I}}$.

By Lemma D.17, the existence of $G'$ violates the maximality condition.

$\square$

**Lemma D.16.** $\{\mathrm{ch}_X(H_i)\}_{i=1}^m$ *is a complete cover.*

*Proof.* By Assumption 1(a) and Assumption 1(b), two observed variables $X_i$ and $X_j$ are dependent if and only if $X_i$ and $X_j$ are $d$-connected. Suppose $\{\mathrm{ch}_X(H_i)\}_{i=1}^m$ is not a complete cover. Then for some intervention target $I \in \mathcal{I}$, there exists one clique $C$ in $D(G^{(I)})$ that is not shattered and that clique contains one edge that is not covered by other shattered cliques in $D(G^{(I)})$. Suppose that edge is between $X_1$ and $X_2$, then there exist $H_1$ and $H_2$ with $X_1 \in \mathrm{ch}_X(H_1)$ and $X_2 \in \mathrm{ch}_X(H_2)$ and $H_1$ is $d$-connected to $H_2$. By Lemma D.13, this is not possible. $\qquad\square$

**Lemma D.17.** *If $G = (X \cup H, E)$ is a maximal measurement model, then there does not exist a measurement model $G' = (X \cup H, E')$ with $E' = E \cup \{H_i \to X_i\}$ for some hidden variable $H_i$ and observed variable $X_i$ such that for any intervention target $I \in \mathcal{I}$, where $\mathcal{I}$ is the complete family of intervention targets,*

$$\mathcal{T}_X(G^{(I)}) = \mathcal{T}_X(G'^{(I)})$$

*Proof.* By the definition of the maximal measurement model, we just need to show that $G'$ also satisfies Assumption 1. If $G'$ satisfies Assumption 1, then the existence of $G'$ would mean that $G$ is not a maximal measurement model. Before delving into the graphical assumptions, one first notices that because for any intervention target $I \in \mathcal{I}$, $\mathcal{T}_X(G^{(I)}) = \mathcal{T}_X(G'^{(I)})$, then $X_i$ must always be $d$-connected to $\mathrm{ch}_X(H_i)$ in both $G^{(I)}$ and $G'^{(I)}$ under any intervention target.

For Assumption 1(a), if a distribution $P$ is Markov to $G$, then it must be Markov to $G'$. This is because $G'$ has one more edge than $G$, any active path in $G$ would remain active in $G'$ as well. Thus $\mathcal{T}(G') \subseteq \mathcal{T}(G) \subseteq \mathcal{T}(P)$. A similar argument can be made for the interventional case.

For Assumption 1(b), we know that for any intervention target $I$, $X_i \perp\!\!\!\perp_{P^{(I)}} X_j \implies$ d-sep$_{G^{(I)}}(\{X_i\}\{X_j\} \,|\, \emptyset)$. Because $\mathcal{T}_X(G^{(I)}) = \mathcal{T}_X(G'^{(I)})$, Assumption 1(b) is true for $G'$ as well.

For Assumption 1(c), because $\mathcal{T}_X(G^{(I)}) = \mathcal{T}_X(G'^{(I)})$ for any intervention target and $G', G$ only differs in biparite graph. If Assumption 1(c) is satisfied by $G$, it must be satisfied by $G'$ as well.

Therefore, $G'$ satisfies Assumption 1 and $G$ is not maximal, which is a contradiction. $\qquad\square$

## E  Learning the skeleton of latent structures

Once we have learned the bipartite graph $G_B$, the next step is to learn the DAG $G_H$ over the latent variables $H$. This turns out to be straightforward: Assumption 1(c) suggests that two hidden variables $H_i$ and $H_j$ are $d$-separated if and only if $\mathrm{ch}_X(H_i)$ and $\mathrm{ch}_X(H_j)$ are in different cliques (Lemma E.2). Therefore, the idea is to learn causal graphs using unconditional $d$-separations of latent variables under interventions.

Define $\mathcal{M}_H$ as follows:

$$\mathcal{M}_H(G) := (\{\langle A, B \rangle : \mathrm{d\text{-}sep}_G(AB \,|\, \emptyset) \text{ for disjoint subsets } A, B \subseteq H\}).$$

Note that $\mathcal{M}_H(G) \subseteq \mathcal{T}_H(G)$ because $\mathcal{M}_H(G)$ only stores unconditional $d$-separations. Therefore, to learn the latent DAG, we would like to answer the following question:

> *Given $\{\mathcal{M}_H(G^{(I)}))\}_{I \in \mathcal{I}}$, can we recover $G_H$?*

*Remark* E.1. This is harder than the fully observational case where we have access to all *conditional* $d$-separations.

The answer to the previous question is affirmative as shown by the next theorem (see Appendix E.2).

**Theorem E.1.** *Under Assumptions 1-2 and suppose the bipartite graph $G_B$ is correct, the skeleton of $G_H$ is identifiable.*

### E.1  Identifying unconditional $d$-separations of the latents

Given the bipartite graph $G_B$ and $\{\mathcal{T}_X(P^{(I)})\}_{I \in \mathcal{I}}$, we can easily construct $\{\mathcal{M}_H(G^{(I)}))\}_{I \in \mathcal{I}}$ by the following lemma.

**Lemma E.2.** *Given intervention target $I$, two hidden variables $H_i$ and $H_j$ are d-separated in $G^{(I)}$ if and only if there does not exist a clique $C \in \Omega_P^{(I)}$ such that $\mathrm{ch}_X(H_i) \cup \mathrm{ch}_X(H_j) \subseteq C$.*

*Proof.* By Assumption 1(a) and Assumption 1(b), two observed variables $X_i$ and $X_j$ are dependent if and only if $X_i$ and $X_j$ are d-connected.

If $H_i$ and $H_j$ are d-separated in $G^{(I)}$, then, by Assumption 1(c), there exists $X_i, X_j \in X$ where $X_i \in \mathrm{ch}_X(H_i)$, $X_j \in \mathrm{ch}_X(H_j)$, such that $X_i$ and $X_j$ are independent. Therefore, $\mathrm{ch}_X(H_i) \cup \mathrm{ch}_X(H_j)$ is not a clique.

On the other hand, if there exist a clique $C \in \Omega_P^{(I)}$ such that $\mathrm{ch}_X(H_i) \cup \mathrm{ch}_X(H_j) \subseteq C$. This means that $\mathrm{ch}_X(H_i) \cup \mathrm{ch}_X(H_j)$ is a clique. Then $H_i$ and $H_j$ must be d-connected, which is a contradiction. $\qquad\square$

### E.2 Algorithm

Now we have Algorithm 1 to find the skeleton.

---

**Algorithm 1** Learning Skeleton of $G_H$

---

**Input:** $\{\mathcal{M}_H(G^{(I)})\}_{I \in \mathcal{I}}$
1   $E \leftarrow \emptyset$ ;                                        `// Set of unoriented edges`
   `/* Step 0`                                                  `*/`
2   **if** $\{\mathcal{M}_H(G^{(I)})\}_{I \in \mathcal{I}}$ *only has one distinct element* **then**
     $\lfloor$ **Output:** $E$
   `/* Step 1:  Remove any pair of hidden variables that appears twice`     `*/`
3   **for** $\mathcal{M}_H(G^{(I)}) \in \{\mathcal{M}_H(G^{(I)})\}_{I \in \mathcal{I}}$ **do**
4     **for** $\langle H_1, H_2 \rangle \in \mathcal{M}_H(G^{(I)})$ **do**
5       If $\langle H_1, H_2 \rangle$ appears twice with at least two different intervention targets, delete $\langle H_1, H_2 \rangle$ from $\{\mathcal{M}_H(G^{(I)})\}_{I \in \mathcal{I}}$.
   `/* Step 2:  Add unoriented edges`                                   `*/`
6   **for** $\mathcal{M}_H(G^{(I)}) \in \{\mathcal{M}_H(G^{(I)})\}_{I \in \mathcal{I}}$ **do**
7     **for** $\langle H_1, H_2 \rangle \in \mathcal{M}_H(G^{(I)})$ **do**
8       $\lfloor$ $E \leftarrow E \cup \{\langle H_1, H_2 \rangle\}$
**Output:** $E$

---

**Theorem E.1.** *Under Assumptions 1-2 and suppose the bipartite graph $G_B$ is correct, the skeleton of $G_H$ is identifiable.*

*Proof.* The proof relies on the correctness of Algorithm 1 which we show here.

First, suppose $\{\mathcal{M}_H(G^{(I)})\}_{I \in \mathcal{I}}$ only has one distinct element. Then there is no edge between latent variables in $G_H$. Suppose, on the contrary, there is an edge $H_1 \rightarrow H_2$ between two hidden variables $H_1$ and $H_2$ in $G_H$, then by Lemma E.2, $\mathrm{ch}_X(H_1) \cup \mathrm{ch}_X(H_2)$ must be a clique in $D(G^{(\emptyset)})$. But by Lemma E.3, when the intervention target $I$ is $H_2$, $H_1$ and $H_2$ would be d-separated, and by Lemma E.2, $\mathrm{ch}_X(H_1) \cup \mathrm{ch}_X(H_2)$ must not be a clique in $D(G^{(I)})$. Therefore, there must be at least two distinct elements in $\{\mathcal{M}_H(G^{(I)})\}_{I \in \mathcal{I}}$.

Now, let's consider the case after step 0. Let's denote the unoriented edge set returned by Algorithm 1 as $E_1$ and the true unoriented edge set of $G_H$ as $E_2$.

     A. $E_2 \subseteq E_1$. By Lemma E.3, $\{\mathcal{M}_H(G^{(I)})\}_{I \in \mathcal{I}}$ has all pairs of hidden variables. And we removed all the pair that does not have an edge in $E_2$ by Lemma E.5.

     B. $E_1 \subseteq E_2$. Suppose $\exists \langle H_1, H_2 \rangle \in E_1$ such that $\langle H_1, H_2 \rangle \notin E_2$. Note that $\langle H_1, H_2 \rangle$ only appears with one intervention target and by Lemma E.5, this is not possible. $\qquad\square$

### E.3 Useful Lemmas

**Lemma E.3.** *Given a DAG $G = (V, E)$. If two nodes $A, B \in V$ are d-connected by the empty set, then at least one of the following must be true*

- A. *They share a common ancestor (common ancestor path)*

- B. *There are directed paths between the two nodes and all the directed paths must have the same direction (directed path)*

*Proof.* Because $A, B$ are d-connected, there must exist active paths between the two nodes. There are four possibilities of active path

- A. $A \to \ldots \to B$. This could be a direct path if every edge points to the same direction. If one of the intermediate edges points in the opposite direction. Then we would have a collider which makes the path inactive.

- B. $A \leftarrow \ldots \leftarrow B$. The same as the first case.

- C. $A \to \ldots \leftarrow B$. This is not possible because there must exist a collider on the path.

- D. $A \leftarrow \ldots \to B$. $A$ and $B$ must share a common ancestor in this path unless there is a collider which is not possible.

Obviously, we cannot have two directed paths pointing in opposite directions because that would create a circle. □

**Lemma E.4.** *For any two hidden variables $H_1$ and $H_2$, there exists at least one intervention target $I \in \mathcal{I}$, such that $\langle H_1, H_2 \rangle \in \mathcal{M}_H(G^{(I)})$.*

*Proof.* If $H_1$ and $H_2$ are $d$-separated in $G^{(\emptyset)}$, then we can just choose $I = \emptyset$.

If $H_1$ and $H_2$ are d-connected in $G^{(\emptyset)}$ and assume, without loss of generality, the directed paths between $H_1$ and $H_2$, if exist, point to $H_2$, then based on Lemma E.3, if we intervene on $H_2$, all the active paths between $H_1$ and $H_2$ would disappear. □

**Lemma E.5.** *If $\{\mathcal{M}_H(G^{(I)})\}_{I \in \mathcal{I}}$ has at least two distinct elements, then for any two hidden variables $H_1$ and $H_2$, there is no direct edge between $H_1$ and $H_2$ if and only if there exists at least two intervention targets $I_1, I_2 \in \mathcal{I}$, such that $\langle H_1, H_2 \rangle \in \mathcal{M}_H(G^{(I_1)})$ and $\langle H_1, H_2 \rangle \in \mathcal{M}_H(G^{(I_2)})$.*

*Proof.* If there is no direct edge between $H_1$ and $H_2$ and there exists only one intervention target $I_1 \in \mathcal{I}$, such that $\langle H_1, H_2 \rangle \in \mathcal{M}_H(G^{(I_1)})$, then $I_1 \neq \emptyset$. There is because if $I_1 = \emptyset$ and $\langle H_1, H_2 \rangle \in \mathcal{M}_H(G^{(\emptyset)})$. Then there are no active paths between $H_1$ and $H_2$ and deleting edges via interventions would not create active paths. Because $\{\mathcal{M}_H(G^{(I)})\}_{I \in \mathcal{I}}$ has at least two distinct elements, there must exist another intervention target such that $\langle H_1, H_2 \rangle$ appears. By the proof of Lemma E.6, one of $H_1$ and $H_2$ is the intervention target $I_1$. Without loss of generality, suppose $I_1 = \{H_1\}$, we can add $H_1 \leftarrow H_2$ to the graph and it would violate the maximality condition. Thus, if there is no direct edge between $H_1$ and $H_2$, then there exists at least two intervention targets $I_1, I_2 \in \mathcal{I}$, such that $\langle H_1, H_2 \rangle \in \mathcal{M}_H(G^{(I_1)})$ and $\langle H_1, H_2 \rangle \in \mathcal{M}_H(G^{(I_2)})$.

If there exists at least two intervention targets $I_1, I_2 \in \mathcal{I}$, such that $\langle H_1, H_2 \rangle \in \mathcal{M}_H(G^{(I_1)})$ and $\langle H_1, H_2 \rangle \in \mathcal{M}_H(G^{(I_2)})$, then suppose there is a direct edge between $H_1$ and $H_2$. Without loss of generality, let's assume $H_1 \to H_2$. Then two variables would *only* be $d$-separated when intervening on $H_2$ which is a contradiction. □

**Lemma E.6.** *In Algorithm 1, after step 1, for any intervention target $I$ such that $\mathcal{M}_H(G^{(I)}) \neq \mathcal{M}_H(G^{(\emptyset)})$, then there exist a hidden variable $H_1$ such that $H_1 \in \langle H_i, H_j \rangle$ for all $\langle H_i, H_j \rangle \in \mathcal{M}_H(G^{(I)})$ and $H_1$ is the intervention target $I$.*

*Proof.* We just need to show that if $\langle H_i, H_j \rangle \in \mathcal{M}_H(G^{(I)})$ and $\langle H_i, H_j \rangle \notin \mathcal{M}_H(G^{(I')})$ for $I' \neq I$, then one of $H_i$ or $H_j$ must be the intervention target $I$. This is because, after step 1, every pair of hidden variables only appear once.

Suppose the opposite is true, then neither $H_i$ nor $H_j$ is being intervened on. So there is no direct edge between $H_i$ and $H_j$ since otherwise $H_i$ and $H_j$ would be $d$-connected. On the other hand, $\langle H_i, H_j \rangle \notin \mathcal{M}_H(G^{(I')})$ for $I' \neq I$, implies that $H_i$ and $H_j$ are $d$-connected in the observational distribution because $\mathcal{M}_H(G^{(I)}) \neq \mathcal{M}_H(G^{(\emptyset)})$. Then by Lemma E.3, we can find another intervention target $H_i$ or $H_j$ such that $\langle H_i, H_j \rangle$ would appear again which would be a contradiction. $\qquad\square$

## F   Limitations of edge orientations

Unlike known interventions, with only access to CI information, edge orientation might not always be possible even when there are no latent variables as shown in Example 1. But it is also possible as demonstrated by Example 13. This raises the question of which edges provably *cannot* be oriented.

In this section, we attempt to answer this question by studying equivalence relations between DAGs under unknown interventions. Such equivalence relations are finer partitions of the Markov equivalence class. *The results are purely graphical* and can be studied in their own right. If assuming Markov and faithfulness, it also shows the limitations of edge orientations with access to CI information only. In addition, one could use conditional invariances to improve the identifiability of edge orientations. But it might require additional assumptions like direct $\mathcal{I}$-faithfulness[58] which we do not make in this paper.

**Example 13.**   Consider the CPDAG $G$ shown in Figure 14. Let $G_1$ be a consistent extension of $G$ such that $X_1 \to X_2$ and $I_1 = X_2$. Then $\{(X_1, X_2, \emptyset), (X_3, X_2, \emptyset)\} \subseteq \mathcal{T}(G_1^{(I_1)})$. $G_2$ is another consistent extension of $G$ where $X_2 \to X_1$ but there does not exist an intervention target that can induce $\mathcal{T}(G_1^{(I_1)})$. Therefore, if we observe $\mathcal{T}(G_1^{(I_1)})$, then we know that $X_1 \to X_2$.

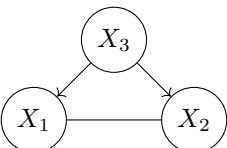

Figure 14: CPDAG of three variables with no latent variables.

In this section, we consider the *general case of an arbitrary, fully observed DAG (i.e. $H = \emptyset$)* instead of the measurement model. Given the skeleton of the DAG, all the compelled edges in the MEC can already be identified without access to interventional distributions. Therefore, without loss of generality, we assume the CPDAG is given and study if unknown hard interventions on single nodes can help orient reversible edges. Furthermore, because isolated edges are covered, they must be reversible in the MEC as well (cf. Lemma G.1), there is no loss of generality in focusing on reversible edges.

To state the problem formally:

> *Given a CPDAG $G$, we want to orient its reversible edges given a list $\{\mathcal{T}(G_*^{(I)})\}_{I \in \mathcal{I}}$ for a family of intervention targets $\mathcal{I}$ and $G_*$, which is the true consistent extension of $G$. In other words, is the tuple $\langle G_*, \mathcal{I} \rangle$ the unique one that can induce $\{\mathcal{T}(G_*^{(I)})\}_{I \in \mathcal{I}}$?*

Let's start with some definitions. Recall the definition of *covered edges*[12]:

**Definition B.3** (Covered Edge). We say an edge $x \to y$ is covered if $x$ and $y$ share the same parent excluding $x$ (i.e. $\mathrm{pa}(x) \cup \{x\} = \mathrm{pa}(y)$).

In particular, an *isolated edge* is a special covered edge.

**Definition 3.3** (Isolated edge). We say an edge $x \to y$ is **isolated** if $x$ does not have any parent ($\mathrm{pa}(x) = \emptyset$) and $y$ only has $x$ as its parent ($\mathrm{pa}(y) = \{x\}$).

*Remark* F.1. Isolated edges are trivially covered.

## F.1 Isolated equivalence class

Chickering [12] show that two Markov equivalent DAGs can be transformed into one another by a sequence of covered edge reversals. The isolated equivalence class is also defined in a transformational fashion. We restate the definition below.

**Definition F.1.** (Isolated equivalence class) Two DAGS $G_1$ and $G_2$ are isolated equivalent, denoted $G_1 \sim_E G_2$, if there exists a sequence of isolated edge reversals.

The next two theorems show that it is impossible to distinguish DAGs in an IEC by looking at $d$-separations only.

**Theorem F.2.** *Let $G_1$ and $G_2$ be two consistent extensions of CPDAG $G = (V, E)$. Suppose*

1. *$G_1$ and $G_2$ only differs in one isolated edge*

2. *All the interventional targets are single nodes and the intervention is hard.*

*Suppose the tuple $(G_1, \mathcal{I}_1)$ induces $\{\mathcal{T}(G_1^{(I)})\}_{I \in \mathcal{I}_1}$.*

*Then there exists a family of interventional targets $\mathcal{I}_2$ such that the tuple $(G_2, \mathcal{I}_2)$ also induces $\{\mathcal{T}(G_1^{(I)})\}_{I \in \mathcal{I}_1}$.*

**Theorem 5.3.** *Suppose $G_1$ and $G_2$ are in the same IEC. If the tuple $(G_1, \mathcal{I}_1)$ induces $\{\mathcal{T}(G_1^{(I)})\}_{I \in \mathcal{I}_1}$, then there exists a family of interventional targets $\mathcal{I}_2$ (possibly the same as $\mathcal{I}_1$) such that the tuple $(G_2, \mathcal{I}_2)$ also induces $\{\mathcal{T}(G_1^{(I)})\}_{I \in \mathcal{I}_1}$.*

*Proof.* Because $G_1$ and $G_2$ are in the same IEC, then, by definition, there exists a sequence of isolated edge reversals. Then we can prove by applying Theorem F.2 repeatedly. $\square$

### F.1.1 Proof of Theorem F.2

**Definition F.3.** (Augmented Active Subgraph) Let $G = (V, E)$ be a DAG and let path $P : V_1 \to \ldots \to V_k$ be an active path given $C$ where $C \subseteq V$. Suppose $H_i$ is a collider, then there must exist at least a node $V_d$ in $\overline{\mathrm{de}}(V_i)$ where $V_d \in C$. We define an **active descendent path** of $V_i$ to be a directed path from $V_i$ to $V_d$. An **augmented active subgraph** of $P$ is a subgraph including the active path $P$ and all the active descendent paths of colliders in the original active path.

*Remark* F.2. Our definition also considers the special case where $V_i = V_d$ and the directed path is just $V_i \to V_i$.

**Lemma F.4.** *Let $G = (V, E)$ be a DAG. Let $I$ be a single node hard intervention target, then,*

$$\mathcal{T}(G) \subseteq \mathcal{T}(G^I)$$

*Proof.* Doing a hard intervention on any node $V_i \in V$ removes all incoming edges to $H_i$.

Suppose the statement is not true, then there exists $(A, B, C) \in \mathcal{T}(G)$ such that $(A, B, C) \notin \mathcal{T}(G^I)$. This means that intervening on $V_i$ creates an active path between $A, B$ given $C$. But deleting edges would not create a new path. The only way an active path can emerge is if a previously blocked path becomes unblocked after an intervention.

A path is blocked if either:

1. any non-collider is being conditioned on.

2. no descendants of collider (including itself) is being conditioned on

For the first condition, we know that deleting edges would not make non-colliders colliders. On the other hand, deleting edges would also not make non-descendants descendants (intervening on the collider itself would destroy the path). Therefore, hard intervening on a single node would not create new active paths. $\square$

**Lemma F.5.** *Let $G$ be a DAG. Suppose there is an active path $P$ in $G$, and the hard intervention target $I$ is not on the augmented active subgraph of the active path. Then this path stays active.*

*Proof.* Because the intervened node is not on the path nor on the directed path going out from one of the colliders on the active path. Deleting its incoming edges would not change these paths. □

**Lemma F.6.** *Let $G_1$ and $G_2$ be two DAGs that share the same augmented active subgraph of active path $P$ given conditioning set $C$. If there is a single node hard intervention on one of the nodes of that augmented active subgraph in both $G_1$ and $G_2$, then either the path is active in both $G_1$ and $G_2$ or it is not active in either $G_1$ or $G_2$.*

*Proof.* Intervening on any node on the active path would destroy (if the intervened node has incoming edges) or sustain (if the intervened node has no incoming edges) the path in both $G_1$ and $G_2$.

Suppose we intervene on a descendant of a collider on the active path that is not the collider itself. Any active descendant path that has the intervened node on it would get destroyed. The path would stay active if there exists an alternative active descendant path that does not involve the intervened node, which is shared between $G_1$ and $G_2$ because they share the same augmented active subgraph. Note that intervening on a node would not create a new descendant path. □

*Proof of Theorem F.2.* For notation, let's suppose the isolated edge is $V_i \to V_j$ in $G_1$.

To prove this, we just need to show that for any $I_i \in \mathcal{I}_1$, there exists $I_j$ such that

$$\mathcal{T}(G_1^{(I_i)}) = \mathcal{T}(G_2^{(I_j)})$$

In particular, we will show that we can choose $I_j = \{I_i\}$ when $I_i \neq \{V_i\}$ and $I_j = \{V_j\}$ when $I_i = \{V_i\}$.

By the definition of Markov equivalence, we know that

$$\mathcal{T}(G_1) = \mathcal{T}(G_2)$$

First of all, note that an augmented active subgraph given conditioning set $C$ in $G_1$ must exist and be active given conditioning set $C$ in $G_2$ and vice versa. We know that two DAGs in the same Markov equivalence class share the same skeleton and v-structures. Therefore, for any augmented active subgraph given conditioning set $C$ in $G_1$, there is a subgraph with the same skeleton in $G_2$. If the edges of the augmented active subgraph in $G_1$ have the same orientations in $G_2$, then it is active in $G_2$. If some edges of the augmented active subgraph in $G_1$ have different orientations than that of $G_2$. Then the subgraph can only differ in one isolated edge. Because the endpoints of isolated edge cannot have other parents, the isolated edge cannot be in the descendant paths. The isolated edge can be in the active path itself, with edges going out from endpoints of isolated edge. When that edge is reversed, the path is still active.

To show that $\mathcal{T}(G_1^{(I_i)}) = \mathcal{T}(G_2^{(I_j)})$, we first need to show that for any $(A, B, C) \in \mathcal{T}(G_1^{(I_i)})$, $(A, B, C) \in \mathcal{T}(G_2^{(I_j)})$. By Lemma F.4, $\mathcal{T}(G_1) \subseteq \mathcal{T}(G_1^{(I_i)})$ and $\mathcal{T}(G_2) \subseteq \mathcal{T}(G_2^{(I_j)})$. We can only consider tuple $(A, B, C)$ where $(A, B, C) \notin \mathcal{T}(G_1)$.

If $(A, B, C) \notin \mathcal{T}(G_1)$, but $(A, B, C) \in \mathcal{T}(G_1^{(I_i)})$, then all the active paths between $A$ and $B$ given $C$ must be blocked or deleted after intervention on $I_i$. Let's consider the following two cases:

1. Suppose an active path has the same augmented subgraph in both $G_1$ and $G_2$, then it will get blocked in $G_2$ when intervened on $I_j = I_i$. Note that $I_i$ is neither $V_i$ nor $V_j$. Because $V_i \to V_j$ is isolated, they do not have other parents. Intervening on them will not block/delete that path.

2. Suppose an augmented subgraphs in $G_1$ and an augmented subgraphs in $G_2$ differ in one isolated edge. If the intervention target $I_i$ is $V_i$, then we can choose $I_j$ to be $V_j$ (intervening on $V_j$ in $G_2$ will not delete that path). If the intervention target is neither $V_i$ nor $V_j$, then we can keep $I_j$ to be the same as $I_i$. Because $V_i \to V_j$ is an isolated edge, this edge must be on the active path itself and not on the active descendant paths. Suppose the active path $P$ is $V_1 \to ... \to V_k$, then the subpath $P_1 : V_1 \to ... \to V_i$ and $P_2 : V_j \to ... \to V_k$ have the same augmented subgraphs in both $G_1$ and $G_2$. By Lemma F.6, the intervention would have the same effect on them.

Therefore, if $I_j = \{I_i\}$ when $I_i \neq \{V_i\}$ and $I_j = \{V_j\}$ when $I_i = \{V_i\}$. By similar argument, $\mathcal{T}(G_2^{(I_j)}) \subseteq \mathcal{T}(G_1^{(I_i)})$. $\qquad\square$

### F.1.2 Orienting Non-isolated Edges

While it is impossible to distinguish within an IEC, it is possible to do so between IECs.

**Theorem F.7.** *Let $G_1$ and $G_2$ be two DAGs. Suppose*

    A. *$G_1$ and $G_2$ are Markov equivalent.*

    B. *$G_1$ and $G_2$ are not isolated equivalent.*

*Then there exists a single-node hard intervention target $I_1$ such that*

$$\mathcal{T}(G_1^{(I_1)}) \neq \mathcal{T}(G_2^{(I_2)}) \quad \forall I_2$$

*where $I_2$ is also a single-node hard intervention target.*

*Proof.* Let $\Delta(G_1, G_2)$ denote the set of edges in $G_1$ that have opposite orientations in $G_2$.

By theorem 2 of [12], we know that there exists a sequence of $|\Delta(G_1, G_2)|$ distinct covered edge reversals that can transform $G_1$ to $G_2$. And each covered edge reversal creates a DAG in the same Markov equivalence class.

Because $G_1$ and $G_2$ are not isolated equivalent, there exist DAGs $G_m$ and $G_n$ in the sequence such that $G_m$ and $G_n$ differ by a covered edge where endpoints of the edge share at least one additional parent node. Without loss of generality, let $G_m$ and $G_n$ be the first pair in the sequence to have covered but not isolated edge reversal. In particular, the sequence is $G_1 \to ... \to G_m \to G_n \to ... \to G_2$.

For simplicity, suppose the covered edge in $G_m$ is $A \to B$ and the common parent is $C$. Obviously, in $G_n$, the covered edge is reversed ($A \leftarrow B$). Suppose there is an intervention target $I_1'$ on $G_m$ that is $\{B\}$, then edges $C \to B$ and $A \to B$ get removed. Therefore, there exist subsets of vertices $S_1$ and $S_2$ such that $C$ and $B$ are d separated by $S_1$, and $A$ and $B$ are d separated by $S_1$.

However, for all DAGS in the subsequence after $G_m$ including $G_n$ and $G_2$. The edge is reversed to be $A \leftarrow B$ (Note that the sequence has $|\Delta(G_1, G_2)|$ distinct covered edge reversals that can transform $G_1$ to $G_2$. A reversed edge cannot be reversed again). There does not exist a single node intervention target that can remove both edges between $A$, $B$, and $B$, $C$.

Therefore,

$$\mathcal{T}(G_m^{(I_1')}) \neq \mathcal{T}(G_2^{(I_2)}) \quad \forall I_2$$

By our construction, $G_m$ and $G_1$ are in the same IEC. By Theorem F.2, there exists an intervention target $I_1$ such that,

$$\mathcal{T}(G_1^{(I_1)}) \neq \mathcal{T}(G_2^{(I_2)}) \quad \forall I_2. \qquad\square$$

### F.1.3 Useful Lemmas

The first lemma characterizes the uncovered reversible edges.

**Lemma F.8.** *Let $G_1$ be a consistent extension of a CPDAG $G$. Suppose there is an uncovered edge $e : A \to B$ in $G_1$ and the edge is unoriented in the CPDAG, then there must exist a node $C$ in $G_1$ such that $C \to B$ and $C \leftarrow A$.*

*Proof.* Because $e$ is uncovered, there are two cases.

    A. There is a node $C$ in $G_1$ that is a parent of $B$ but not a parent of $A$. Suppose there is no edge between $C$ and $A$. Then because $e_1$ is unoriented, changing its direction would destroy a $v$-structure. Therefore there must be an edge between $C$ and $A$ and because $C$ is not a parent $A$, we have $C \leftarrow A$.

B. There is a node $C$ in $G_1$ that is a parent of $A$ but not a parent of $B$. By a similar argument, Therefore there must be an edge between $C$ and $B$. Because $C$ is not a parent of $B$, we have $C \leftarrow B$. In this case, we have a circle. Therefore, this is not a valid case. $\quad\square$

The next lemma characterizes reversible edges in an IEC.

**Lemma F.9.** *Let $G_1$ and $G_2$ be two DAGs in the same IEC. Let $e_1$ be a reversible edge in $G_1$ that has the opposite orientation in $G_2$. Then, $e_1$ must be isolated.*

*Proof.* Let's denote $e_1$ in $G_1$ as $A \leftarrow B$. We can consider two cases.

A. $e_1$ is a covered edge in $G_1$.

   Suppose, on the contrary, $e_1$ is non-isolated. By definition, there exists a common parent $C$.

   Because $e_1$ has the opposite orientation in $G_2$, we have $A \leftarrow B$ in $G_2$. $G_1$ and $G_2$ are isolated equivalent and thus Markov equivalent. They must share the same skeleton. So there must be edges connecting $C, A$, and $C, B$ in $G_2$. By definition, there exists a sequence of isolated edge reversals from $G_2$ to $G_1$. There are only three valid possibilities (the fourth one is ignored because it creates a circle) in $G_2$:

   (a) $C \to A$, $C \leftarrow B$. Out of all the three edges among $A, B, C$, the first edge that might be reversed in the sequence is $C \leftarrow B$. But regardless of the orientation of the edge between $B$ and $C$, one cannot reverse $A \leftarrow B$ in the sequence as it cannot be an isolated edge.
   (b) $C \to A$, $C \to B$. Out of all the three edges among $A, B, C$, the first edge that might be reversed in the sequence is $C \to A$. But regardless of the orientation of the edge between $B$ and $C$, one cannot reverse $A \leftarrow B$ in the sequence as it cannot be an isolated edge.
   (c) $C \leftarrow A$, $C \leftarrow B$. Because $G_1$ and $G_2$ are in the same IEC. The first edge to be reversed must be $B \to A$. But then, we cannot reverse $B \to C$ because it is not covered and we cannot reverse $A \to C$ because it is covered but not isolated. So this case is also not possible.

B. $e_1$ is not a covered edge in $G_1$.

   By Lemma F.8, there must exist a node $C$ in $G_1$ such that $C \to B$ and $C \leftarrow A$. Note that in $G_2$, we have $A \leftarrow B$. By definition, there exists a sequence of isolated edge reversals from $G_2$ to $G_1$. There are only three valid possibilities (the fourth one is ignored because it creates circles) in $G_2$:

   (a) $C \to A$, $C \to B$. Both $C \to A$ and $A \leftarrow B$ have to be reversed. Out of all the three edges among $A, B, C$, the first edge that might be reversed in the sequence is $C \leftarrow B$. But regardless of the orientation of the edge between $B$ and $C$, one cannot reverse $A \leftarrow B$ in the sequence as it cannot be an isolated edge.
   (b) $C \leftarrow A$, $C \leftarrow B$. Because $G_1$ and $G_2$ are in the same IEC. The first edge to be reversed must be $B \to A$. But then, we cannot reverse $B \to C$ because it is not covered.
   (c) $C \to A$, $C \leftarrow B$. All the edges must be reversed. Because $G_1$ and $G_2$ are in the same IEC. The first edge to be reversed must be $B \to C$. But then, we cannot reverse $B \to A$ because it is covered but not isolated and we cannot reverse $C \to A$ because it is not covered. So this case is also not possible. $\quad\square$

## G   Edge orientations for measurement model

Up to this point, we have used unknown interventions to learn the skeleton of the underlying DAG $G$ (Appendix D, Appendix E). For causal interpretation, we must go one step further and orient these edges. Now, orienting the edges in the bipartite graph is easy: In a measurement model, the observed $X$ cannot have any children, so the only possibility is to orient the edges from hidden to observed. Orienting edges in the latent space is a different matter: Since the intervention targets are additionally unknown, this raises additional complications.

We extend Algorithm 1 to Algorithm 2 to orient edges in $G_H$ up to isolated edges which cannot be oriented using CI information only in general (Appendix F).

**Theorem 5.1.** *Let $G$ be a maximal measurement model satisfying Assumption 1 and assume we are given a complete family of interventions (Assumption 2) as well as the bipartite DAG $G_B$. Then the true latent DAG $G_H$ is identifiable up to isolated edges. Moreover, isolated edges cannot be oriented without making additional assumptions.*

*Proof.* The identification is given by Algorithm 2.

Suppose the returned PDAG is $G_H^\#$. First, by Theorem E.1, $G_H^\#$ and $G_H$ share the same skeleton. By Lemma G.2, all the unoriented edges are the reversible edges of the Markov equivalence class $\mathcal{E}(G_H)$. If a reversible edge $H_1 \to H_2$ is not covered or covered but non-isolated in $G_H$, then $H_2$ has at least two incoming edges by definition and Lemma F.8. Then $H_1 \to H_2$ will be correctly oriented because $\mathcal{M}_H(G^{(H_2)})$ has at least two members. Therefore, the only reversible edge that cannot be oriented is the isolated one.

By Lemma F.9, the only reversible edges in the IEC of $G_H$ are isolated in $G_H$. And by Theorem 5.3, we cannot distinguish DAGs in IEC with unknown single-node hard interventions. $\square$

## G.1 Useful Lemmas

**Lemma G.1.** *Let $G$ be a DAG and suppose $X \to Y$ is a covered edge in $G$, then $X \to Y$ is not compelled in $\mathcal{E}(G)$.*

*Proof.* By Lemma 1 of [12], we can reverse this edge to get $G'$ and $G'$ is Markov equivalent to $G$. By definition of compelled edge, it is not a compelled edge. $\square$

**Lemma G.2.** *Every unoriented edge returned by Algorithm 2 is a reversible edge of the Markov equivalence class $\mathcal{E}(G_H)$.*

*Proof.* Recall that an edge is reversible if there exist two members of the equivalence class that have two different orientations of this edge.

Without loss of generality, suppose one of the unoriented edges returned by Algorithm 2 is $H_1 \leftarrow H_2$ in $G_H$. Let's consider the following cases:

A. In $G_H$, $H_1$ has incoming edges other than $H_1 \leftarrow H_2$.

   In this case, when the unknown intervention target $I$ is $\{H_1\}$, there are at least two elements in $\mathcal{M}_H(G^{(I)})$ after step 1 of the algorithm. By Lemma E.6, we can thus orient the edge $H_1 \leftarrow H_2$ in step 2.

B. In $G_H$, $H_1$ has no incoming edge other than $H_1 \leftarrow H_2$ and $H_2$ has at least two incoming edges.

   In this case, when the unknown intervention target $I$ is $H_2$, there are at least two elements in $\mathcal{M}_H(G^{(I)})$ after step 1 of the algorithm. By Lemma E.6, we can thus orient the incoming edge of $H_2$ in step 2 and orient $H_1 \leftarrow H_2$ in step 3.

C. In $G_H$, $H_1$ has no incoming edge other than $H_1 \leftarrow H_2$ and $H_2$ has only one incoming edge.

   Suppose the other edge is $H_2 \leftarrow H_3$ in $G_H$. Then this edge must also be unoriented the Algorithm 2. Because if this edge is oriented, then $H_2$ must be in $\widehat{\mathcal{I}}$ by step 3 which implies that $H_1 \leftarrow H_2$ must also be oriented as well. Therefore, we can reverse $H_1 \leftarrow H_2$ and $H_2 \leftarrow H_3$. Because $H_2 \leftarrow H_3$ is unoriented, $H_3$ must have at most one incoming edge since the other cases have been ruled out by the previous argument. If it has one incoming edge, we can reserve that edge as well. The process continues until we reach one node that does not have any incoming edge. The whole process does not create any new $v$-structures. So we have two Markov equivalent graphs with different orientations of $H_1 \leftarrow H_2$.

D. In $G_H$, $H_1$ has no incoming edge other than $H_1 \leftarrow H_2$ and $H_2$ has no incoming edge.

Consider $G'_H$ where $G'_H$ has the same edges as $G_H$ except that $G_H$ has $H_1 \rightarrow H_2$. Because no $v$-structure is created or destroyed with this edge reversal, $G'_H$ and $G_H$ are Markov equivalent.

If an edge is unoriented, it must fall in the last two cases and be reversible. $\qquad\square$

**Algorithm 2** Learning Skeleton and Orienting Edges

---

**Input:** $H, \{\mathcal{M}_H(G^{(I)}))\}_{I \in \mathcal{I}}$

  9  $E \leftarrow \emptyset$ ;                        `// Set of directed and undirected edges`

10  $\widehat{\mathcal{I}} \leftarrow \emptyset$ ;                             `// Set of intervention targets`

    `/* Step 0`                                                      `*/`

11  **if** $\{\mathcal{M}_H(G^{(I)}))\}_{I \in \mathcal{I}}$ *only has one distinct element* **then**

      $\lfloor$ **Output:** $E$

    `/* Step 1:  Remove any pair of hidden variables that appears twice`      `*/`

12  **for** $\mathcal{M}_H(G^{(I)})) \in \{\mathcal{M}_H(G^{(I)}))\}_{I \in \mathcal{I}}$ **do**

13      **for** $\langle H_1, H_2 \rangle \in \mathcal{M}_H(G^{(I)}))$ **do**

14          If $\langle H_1, H_2 \rangle$ appears twice with at least two different intervention targets, delete $\langle H_1, H_2 \rangle$ from $\{\mathcal{M}_H(G^{(I)}))\}_{I \in \mathcal{I}}$.

15  Remove $\mathcal{M}_H(G^{(I)})$ from $\{\mathcal{M}_H(G^{(I)})\}_{I \in \mathcal{I}}$ if $\mathcal{M}_H(G^{(I)})$ has no element.

    `/* Step 2:  Add edges for colliders`                              `*/`

16  **for** $\mathcal{M}_H(G^{(I)}) \in \{\mathcal{M}_H(G^{(I)})\}_{I \in \mathcal{I}}$ **do**

17      **if** $|\mathcal{M}_H(G^{(I)})| > 1$ **then**

18          Find the common hidden variables $H_*$ shared by all tuples in $\mathcal{M}_H(G^{(I)})$

19          $\widehat{\mathcal{I}} \leftarrow \widehat{\mathcal{I}} \cup \{H_*\}$

20          **for** $\langle H_1, H_2 \rangle \in \mathcal{M}_H(G^{(I)}))$ **do**

21              **if** $H_1 = H_*$ **then**

22                 $\lfloor$ $E \leftarrow E \cup \{H_2 \rightarrow H_1\}$

23              **else**

24                 $\lfloor$ $E \leftarrow E \cup \{H_1 \rightarrow H_2\}$

25          Remove $\mathcal{M}_H(G^{(I)})$ from $\{\mathcal{M}_H(G^{(I)})\}_{I \in \mathcal{I}}$

    `/* Step 3:  Add compelled edges`                                 `*/`

26  $NewInv \leftarrow True$ ;  `// Flag to check if new intervention targets are added to` $\widehat{\mathcal{I}}$

27  **while** *NewInv is True* **do**

28      $NewInv \leftarrow False$

29      **for** $\mathcal{M}_H(G^{(I)})) \in \{\mathcal{M}_H(G^{(I)}))\}_{I \in \mathcal{I}}$ **do**

30          Let the only element in $\mathcal{M}_H(G^{(I)})$ be $\langle H_1, H_2 \rangle$

31          **if** $H_1 \in \widehat{\mathcal{I}}$ **then**

32              $E \leftarrow E \cup \{H_1 \rightarrow H_2\}$

33              $\widehat{\mathcal{I}} \leftarrow \widehat{\mathcal{I}} \cup \{H_2\}$

34              $NewInv \leftarrow True$

35              Remove $\mathcal{M}_H(G^{(I)})$ from $\{\mathcal{M}_H(G^{(I)})\}_{I \in \mathcal{I}}$

36          **else if** $H_2 \in \widehat{\mathcal{I}}$ **then**

37              $E \leftarrow E \cup \{H_2 \rightarrow H_1\}$

38              $\widehat{\mathcal{I}} \leftarrow \widehat{\mathcal{I}} \cup \{H_1\}$

39              $NewInv \leftarrow True$

40              Remove $\mathcal{M}_H(G^{(I)})$ from $\{\mathcal{M}_H(G^{(I)})\}_{I \in \mathcal{I}}$

    `/* Step 4:  Add unoriented edges`                              `*/`

41  **for** $\mathcal{M}_H(G^{(I)})) \in \{\mathcal{M}_H(G^{(I)}))\}_{I \in \mathcal{I}}$ **do**

42      Let the only element in $\mathcal{M}_H(G^{(I)})$ be $\langle H_1, H_2 \rangle$

43      $E \leftarrow E \cup \{H_1 - H_2\}$

    **Output:** $G_H^{\#} = (H, E)$

---

