# OpenReview forum: "Learning Nonparametric Latent Causal Graphs with Unknown Interventions"
_NeurIPS.cc/2023/Conference — NeurIPS 2023 poster_

### Official Review · Reviewer_Yp1n · 2023-07-03

**Soundness:** 3 good
**Presentation:** 3 good
**Contribution:** 3 good
**Rating:** 6
**Confidence:** 4

**Summary:**

The paper studies the problem of recovering causal relationships under the measurement model where there are latents but no direct causal edges between observed covariates. The authors introduced two graphical concepts -- imaginary subsets and isolated edges -- and show how they relate to sufficient conditions for recovery (under some additional assumptions). The assumptions are discussed at length in the appendix and a two phased recovery algorithm is proposed.

**Strengths:**

The paper is well-written and easy to follow in general. I also appreciate the in-depth discussion of the assumptions.

**Weaknesses:**

Assumption 1(d) seems redundant and I think it is implied by assumption 1(c); rewriting it in terms of a lemma or consequence of assumption 1(c) would strengthen the paper and reduce the number of assumptions required. Consider the following argument: Fix any two latents $H_i$ and $H_j$. By assumption 1(c), $H_i$ has a child $X_i$ that is not a child of $H_j$, and $H_j$ has a child $X_j$ which is not a child of $H_i$. We now consider the contrapositive of 1(d). Suppose $X_i$ and $X_j$ are d-connected. In the measurement model, this means that there is a path $X_i \gets H_i - \ldots - H_j \to X_j$ which has no colliders. Then, this same path is a witness to $H_i$ and $H_j$ being d-connected.

I am unsure how interesting Definition 5.2 on Isolated equivalence class (IEC) is. I do not think it is fair to compare its significance to Chickering's "covered edge reversal" characterization, which I believe is significantly more subtle and interesting. For instance, while there exists a sequence of covered edge reversals (say edge $e_1$, then $e_2$, then $e_3$, ..., then $e_r$) between any two DAGs in the same Markov equivalence class (MEC), the edges may actually NOT be covered edges midway through the transformation and one cannot arbitrarily reverse the set of edges $\{e_1, \ldots, e_r\}$ in any ordering whilst ensuring that we always get a DAG from the same MEC. Chickering further gives a constructive algorithm which tells us how to find this sequence $(e_1, \ldots, e_r)$. In contrast, IEC seems trivial since it involves a union of disjoint edges, where the size of IEC is always 2^(number of isolated edges) and every edge can be reversed at any point in time.

Experimental details are lacking: Section 6 is short and there is nothing in the appendix about the experiments. It is hard to judge or appreciate any empirical contribution. I feel that the authors should have just focused on presenting this work as a theoretical contribution (which I think is already sufficient on its own, modulo the questions below).

**Questions:**

Caption for Figure 1:
From my understanding of Definition 3.2, imaginary subsets have to be maximally valid. Why is $\{X_5, X_6\}$ maximal?

Suggestion for Figure 1:
In the additional page that comes with paper acceptance (if this gets accepted), it would be nice to include a picture of D(P), mention that $\Omega_P = \\{ \\{1,2,5,6\\}, \\{3,4,5,6\\} \\}$, and then refer to Fig 1 on Lines 146 and 200.

Missing assumption:
This work implicitly assumes infinite samples / population regime / access to a d-separation oracle, right? Please state this explicitly.

Assumption 2:
This assumption feels very strong... For example, such an assumption trivially solves the causal graph discovery problem in the causal sufficient setting (w/o latents but observed covariates have edges amongst themslves) if we have access to all $n$ interventional essential graphs. I understand that the model studied here has latents, but then the measurement model seems to also simplify things a lot. Why does Assumption 2 not immediately trivialize the entire (or part of the) recovery objective? For example, if there is no repeated interventional distributions (i.e. $|\mathcal{I}| = m+1$), then isn't Line 223 trivial? I understand the discussion of this assumption in the appendix (Line 651 should be emphasized in the main text), but it seems that, in the worst case, we just "lump" all downstream latents together. What am I missing? What is the subtlety that I am not getting?

Line 184, footnote 1:
Do you mean "In other words, removing any more..." instead of "In other words, adding any more..."? We can always add redundant edges while maintaining the underlying model, right? e.g. a clique can encode any arbitrary distribution, including the product distribution.

Theorem 3.4:
Is there a "necessary" counterpart to this result? For instance, I was under the impression that existence of imaginary subsets makes G unidentifiable. If that is the case, you should perhaps write something like "G is identifiable if and only if no imaginary subsets" for (a). Characterizations like these would greatly strengthen the paper's contribution.

Line 250, and also appendix E; Misconception about maximality:
I think I have a misconception about maximality, which is affecting my understanding of the paper's correctness. Why is $\{X_1, X_2\}$ a maximal valid subset if $\{X_1, X_2, X_5\}$ is one? Doesn't the fact that the former subset being a proper subset of the latter make it *not* maximal? Could you kindly resolve my misconception? Thanks! (I will revise the soundness score and overall rating accordingly.)

Line 377:
I am unsure what the last sentence is trying to imply. Could you clarify? I have the following guesses (all of which could be wrong):
- Are the theoretical assumptions unnecessary?
- Did you just "get lucky" with the experiments?
- Are you suggesting that the assumptions are not required for the class of models which you have ran experiments on?

Table 1:
How are the errors split across $G_B$ and $G_H$? From my understanding, the algorithm to recover $G_H$ crucially depends on $G_B$ being correctly recovered, right? It is unclear to me what we can conclude if $G_B$ was recovered with errors and then subsequently used to recover $G_H$ --- how do the errors propagate? Can you say something about it in theory?

Figure 2 and 3:
Isn't Figure 3 just Figure 2?

Example 12:
Why does Figure 1 satisfy assumption 3? $H_1$ and $H_2$ have no pure child.

**Limitations:**

Nil.

---

> ### Author Rebuttal · Authors · 2023-08-10
>
> Thanks for the questions and suggestions!
>
> **Assumption 1(d)**
>
> 1(d) is not implied by 1(c). In your example, $X_i$ and $X_j$ could be connected via other hidden variables. Even when $H_i$ and $H_j$ are disconnected,  $X_i$ and $X_j$ might still be connected. We give such an example in Appendix C.3. In particular, consider Fig 5, $X_1$ is not a child of $H_2$ and $X_3$ is not a child of $H_1$. However, under the intervention target $H_2$, $X_1$ and $X_3$ are still d-connected via parent $H_3$.
>
> **IEC**
>
> Apologies for any confusion. There seems to be a potential misunderstanding regarding the concept of isolated edges. Despite what the name might suggest, an isolated edge X->Y does not mean that X, Y are disconnected from all other nodes. In fact, X and Y can still have outcoming edges (definition 3.3) and X->Y is not just an isolated connected component. So IEC doesn't just involve a union of disjoint edges. For example, consider a graph of three variables: A, B, C with two edges A->B, A->C. These two edges are both isolated edges but not disjoint. In fact, one can reverse edge A->B to be B->A. In that case, A->C is no longer an isolated edge. Therefore, similar to MEC where the edges may actually NOT be covered edges midway through the transformation, isolated edges can have the same problem.
>
> We agree that the characterization of isolated edges is simpler than covered edges because isolated edges are special cases of covered edges. On the other hand, the introduction of IEC is used merely as a tool to characterize the unorientability of isolated edges (theorem 5.3 and G.7) and is not intended to suggest that this is deeper or more fundamental than the MEC.
>
> **“Experimental details”**
>
> We will provide more details on experiments in the updated version. Some additional details include that the weights are generated by uniformly sampling from $[-2, -0.5] \cup [0.5, 2]$ and the variances are set to 1. More details can also be found in the codebase we included in the supplementary files.
>
> **“Misconception about maximality”**
>
> Again, we apologize for any confusion: The definition of a maximal valid subset clearly states that X’ and X’’ must be contained in the *same* clique in the UDG. A maximal valid subset X’ is maximal in the sense that for any clique containing X’ there does not exist X’’ that’s also in the same clique and is a superset of X’. In other words, X’ is maximal if its existence cannot be completely explained by another subset. We apologize for the confusion, however, upon inspecting our definitions, everything is correct as stated. We will add a clarification of this point in the camera ready. See below as well for a correction to Figure 1.
>
> **Why is X_5, X_6 maximal?**
>
> Thanks for pointing this out! In fact, as reported in Figure 1, {$X_5, X_6$} is not maximal. Thanks to your careful attention, we realized there is a mistake in Figure 1 that needs to be corrected. The original Figure 1 was simplified for the purpose of providing a clear and easily understandable demonstration, but in the process of simplifying the DAG, the maximal valid subsets became slightly different than what the caption states.
>
> In the attached pdf file, we present a modification to Fig 1 that corrects the issue and preserves all the statements about maximal valid subsets and imaginary subsets. We have also taken your advice to include UDGs under each intervention.
>
> To demonstrate why both {$X_1, X_2$} and {$X_1, X_2, X_5$} can be maximal valid subsets, let’s refer to the updated figures in the pdf file. Figure 2(c) in the pdf file shows that under intervention target $H_4$, there are three maximal cliques: {$X_1, X_2, X_4$}, {$X_1, X_2, X_5$}, {$X_3, X_5, X_6$}. Because {$X_1, X_2, X_4$} contains {$X_1, X_2$} but not {$X_1, X_2, X_5$}, these two sets can both be maximal without contradicting the definition.
>
>
> **Missing assumption**
>
> This is stated at L49: “Given a set of interventional distributions”, as opposed to samples. Of course, the population regime is equivalent to assuming distributions as input, which is standard in the literature when discussing identifiability. Since the goal of this paper revolves around identifiability, we leave estimation as an intriguing future direction. We’ll make these points clear in the updated version.
>
> **Assumption 2**
>
> This is a common assumption and in fact, it is an open question whether or not this can be relaxed [a-b]. [a] also shows that this assumption is necessary (Section 3.3 of [a]).
>
> Moreover, compared to classical work on interventions in graphical models, the current literature on causal representation learning (including our submission as well as [a,b]) considers a strictly harder setting since the interventions are both latent and unknown. Since the intervention target is latent, we only have access to partial information (observed variables) regarding the effect of interventions. And since the intervention target is unknown, compared to the known intervention target, we additionally have the unorientability problem of isolated edges which is discussed in section 5.2 (L357).
>
> [a] Seigal, Squires, Uhler. "Linear causal disentanglement via interventions." arXiv preprint arXiv:2211.16467 (2022).
>
> [b] Varici, Burak, et al. "Score-based causal representation learning with interventions." arXiv preprint arXiv:2301.08230 (2023).
>
> **Line 377**
>
> Please see the global review.
>
> **Line 184, footnote 1**
>
> Sorry for the confusion. What we meant is that adding edges until we cannot add more without changing (i.e. adding _or_ removing) the conditional independence statements. This notion of maximality is the same as in maximal ancestral graphs (MAGs), as mentioned on L180. See [c] for more details on MAGs, which are standard graphical models for incorporating latent variables. We have double-checked our definition and can confirm it is correct as stated.
>
> [c] Richardson and Spirtes. "Ancestral graph Markov models." The Annals of Statistics (2002).

---

> > ### Comment · Reviewer_Yp1n · 2023-08-11
> >
> > Thank you for your patience and efforts to clearing my doubts and misunderstandings. Also, thanks for sharing about the "pure child assumption" is similar to the notion of "separability assumption" in NLP (I don't work on NLP problems and this is the first time I learnt about this!)
> >
> > **Assumption 1(d)**
> >
> > You are right. Thank you for clarifying my misunderstanding.
> >
> > **IEC**
> >
> > Thank you for clarifying my misunderstanding.
> >
> > **Maximal clique**
> >
> > Thank you for explaining why both {X1, X2} and {X1, X2, X5} can be both maximal. My confusion was not that it must be the *same* clique, but that it must be *any clique*, under *any intervention*. The pictures in the attached PDF were very helpful for me in clearing this misconception.
> >
> > Now that I understand what you mean by maximal clique, do you mean to have $X' \subseteq X''$ instead of $X' \subsetneq X''$ in the Definition 4.2 in your submission? Otherwise, you are saying {X1, X2} $\subsetneq$ {X1, X2, X5}, which confuses me again...
> >
> > **Missing assumption**
> >
> > Thanks. Please make it explicit and clear to other readers. It is indeed an intriguing future direction.
> >
> > **Assumption 2**
> >
> > Thank you for sharing the references. These two works indeed use the similar assumption. It is also interesting that [a] has a worst-case necessity for that assumption, though they study a slightly different setting from what you study (maybe you can give some explanation why their setting is a special case of yours?). Please include some discussion about this in the paragraph above Section 3.2, where you discussed the other assumptions. I think it will benefit the other readers. Thanks!
> >
> > **Line 184, footnote 1**
> >
> > Sorry, I still don't get it... I know about a bit about ancestral graphs, though I'm not an expert on it. My understanding of causal graphs is that the *absence* of edges encode assumptions about independencies in the model. For example, [c] states that "a graph is maximal if every missing edge corresponds to at least one independence in the corresponding independence model". The *presence* of an edge itself doesn't say much: a fully connected clique is always a valid consistent causal graph but it yields no useful information. We are talking about the same thing right...?
> >
> > **References**
> >
> > [a] Squires, Seigal, Bhate, Uhler. "Linear causal disentanglement via interventions." ICML (2023).
> >
> > (I was checking your references in the rebuttal and noticed that the author list is slightly different. Also, I think you should cite the conference version instead of the arXiv one; see https://openreview.net/pdf?id=1VDuHddxtA)
> >
> > [c] Richardson and Spirtes. "Ancestral graph Markov models." The Annals of Statistics (2002).

---

> > > ### Author Response · Authors · 2023-08-12
> > >
> > > Thanks for the quick reply!
> > >
> > > **Maximal clique**
> > >
> > > Sorry about the confusion. The notation $\subsetneq$ means “proper subset”, i.e.if $A \subsetneq B$, then $A$ is a subset of $B$ but not equal to $B$. This is not to be confused with $\not\subset$. Since there is room for confusion here, we will clarify this in the final version.
> > >
> > > **Assumption 2**
> > >
> > > You’re right that [a] studies a slightly different setting, although the high-level goal of identifying latents under unknown interventions in a measurement model is the same. To clarify, [a] shows the necessity of Assumption 2 under their assumptions, which are slightly different from our setting. We have independently shown that this assumption is needed in our setting with Example 6 in Appendix C.4.
> > >
> > > There are three main differences with [a]: (1) They study linear functions while we focus on nonparametric identification; (2) They allow the bipartite graph between latents and observed to be fully connected while we have graphical constraints (Assumptions 1(c) and (d)); (3) They consider noiseless transformations between latents and observed while we allow noisy transformations. Though we briefly touch on similar assumptions in L747 in Appendix C.4, We’ll clarify this further in the paper. Thanks for the suggestion!
> > >
> > > **Line 184, footnote 1**
> > >
> > > Your intuition is right, however, the situation is more nuanced with latent variables. In Example 7 in Appendix D, we construct two DAGs $G_{(a)}$ and $G_{(b)}$, and two models $P_{(a)}$ and $P_{(b)}$, that generate identical d-separation and CI relations over X. But they differ over (X,H): $P_{(a)}$ satisfies $H_1 \perp H_3 | {H_2, H_4}$, whereas $P_{(b)}$ does not. (This is clear from the extra edge $H_1\to H_3$ in $G_{(b)}$.)
> > >
> > > [We realize now that this point was never made explicit, and we will definitely revise this example and the discussion of maximality to reflect this discussion. We’d like to thank you for surfacing this confusion so it can be properly addressed in the final version.]
> > >
> > > So, since these models cannot be distinguished on the basis of the observed data P(X), what should we do? We argue that we should only remove an edge if its removal can be justified on the basis of what we actually observe, i.e. the data X. Although $P_{(a)}$ and $P_{(b)}$ can (in principle) be distinguished, we need to observe H to do so, which we cannot do in practice.
> > >
> > > More generally, here is what is happening: In general, of course, there are multiple DAGs that are Markov to a given distribution, and the question is how do we decide on the correct “minimal” representation. Without latents, there is no ambiguity: We can always test all possible CI relations and obtain a complete picture to obtain a minimal I-map. With latents, we must be careful:
> > > - Of course, if we can check CI relations over all of (X,H), then the usual notion of a minimal I-map prevails. But in practice, we cannot access P(X,H) since H is unobserved.
> > > - Thus, in practice, we should restrict our attention to information about P(X) only. In this case, we argue that we should only remove an edge if its removal can be justified on the basis of information about P(X) _only_. This is the essence of maximality: We only remove an edge if it follows from the observed data X. Otherwise, we remain agnostic: We do not want to remove an edge that may in fact reflect a “real” dependence over H.
> > >
> > > This is the essence of maximality, and the intuition is the same as for maximal ancestral graphs.
> > > As a result, our characterization of the maximal measurement model aligns with the essence of the maximal ancestral graph. Measurement models have latent variables and we only have access to partial information (ie., observed variables). Since two measurement models can encode the same set of conditional independencies over X and as you have pointed out, the absence of edges encodes nontrivial information, the removal of an edge should be justified carefully on the data we have available.
> > >
> > > **References**
> > >
> > > Thanks for pointing that out! We didn’t realize the author list has changed since we drafted the paper. We will update it accordingly.

---

> > > > ### Comment · Reviewer_Yp1n · 2023-08-12
> > > >
> > > > Thank you very much for your patience and clarifying my concerns. I have updated my scores accordingly :)

---

> > > > > ### Author Response · Authors · 2023-08-13
> > > > > **Thanks!**
> > > > >
> > > > > We are happy to do so, and pleased your concerns have been addressed! We sincerely appreciate your time and effort to understand our work, and for updating your score accordingly.

---

### Official Review · Reviewer_tfcU · 2023-07-06

**Soundness:** 3 good
**Presentation:** 4 excellent
**Contribution:** 2 fair
**Rating:** 5
**Confidence:** 3

**Summary:**

- The paper studies causal representation learning, or more precisely the identification of the causal graph between observed and latent variables, from interventional data with unknown intervention targets.
- Its main contribution is an identifiability result for the causal graph. This theorem makes no assumptions on the functional form of the causal model or the mixing function, but is based on several graphical requirements. In various ways, the paper requires that different latents affect different sets of observed variables.
- The authors spend a large part of the paper discussing these assumptions and providing sufficient conditions for them.
- In the end, they also briefly demonstrate their algorithm on toy data.
- Unlike most of the CRL literature, the paper does not study the identification of the latent *variables*. The authors delegate this task to "existing work, since one can [...] use deep latent-variable-models to infer the latent distributions from the latent structure".

I have read the author's rebuttal. They have addressed my questions clearly.

**Strengths:**

- It is great that the authors can prove identifiability from observational, unlabelled data, and without functional assumptions. This makes the results potentially quite practical, barring limitations from the graphical assumptions (see below).
- While I have not been able to check the proof in detail and I do have some questions (see below), I overall believe that the key results are correct.
- The paper is very thorough, with precise statements, extensive discussion, useful examples, and thorough appendices. There is a lot in here that may be useful beyond the concrete identifiability result.
- It is also extremely well-written, really a joy to read. I appreciate the frequent signposting. Great job!

**Weaknesses:**

- Different from virtually all other CRL works, the authors choose to focus *only* on the identifiability of the causal structure and entirely disregard the identification of the causal variables.
	- Arguably, in most applications of CRL, the latent variables are at least as important as a result.
	- The strategy of solving the structure learning problem first and delegating the variable identification to a latent-variable model makes sense, but deserves a discussion that goes beyond the two lines that the authors have reserved for it.
	- Could you perhaps quote or sketch what kind of guarantees on the identification of the latent variables one can expect when following such a two-step procedure?
- As with most of the CRL literature, a key question is whether the assumptions are too unrealistic or too difficult to verify to make the results useful beyond pure academic curiosity. I am particularly worried that assumptions 1(c), 1(d), and the lack of imaginary subsets do not apply to typical CRL settings.
	- For instance, I find it difficult to imagine these assumptions applying to any of the systems sketched in lines 18-20 in the introduction. Could the authors discuss this and provide perhaps some semi-realistic examples of systems that satisfy them?
	- Negative results are equally valuable though, and I appreciate the counter-examples that show the lack of identifiability when these assumptions are violated.
- I am also concerned about the restriction to maximal measurement models.
	- This seems to be a bit at odds with Okham's razor: if multiple models explain the data, why should we focus on the most complex model that explains the data? Should we not identify the family of all models, or the simplest model?
	- It would be great if the authors could comment on how strong this assumption is and in what kind of systems they expect it to be satisfied.
- The experiments are very limited and really just a minimal proof of concept.
	- It would make the paper stronger if the authors would design experiment that test whether the approach scales to interesting problems and that test how robust the approach is to violations of the assumptions.
	- In addition, a comparison to other methods (for instance on problems that satisfy  functional assumptions made by other papers) would be interesting.

**Questions:**

- In line 150, the authors stress that they consider the *set*, not the tuple, of interventional distributions. What motivates this choice?
- Is Assumption 1(b) the same as the common assumption of faithfulness?
- In Theorem 3.4, what does "using CI information only" mean exactly? Could we get stronger results when using the full distribution(s), not just the conditional independence patterns?
- In Theorem 3.4, what does "G is identifiable" mean exactly? Identification up to a graph isomorphism (so for instance allowing for a permutation of the latent variable)?
- In line 227, what is meant by "sequel"?
- If we had a dataset with multi-target interventions in addition to the single-target interventions, would that allow for stronger statements? Perhaps we could relax the requirement of not having imaginary subsets?
- I'm a bit confused why we can have identifiability of the latent graph except for the direction of isolated edges $a \to b$ . It seems that any other isolated chain, like $a \to b \to c$ disconnected from all other latents, would be similar. Could the authors provide some intuition for why this latter graph can be identified, while the isolated edge cannot?
- Is the interventional MEC a subset of the IEC?
- In line 372, what are m and n?

**Limitations:**

- The authors are very clear about the assumptions of their theoretical results.
- Nevertheless, I believe it deserves more discussion whether these assumptions fit realistic problems (see above).

---

> ### Author Rebuttal · Authors · 2023-08-10
>
> Thank you for the detailed comments! We're glad you find the paper a joy to read.
>
> **“Focus only on the causal structure”**
>
> We completely agree that learning latent distributions is important! At the same time, without causal structure, latent distributions may not be interpreted causally, and therefore we suggest that gaining an understanding of latent causal structures is of equal importance. In some sense, learning the structure is necessary: To understand causal relations, one needs to know what happens when we intervene on these learned features. Therefore, this paper studies the equivalently important problem of structure learning and we believe that it lays the foundation for future research on CRL.
>
> In this sense, since most related work focuses on the latent distribution, we are studying a complementary but equally important aspect: What are the minimal conditions needed to recover the latent causal structure? We believe that understanding these two problems (structure vs representations) separately is crucial to understanding what assumptions matter, why they matter, and for which aspects of the problem they are needed.
>
> **“The assumptions are too unrealistic or too difficult to verify”**
>
> Please see the global response.
>
> **“maximal measurement models”**
>
> Maximality helps uniquely identify latent causal graphs as in many cases, there could be multiple measurement models that can explain the observed interventional distributions but only one maximal measurement model (Appendix D). Such construction is similar to the definition of maximal ancestral graphs (MAGs, see L180-181). See [c] for more details on MAGs, which are standard graphical models for incorporating latent variables. Intuitively, a conditional independence statement could be explained by multiple combinations of missing edges. We choose the simplest explanation with the least amount of missing edges.
>
> [c] Richardson and Spirtes. "Ancestral graph Markov models." The Annals of Statistics (2002).
>
> **“The experiments are very limited”**
>
> Please see the global response.
>
> **“set vs tuple”**
>
> This is a good observation! Because we allow interventional distributions to be the same under different interventions and the interventional targets are unknown, sampling from the same interventional distributions could be viewed as the same. Thus, we consider sets instead of tuples since this allows for a more realistic setting where different interventions are indistinguishable. Moreover, learning from a tuple is easier than learning from a set, since one can reduce the tuple problem to set problem because one can create a set from a tuple by removing duplicates. Thus, there is no loss of generality in our setting.
>
> **“Assumption 1(b) and faithfulness?”**
>
> Assumption 1(b) is substantially weaker than the usual faithfulness assumption (L166-168 and detailed discussion in appendix C.1). Here, we only consider pairwise, marginal independencies between observed variables. Ordinary faithfulness includes full conditional independencies of arbitrary order including latent variables.
>
> **“In Theorem 3.4, what does "using CI information only" mean exactly?”**
>
> Yes, one could use more information about the distributions to potentially orient more isolated edges, and this is a natural direction for future work. Our results imply that this will require additional assumptions. One example is direct I-faithfulness [c], which uses distributional info (L368).
>
> [c] Squires, Chandler, Yuhao Wang, and Caroline Uhler. "Permutation-based causal structure learning with unknown intervention targets." Conference on Uncertainty in Artificial Intelligence. PMLR, 2020.
>
> **Confusion about isolated edges a->b**
>
> Your intuition is right. a->b->c cannot be oriented either. Note that a->b is an isolated edge. This is because the only parent of b is a and c is a child of b (Definition 3.3). Thus one can reverse it such that we have a<-b->c. Now, b->c is an isolated edge, one can reverse it again to get a<-b<-c. Thus a->b->c and a<-b<-c are in the same IEC because there exists a sequence of isolated edge reversals between them. Theorem 5.3 shows that these two graphs are indistinguishable using CI information only.
>
> We apologize for the confusion and feel like there is a potential misunderstanding due to the naming of “isolated edges”. An isolated edge X->Y does not mean that X, Y are disconnected from all other nodes. In fact, X and Y can still have outcoming edges and X->Y is not just an isolated connected component. For instance, in your example, a->b is an isolated edge but is connected to other edges as well.
>
>
> **“In Theorem 3.4, what does "G is identifiable" mean exactly? Identification up to a graph isomorphism (so for instance allowing for a permutation of the latent variable)?”**
>
> This is correct. Up to different labeling of the latent variables (graph isomorphism). But such reordering is trivial because of assumption 1(c). Different latent variables will have different sets of observed children. Thus one can use the children set to uniquely identify different latent variables.
>
>
> **“If we had a dataset with multi-target interventions in addition to the single-target interventions, would that allow for stronger statements? Perhaps we could relax the requirement of not having imaginary subsets?”**
>
> This is a good question! Extending our theory to multi-target interventions is an exciting direction. Having access to multi-target interventions would definitely help relax assumption 2 and maybe allow for stronger results.
>
> **“Is the interventional MEC a subset of the IEC?”**
>
> Interventional MEC is a bit different from IEC. IEC is for unknown interventions and interventional MEC is for known interventions.
>
>
> **“In line 372, what are m and n?”**
>
> Sorry for the confusion. We define m and n as the number of latents and observed respectively on line 103. We will clarify it again in the experiment section.

---

> > ### Comment · Reviewer_tfcU · 2023-08-10
> >
> > Thank you for the rebuttal. You have answered my questions thoroughly and clearly. At the moment, I have no follow-up questions, though I will think about your work more in the next days.

---

> > > ### Author Response · Authors · 2023-08-13
> > >
> > > We're pleased to learn that your concerns have been resolved. Feel free to reach out with any future questions you may have.

---

### Official Review · Reviewer_bZuh · 2023-07-11

**Soundness:** 3 good
**Presentation:** 2 fair
**Contribution:** 2 fair
**Rating:** 4
**Confidence:** 2

**Summary:**

this paper aims the learn the causal structure in the latent space, by using interventional data, where the intervention targets are unknown, but with certain restrictions. This paper's focus is to provide some theoretical analysis, to show that, under what assumptions, we can recover (up to what level) the causal structure (include the bipartite DAG between observable variables and latent variables, and the DAG within the latent variables).

**Strengths:**

1 - the paper deals with a very challenging, or even ill-posed problem, about recovering the causal structure in latent space.
2 - the theoretical part seems to be sound.
I have checked the illustration and the proofs in appendix C, the idea of the necessity of the assumptions in (1) is well presented
3 - recovering latent causal structure has great potential for AI or AGI area.

**Weaknesses:**

The most concern from my side is that, what this paper is telling lacks real-world relevance. this makes me doubt if this work can be helpful for practical usage. It seems to be more like a analytical deduction in that, given what assumptions, what I can achieve. but whether these assumptions are relevant in real-world is unclear. Some details
1) Can this work be evaluated from "causal representation learning" perspective?
for example, image pixels are generated from some latent concepts/entities, which seems to be quite aligned with the motivation of this work, can this work be experimented on any such case to justify its real usefulness?
2) justification of the required assumptions
many assumptions used in this work are untestable, although the authors discussed some of them, and pointed out that they are not be ignored, which is fine. But without real-world relevance, how can I know in what situation should I apply the proposed algorithm? for example, a complete family of targets seems to be too strong, from the perspective of "causal discovery from interventional data".
3) Better re-structure the paper writing
Overall, the paper is very dense and lacks intuitions. Regarding identifying the latent variables (or at least, detecting the number of latent variables), one sentence in line 245 is very interesting: "Proposition 4.1 suggests that we assign a latent variable to each maximal valid subset", I think this has a potentially nice intuition about identifying latent variables by using dependencies among the observational variables. I suggest the authors to provide an overview, with intuitions about the key idea, rather than defer then to Section 4 and 5.
4) claim of Example 1 is improper.
even with unknown interventional target, and by intervening once, we can still recover the orientation between X1 and X2. you can further checking if there is marginal independence between X1 and "whether the intervention is performed", and X1 and "whether the intervention is performed", this information can further help you to identify the causal direction. you can check more details from [1].

[1] Mooij, J. M., Magliacane, S., & Claassen, T. (2020). Joint causal inference from multiple contexts. The Journal of Machine Learning Research, 21(1), 3919-4026.

The second concern is the weakness of experiment.
as the author pointed, "Compared to these existing works, our focus is on nonparametric models with unknown, hard, single-node interventions on the latent variables", I think this is a clear configuration, and we can certainly conducts comparison with other SOTAs (such as using parametric approach)


**Questions:**

1) can we give a definition of clique in main body?

2) in line 224~226, the paper said "In other words, G can be maximally identified in the sense that any edge in the latent space that isn’t oriented cannot be oriented from the given list of interventions using CI information only: Additional assumptions are needed (e.g. conditional invariances and direct I-faithfulness)."
I want to know, if the additional assumptions are included, how much further we can achieve? from my perspective, conditional invariance and direct I-faithfulness are fair assumptions, they basically say that "when you do intervention, the data distribution should be changed in a rational way; otherwise, intervention cannot introduce detective changes thus you can exploit nothing from interventional data"

---

> ### Author Rebuttal · Authors · 2023-08-10
>
> Thank you for your thorough review and suggestions!
>
> **“The most concern from my side is that what this paper is telling lacks real-world relevance. this makes me doubt if this work can be helpful for practical usage.”**
>
> For example, one practical application is topic models. In topic models, the pure child assumption is fairly common [a], which is strictly stronger than assumptions 1(c) and 1(d) (Remark 4.2). And we show in the paper (Section 4.3) how to identify causal graphs under the pure child assumption. Applying our results to image data is a very interesting future direction.
>
> [a] Arora, Sanjeev, et al. "A practical algorithm for topic modeling with provable guarantees." International conference on machine learning. PMLR, 2013.
>
> **“justification of the required assumptions many assumptions used in this work are untestable.”**
>
> This paper studies the theoretical limit of nonparametric identifiability of latent causal graphs, and in Appendix C we show that our assumptions are tight. We feel like understanding the capabilities and limits of these assumptions can build the foundation for future work on CRL. On the other hand, our empirical results (section 6) show that even when our assumptions are not enforced, we can still get approximate recovery with low error rate as long as graphs are generated with sparsity.
>
> Finally, the complete family of targets is needed to get exact graph recovery (section C.4). One can easily relax this assumption to get partial identification (i.e., lumping some latent variables together).
>
> **Better re-structure the paper writing**
>
> Thanks for the suggestion. You’re right that the intuition behind our proof to identify latent variables is to use dependencies among the observational variables. By examining what’s invariant and what’s changing, one can recover the latent graph.
>
> **Claim of Example 1 is improper. even with unknown interventional target, and by intervening once, we can still recover the orientation between X1 and X2. you can further checking if there is marginal independence between X1 and "whether the intervention is performed", and X1 and "whether the intervention is performed", this information can further help you to identify the causal direction. you can check more details from [1].**
>
> Typically, one needs additional information to identify the interventional target. For instance, [b] uses direct I-faithfulness. Once the interventional target is known, then one can solve the isolated edge orientation problem. For JCI, could you clarify what you do mean by “whether the invention is performed”? If you meant the observed context variables, then one still needs to have observed context variables that satisfy assumptions on how system variables and context variables interact.
>
> [b] Squires, Chandler, Yuhao Wang, and Caroline Uhler. "Permutation-based causal structure learning with unknown intervention targets." Conference on Uncertainty in Artificial Intelligence. PMLR, 2020.
>
>
> **“Comparison with other SOTAs”**
>
> Our paper studies the general setting that allows arbitrary nonlinear transformation between latents and observed and among latents. It is likely that other methods utilizing functional assumptions like linearity can perform better if the functional form is known. Since our paper is primarily theoretical, our experiments are just a proof of concept to demonstrate that even for nonlinear transformations one can still recover the measurement model with low error.
>
>
> **“Can we give a definition of clique in main body?”**
>
> Thanks for the suggestion. We’ll add the definition to the main body. A clique is a subgraph where every pair of distinct vertices are adjacent.
>
> **“I want to know, if the additional assumptions are included, how much further we can achieve?”**
>
> With additional assumption like direct I-faithfulness [b], one can identify isolated edges. In this paper, we want to study nonparametric identifiability with minimal assumptions. Even without assumptions like direct I-faithfulness, intervention **can** still introduce detective changes. This is why all the non-isolated edges can be oriented (Theorem G.7). Our results demonstrate how to orient non-isolated edges without making additional assumptions. Thus, additional assumptions are not totally necessary.

---

> > ### Comment · Area_Chair_XRXf · 2023-08-18
> >
> > Thank you for the rebuttal. Since the reviewer has not replied, I have been asked to respond. It seems to me that most of the reviewer's concerns are addressed, and I will take this into account unless they reply here further.
> > Best,
> > the AC

---

> > > ### Author Response · Authors · 2023-08-19
> > >
> > > Thanks for your reply! We will look forward to additional feedback from the reviewer and we really appreciate your commitment to the quality of the review.

---

### Official Review · Reviewer_3v5g · 2023-07-27

**Soundness:** 2 fair
**Presentation:** 3 good
**Contribution:** 3 good
**Rating:** 6
**Confidence:** 4

**Summary:**

The paper discusses the identification and reconstruction of latent causal graphs from unknown interventions in the latent space. The main focus is on uncovering the latent structure in a measurement model, where the dependence between observed variables is less significant than the dependence between latent representations without parametric assumptions.
The paper presents a characterization of the limits of edge orientations within the class of Directed Acyclic Graphs (DAGs) induced by unknown interventions. The paper concludes with an experimental evaluation that shows the recovery of the causal graph using structural Hamming distance as the error metric of the true vs. learned causal structure.

********** POST REBUTTAL *******

Thank you to the authors for their responses. I’m satisfied with the clarifications and increased my score

**Strengths:**

Creating models that identify the causal structure in scenarios where parametric assumptions are not applicable is an important problem in causal inference.

The proposed approach uses interesting and clever concepts to provide a novel perspective of latent causal graphs.

The proposed approach provides an intriguing perspective about the DAGs’ equivalence class associated to unknown interventions.

**Weaknesses:**

Some details of the paper could be further clarified such as why not used stablished concepts of Markov equivalence classes, etc. Or why not consider non-parametric learning of causal structure such as the one in Gao, et al. (2020) or Azadkia et al. (2021)

Another aspect to consider is the fact that the evaluation is done with two settings and 100 runs for only 4 combinations of M,N.

In general the paper main contributions seem interesting from a theoretical perspective and because of that a more thorough discussion could have improved the paper to compensate for the limited evaluation.

**Questions:**

Could you elaborate on why is needed to rely on the concept of imaginary subsets?

What was the main technical challenge to define the additional concepts of imaginary subsets and isolated edges?

**Limitations:**

The paper does not describe potential societal impacts.

---

> ### Author Rebuttal · Authors · 2023-08-10
>
> Thank you for your questions! We also agree that this problem is important!
>
> **“Some details of the paper could be further clarified such as why not used stablished concepts of Markov equivalence classes, etc. Or why not consider non-parametric learning of causal structure such as the one in Gao, et al. (2020) or Azadkia et al. (2021)”**
>
> Thanks for the question! These papers do not consider latent variables or the measurement model and hence are not applicable to our setting. Markov equivalence applies to observational data without interventions, thus with interventions, standard graphical concepts such as Markov equivalence are not as useful. (Note that the MEC is of course still valid, it just does not account for interventions.)
>
> **“Another aspect to consider is the fact that the evaluation is done with two settings and 100 runs for only 4 combinations of M,N. In general the paper main contributions seem interesting from a theoretical perspective and because of that a more thorough discussion could have improved the paper to compensate for the limited evaluation.”**
>
> Thanks for acknowledging our theoretical contributions! We have detailed and thorough discussion of the assumptions and limitations of our theory in the appendix. For instance, we discuss why our assumptions are necessary in appendixes C and D. We also discuss in detail the difficulty with identifying bipartite graphs and why dealing with imaginary subsets is nontrivial with many examples in appendix E. Currently, because our paper is primarily theoretical, the experiments are shown as a proof of concept. It would be an exciting future direction to extend our results to real-world datsets. Other methods that rely on functional assumptions could potentially surpass our approach when supplemented with extra functional information (such as linearity). Nonetheless, our theory holds greater generality as it pertains to a wide range of nonlinearities.
>
>
>
> **“Could you elaborate on why is needed to rely on the concept of imaginary subsets?”**
>
> Imaginary subsets arise because we allow arbitrary connections between the latent variables. This is explained very briefly at L196-200, and we are happy to include more detail on this as follows: It is possible that two observed nodes can stay d-connected under any interventions even when they do not share the same parent. The densely connected latent graph makes them appear as if they share the same parent. The existence of imaginary subsets complicates the identification of the bipartite graph (example 8 and example 9). In this paper, we show additional assumptions one can have to get rid of imaginary subsets (section 4.2) including one testable assumption (no fractured subset, corollary 4.7). We also show how to identify the bipartite graph even with the presence of imaginary subsets under pure child assumptions (section 4.3). These are explained in detail in both Section 4 and Appendix E.
>
>
> **“What was the main technical challenge to define the additional concepts of imaginary subsets and isolated edges?”.**
>
> The main technical challenge with defining imaginary subsets is finding a concise way to encapsulate the difficulties with learning bipartite graphs and define them in a way that’s useful for guaranteeing identifiability. For instance, in section 4.2, we define fractured subsets which is a testable necessary condition of imaginary subsets. This is made possible by our precise notion of imaginary subsets. The main technical challenge with isolated edges is to identify whether they are the only non-orientable edges under unknown interventions, which we show by introducing isolated equivalence classes (section 5).

---

> > ### Comment · Reviewer_3v5g · 2023-08-16
> > **Comment**
> >
> > Thank you for your answers which clarify my questions and improve a lot the content of your paper. Please:
> > - include in the main body of your paper the assumptions detailed in appendices C and D,
> > - add a small discussion that details your answer (about how your model deals with latent variables, etc.), and
> > - add the description of the main technical challenge to the problem description.
> > Other than that. I don't have further suggestions other than what the other reviewers pointed out.

---

> > > ### Author Response · Authors · 2023-08-17
> > >
> > > Thank you for taking the time to read our response, and we will definitely include these changes in our updated version. Your suggestions have helped us clarify our main ideas better.
> > >
> > > Do you have any further questions or clarifications? If you are satisfied with the response, we hope you will consider increasing the score.

---

> > > > ### Comment · Reviewer_3v5g · 2023-08-18
> > > >
> > > > I will increase the score indeed. Again, thank you and I don't have further suggestions.

---

> > > > > ### Author Response · Authors · 2023-08-19
> > > > >
> > > > > We really appreciate you taking your time to carefully review our paper. We are glad to know that your concerns have been addressed and you will increase your score. Thanks for your support.

---

### Official Review · Reviewer_ZNv1 · 2023-07-27

**Soundness:** 3 good
**Presentation:** 2 fair
**Contribution:** 3 good
**Rating:** 6
**Confidence:** 1

**Summary:**

This paper studies the problem of nonparametrically constructing latent causal graphs given unknown _interventions_ in a measurement model. In particular the paper given a constructive proof of establishing sufficient conditions under which one can find the latent causal structure between observed and hidden variables.

Authors achieve this by first defining a notion of *imaginary subsets,* which is a subset of observed variable that are not children of a single latent variable. Authors show that if a graph does not have any imaginary subsets then up to some assumptions on the graph, one can identify a causal structure. Author further give sufficient conditions that guarantee there don't exist imaginary subsets in the graph.

**Update**: After reading other reviews and rebuttals. I have updated the score from 5 to 6.

**Strengths:**


- Authors identify sufficient conditions for when causal structure is possible and construct a causal graph under this setting.
- Paper is thorough and identify limitations with respect to  limitations of the proposed construction.

**Weaknesses:**

- Overall though the paper is difficult to follow for non-experts. It heavily relies on jargon that is used commonly in causal inference and thus proves to be a difficult read. For instance from the abstract authors use unknown *intervention* starting in the abstract which is not defined much later in the paper.
- The paper can really benefit from a running example that could guide the reader through all the definitions because in the current state one it is difficult to follow.

**Questions:**

In theorem 3.5, one needs to know $\{ P^{(I)}}_{I\in \mathcal{I}}$. Isn’t this a very strong assumption as one needs to exactly know the intervention distributions for all hidden variables since $\mathcal{I} = \{0, \{H_1\}, …, \{H_m\}\}$.

**Limitations:**

Authors have addressed limitations of the work.

---

> ### Author Rebuttal · Authors · 2023-08-10
>
> Thanks for your review!
>
> **“Overall though the paper is difficult to follow for non-experts. The paper can really benefit from a running example that could guide the reader through all the definitions because in the current state one it is difficult to follow.”**
>
> Thanks for the suggestion! “Intervention” is a standard term in the causal inference community. We actually use Figure 1 as our running example which covers most of our key graphical concepts. We will be happy to add some more detail about this example in the camera ready.
>
> **“Theorem 3.5”**
>
> This is a common assumption and in fact, it is an open question whether or not this can be relaxed [a-b]. [a] also shows that this assumption is necessary (Section 3.3 of [a]).
>
> Since our focus is on the nonparametric side, we have not yet looked into optimizing the number of interventions. The complete family of intervention targets is needed to guarantee exact latent graph recovery (appendix C.4), but one might be able to relax these assumptions to get partial recovery. On the other hand, knowing all the interventional distributions does not mean we know the interventional target. In fact, we argue in the paper that two interventional distributions could be the same and thus we might not even know the number of latents.
>
> Moreover, compared to classical work on interventions in graphical models, the current literature on causal representation learning (including our submission as well as [a,b]) considers a strictly harder setting since the interventions are both latent and unknown. Since the intervention target is latent, we only have access to partial information (observed variables) regarding the effect of interventions. And since the intervention target is unknown, compared to the known intervention target, we additionally have the unorientability problem of isolated edges which is discussed in section 5.2 (L357).
>
> [a] Seigal, Squires, Uhler. "Linear causal disentanglement via interventions." arXiv preprint arXiv:2211.16467 (2022).
>
> [b] Varici, Burak, et al. "Score-based causal representation learning with interventions." arXiv preprint arXiv:2301.08230 (2023).

---

> > ### Comment · Reviewer_ZNv1 · 2023-08-18
> > **Response to the Rebuttal**
> >
> > Thank you for the explanations. After carefully reading other reviews and corresponding rebuttals, I have updated my score on the submission.

---

> > > ### Author Response · Authors · 2023-08-18
> > >
> > > Thank you for your support and for taking the time to thoroughly evaluate our submission!

---

### Author Rebuttal · Authors · 2023-08-10

We want to thank all the reviewers for their thoughtful comments and suggestions! We also want to address some common questions.

**Assumptions**

Two reviewers have questions about the real-world relevance of our assumptions (bZuh, tfcU). First of all, we completely agree with Reviewer tfcU that negative results are equally valuable. One of the main objectives of this paper is to study the theoretical limits of nonparametric identifiability of latent causal graphs. The fact that our assumptions are tight (L164-176 and Appendix C) is crucial: Any hope of relaxing these assumptions will require making alternative assumptions. We show what is possible and impossible under this setting and why additional assumptions are needed.

In particular, we show in the paper why assumptions 1(c) and 1(d) are necessary for our problem (appendix c.2 and c.3). A practical example where such assumptions might be satisfied is topic modeling. In topic modeling, one often assumes the existence of pure children (or anchor words) [a]. But our 1(c) and 1(d) assumptions are strictly weaker assumptions than the pure child assumption (Remark 4.2) and we show how to identify under the pure child assumption even with imaginary subsets (section 4.3, theorem 4.8).

[a] Arora, Sanjeev, et al. "A practical algorithm for topic modeling with provable guarantees." International conference on machine learning. PMLR, 2013.

**Experiments**

Three reviewers have questions about limitations of experiments (3v5g, bZuh, tfcU), although Reviewer 3v5g does agree that the paper is “interesting from a theoretical perspective”. We also appreciate Reviewer Yp1n for acknowledging that the theoretical contribution of this paper is “already sufficient on its own”.

Since our paper is primarily theoretical, the purpose of our experiments is simply to verify the theory, illustrate that it is easy to implement, and serve as a proof of concept, as Reviewer tfcU has suggested. It is likely that other methods which use functional assumptions might outperform our method if the additional functional information (i,e., linearity) is available. Nevertheless, our theory is more general and applies to arbitrary nonlinearities.

We also want to apologize to Reviewer Yp1n for the confusion at L377. To clarify, we simply mean that we did not enforce Assumptions 1 and maximality strictly in the experiments, and in spite of this, the method still performs well. We will modify this sentence to say: “The empirical results show that our method is robust at recovering DAGs with low errors when the graphs are generated with sparsity.” We suspect this is a combination of two factors: 1) The simulated DAGs are sparse, which intuitively suggests they are likely (but not guaranteed) to satisfy Assumption 1, and 2) There is some mild robustness to misspecification. Since we know the assumptions cannot be relaxed, we know that violations of the assumptions must lead to nonidentifiability, however, the _severity_ of nonidentifiability could be mild (we stress that we do not have any justification of this beyond the encouraging results of the simulations). This is an interesting observation we look forward to investigating more deeply in future work.

---

### Author Response · Authors · 2023-08-17

Dear reviewers: As the discussion comes to a close, we would like to check in to see if there are any more questions or clarifications we can provide. We are happy to engage in additional discussions, and provide further clarification if there are any remaining questions. Once again, thank you for the consideration!

---

### Decision · Program_Chairs · 2023-09-21

**Decision:**

Accept (poster)

**Comment:**

This work addresses the rapidly blossoming theory of identifying latent causal models, considering the setting with nonparametric assumptions. In the rebuttal, the concerns of the reviewers were addressed, so I recommend acceptance. Please incorporate the changes requested by the reviewers in the camera ready revision.